# Tight bounds for maximum $\ell_1$-margin classifiers

## Abstract

Popular iterative algorithms such as boosting methods and coordinate descent on linear models converge to the maximum $\ell_1$-margin classifier, a.k.a. sparse hard-margin SVM, in high dimensional regimes where the data is linearly separable. Previous works consistently show that many estimators relying on the $\ell_1$-norm achieve improved statistical rates for hard sparse ground truths. We show that surprisingly, this adaptivity does not apply to the maximum $\ell_1$-margin classifier for a standard discriminative setting. In particular, for the noiseless setting, we prove tight upper and lower bounds for the prediction error that match existing rates of order $\frac{\|w^*\|_1^{2/3}}{n^{1/3}}$ for general ground truths. To complete the picture, we show that when interpolating noisy observations, the error vanishes at a rate of order $\frac{1}{\sqrt{\log(d/n)}}$. We are therefore first to show benign overfitting for the maximum $\ell_1$-margin classifier.

## 1 Introduction

The ability to generalize in high-dimensional learning tasks is crucially based on structural assumptions on the underlying ground truth. Probably the most commonly studied assumption is that the observations only depend on few input features, also called sparsity of the ground truth. Popular iterative algorithms widely used in practice to train models in such settings include coordinate descent (see Wright (2015) for a survey) and boosting methods (e.g., Adaboost Freund and Schapire (1997)). Numerous influential works (Bartlett et al., 1998; Rudin et al., 2004; Zhang and Yu, 2005; Shalev-Shwartz and Singer, 2010; Schapire and Freund, 2013; Telgarsky, 2013; Gunasekar et al., 2018) make an important step towards mathematically understanding these algorithms by showing that these solutions have the implicit bias of converging to the maximum $\ell_1$-margin classifier (sparse hard-margin SVM).

However, so far, there exists relatively little analysis on the generalization capabilities of the maximum $\ell_1$-margin classifier; existing nonasymptotic results only consider general (non-sparse) ground truths and adversarial corruptions (Chinot et al., 2021), while asymptotic results consider regimes where the prediction error does not vanish (Liang and Sur, 2022). In this paper, we derive tight matching upper and lower bounds for the prediction error in a high-dimensional discriminative classification setting with (hard) sparse ground truths. Our theory holds for Gaussian covariate distributions and the tightness of our bounds crucially rely on Gaussian comparison results (Gordon, 1988; Thrampoulidis et al., 2015) (for comparison with previous work, see Section 3.3). Our tight non-asymptotic bounds allow us to answer two open problems regarding maximum $\ell_1$-margin classifier related to its adaptivity to sparsity (Problem 1) and benign overfitting (Problem 2).

**Problem 1: adaptivity to sparsity** Intuitively, linear estimators relying on the $\ell_1$-norm should *adapt* to (hard) sparse ground truths by achieving faster rates than for ground truths where only the $\ell_1$-norm is bounded. For instance, this gap has been proven for $\ell_1$-norm penalized maximum average margin classifiers (Zhang et al., 2014), as well as basis pursuit (which achieves exact recovery only under sparsity assumptions (Donoho, 2006; Candes and Tao, 2006)) and the LASSO (Tibshirani, 1996; Van de Geer, 2008) in linear regression settings.

However, so far there are no results in the literature that show adaptivity to sparsity of the (interpolating) maximum $\ell_1$-margin classifier in high-dimensional discriminative learning tasks. In fact, recent work (Chinot et al., 2021) posed the following open problem:

*(Q1): Is the maximum $\ell_1$-margin classifier adaptive to sparsity for noiseless data?*

In Section 3.1 we show that surprisingly, the answer is negative: The tight rate $\frac{\|w^*\|_1^{2/3}}{n^{1/3}}$ for (hard-) sparse normalized ground truths $w^*$ in Theorem 1 is of the same order as the upper bounds in (Chinot et al., 2021) that hold for general ground truths.

**Problem 2: benign overfitting**   Motivated by empirical observations for largely over-parameterized models (Zhang et al., 2021; Belkin et al., 2019), a line of research recently emerged showing "benign overfitting" (Bartlett et al., 2020) for linear interpolating classifiers. More specifically, these papers show that the prediction error yields vanishing rates, although the model interpolates noisy observations, where a constant fraction of labels are randomly corrupted (Muthukumar et al., 2021; Donhauser et al., 2022; Shamir, 2022).

However, so far no such results exist for the maximum $\ell_1$-margin classifier. Existing upper bounds in (Chinot et al., 2021) are tight for arbitrary (adversarial) corruptions but require the fraction of corrupted labels to go to zero to reach vanishing rates. It is unclear whether these rates can be improved for random (non-adversarial) corruptions:

*(Q2): Does the prediction error for the maximum $\ell_1$-margin classifier yield vanishing rates when a constant fraction of the labels are randomly corrupted?*

In Section 3.2, we show that this is indeed true: The maximum $\ell_1$-margin classifier achieves a logarithmic rate of order $\frac{1}{\sqrt{\log(d/n)}}$ in Theorem 2 — which is much slower than for the noiseless case and far from being minimax optimal (Wainwright, 2009; Abramovich and Grinshtein, 2018), but nonetheless vanishing in high-dimensional regimes when $d > n^{1+\epsilon}$. We therefore complement the literature on benign overfitting for maximum $\ell_p$-margin classifiers with $p > 1$, which can even achieve much faster polynomial rates (Donhauser et al., 2022).

## 2   Setting

In this section we introduce the data model, prediction error and maximum $\ell_1$-margin classifier. We study a standard discriminative data model which is commonly studied in the 1-bit compressed sensing literature (see e.g., Boufounos and Baraniuk (2008); Plan and Vershynin (2012) and references therein).

We assume that we observe $n$ pairs of i.i.d. input features $x_i \overset{\text{i.i.d.}}{\sim} \mathcal{N}(0, I_d)$ and associated labels $y_i = \mathrm{sgn}(\langle x_i, w^* \rangle)\xi_i$ where $w^*$ is the (normalized) ground truth (i.e., $\|w^*\|_2 = 1$). Unlike previous works (Chinot et al., 2021), our proofs crucially rely on the Gaussianity of the input features (see Section 3.3 for a comparison with existing proof techniques). We say that the label $y_i$ is clean if $\xi_i = 1$ and corrupted if $\xi_i = -1$. We study the two cases where either all labels are clean (noiseless), i.e. $\forall i:\ \xi_i = 1$, or where the corruptions $\xi_i \in \{-1, 1\}$ are randomly drawn from a distribution $\mathbb{P}_\sigma$ (noisy) only depending on the features in the direction of the ground truth:

$$\xi_i | x_i \overset{\text{i.i.d.}}{\sim} \mathbb{P}_\sigma(\cdot; \langle x_i, w^* \rangle). \tag{1}$$

As proposed in (Donhauser et al., 2022), we make the following technical assumption on the noise distribution $\mathbb{P}_\sigma$:

**Assumption 1** (Noise model). *The function $z \mapsto \mathbb{P}_\sigma(\xi = 1; z)$ is a piece-wise continuous function such that the minimum $\nu_f := \underset{\nu}{\arg\min}\ \mathbb{E}_{Z \sim \mathcal{N}(0,1)} \mathbb{E}_{\xi \sim \mathbb{P}_\sigma(\cdot; Z)} (1 - \xi\nu|Z|)_+^2$ exists and is positive $\nu_f > 0$.*

This assumption is rather weak and satisfied by most noise models in the literature, such as

- *Logistic regression* with $\mathbb{P}_\sigma(\xi_i = 1; z) = h(z\sigma)$ and $h(z) = \frac{e^{|z|}}{1 + e^{|z|}}$ and $\sigma > 0$.

- *Random label flips* with $\mathbb{P}_\sigma(\xi = 1; \langle x_i, w^\star \rangle) = 1 - \sigma$ and $\sigma \in (0, \frac{1}{2})$.

- *Random noise before quantization* where $y_i = \mathrm{sgn}(\langle w^*, x_i \rangle + \tilde{\xi}_i)$ with $\tilde{\xi}_i | x_i \sim \mathcal{N}(0, \sigma^2)$ and $\sigma^2 > 0$.

Given the data set $\{(x_i, y_i)\}_{i=1}^n$, the goal is to obtain an estimate $\hat{w}$ that directionally aligns with the normalized ground truth $w^*$ and thus has a small prediction error:

$$\mathrm{R}(\hat{w}) := \mathbb{E}_{x \sim \mathcal{N}(0, I_d)} 1[\mathrm{sgn}(\langle x, \hat{w} \rangle) \neq \mathrm{sgn}(\langle x, w^* \rangle)] = \frac{1}{\pi} \arccos\left( \left\langle \frac{\hat{w}}{\|\hat{w}\|_2}, w^* \right\rangle \right). \tag{2}$$

By the Taylor series approximation, one can directly see that a small prediction error corresponds to a small directional estimation error, which is commonly studied in the 1-bit compressed sensing literature (Boufounos and Baraniuk, 2008) since

$$\mathrm{R}(\hat{w}) \approx \frac{1}{\pi} \left\| \frac{\hat{w}}{\|\hat{w}\|_2} - w^* \right\|_2. \tag{3}$$

We study the *maximum $\ell_1$-margin interpolators*, or equivalently, the *sparse hard-margin SVM* solution defined by

$$\hat{w} = \arg\min_w \|w\|_1 \quad \text{s.t} \quad \forall i : \ y_i \langle x_i, w \rangle \geq 1.$$

**Remark 1.** *While our two main results in Section 3, Theorem 1 and 2, are stated for the maximum $\ell_1$-margin classifier, the bounds in the theorems hold uniformly for all interpolating classifiers with large (close to the optimal) $\ell_1$-margin (see Proposition 5 and 7)*

## 3 Main Results

In this section we state our main result for the noiseless (Theorem 1 in Section 3.1) and noisy setting (Theorem 2 in Section 3.2). For both results, we assume that the data is distributed as described in Section 2. Furthermore, we present a discussion comparing our main results with existing results based on hyperplane tessellation in Section 3.3.

### 3.1 Main result for noiseless observations

Our first main result stated in the following theorem provides tight upper and lower bounds in the noiseless setting:

**Theorem 1** (Noiseless classification)**.** *Assume that $\forall i$, $\xi_i = 1$ and $w^*$ is a $s$-sparse vector with $s \leq n^{2/3} \log^{-14/3} d$. There exist universal constants $\kappa_1, \kappa_2, \kappa_3, c_1, c_2, c_3 > 0$ such that for any $n \geq \kappa_1$ and $\kappa_2 m_n \leq d \leq \exp(\kappa_3 n^{1/12})$, the prediction error is upper- and lower-bounded by*

$$\left| \mathrm{R}(\hat{w}) - \left( \frac{\kappa_0 \|w^*\|_1^2}{n \log^{1/2}(d/m_n)} \right)^{1/3} \right| \lesssim \left( \frac{\|w^*\|_1^2}{n \log(d/m_n)} \right)^{1/3},$$

*with probability at least $1 - c_1 d^{-1} - c_2 \exp\left( -c_3 \frac{n^{1/3}}{\log^4(d/m_n)} \right)$ over the draws of the data set where we define $\kappa_0 = \frac{8}{\sqrt{3}\pi^{5/2}}$ and $m_n \asymp (n \|w^*\|_1)^{2/3} \log^{1/3}(d/(n \|w^*\|_1)^{2/3})$ (the exact expression is given in Equation (14) in Section B).*

The proof of the theorem is deferred to Appendix B and an overview is given in Section 4. Furthermore, we refer to Appendix 3.4 for a discussion of the assumptions. We now discuss the implications of the theorem in the following paragraphs.

**Adaptivity to sparsity** Existing upper bounds (Chinot et al., 2021) for the maximum $\ell_1$-margin classifier hold for any normalized ground truth $w^\star$ (with $\|w^\star\|_2 = 1$) and are of order $\mathrm{R}(\hat{w}) = O\left( \frac{\|w^\star\|_1^2}{n} \right)^{1/3}$ up to logarithmic factors. Our matching upper and lower bounds in Theorem 1 show that these rates can only be improved by logarithmic factors under the assumption that the ground truth is sparse. Maybe unexpectedly, we therefore conclude that the maximum $\ell_1$-norm classifier cannot (or only very mildly) adapt to sparsity of the ground truth!

**Suboptimality of maximum $\ell_1$-margin**  This lack of adaptivity stands in stark contrast to other $\ell_1$-norm constrained classifiers from the one-bit CS literature that can e.g. achieve rates of order $\frac{\|w^*\|_0 \log(d)}{\sqrt{n}}$ under sparsity assumptions (e.g., Zhang et al. (2014); Awasthi et al. (2016)). We remark that even faster min-max optimal bounds of order $\frac{\|w^*\|_0 \log(d)}{n}$ can be obtained by other classifiers (Gopi et al., 2013; Jacques et al., 2013). Intuitively, the reason for the suboptimality of the rates for the maximum $\ell_1$-margin classifier can be explained by the fact that the ground truth $w^*$ has a small margin of order $\Theta(\frac{1}{\sqrt{n}})$ with high probability, while the maximum $\ell_1$-margin classifier has a larger margin at least of order [1] $\Omega(\frac{1}{(n\|w^*\|_1)^{1/3}})$. That is, the max-$\ell_1$-margin classifier overfits to samples close to the decision boundary.

### 3.2  Main result for noisy observations

Our second main result considers the high noise regime where a constant fraction of the labels are (randomly) corrupted with high probability. We show in the following theorem that the prediction error vanishes for this setting at a logarithmic rate:

**Theorem 2** (Noisy classification)**.** *Assume that the corruptions $\xi_i$ follow the law in Equation (1) with $\mathbb{P}_\sigma$ independent of $n, d$ and satisfy Assumption 1. Furthermore, assume that $w^*$ is $s$-sparse with $s \lesssim n/\log^4(d/n)$. There exist universal constants $\kappa_1, \kappa_2, \kappa_3, c_1, \ldots, c_4 > 0$ such that for any $n \geq \kappa_1$ and $\kappa_2 n \leq d \leq \exp(\kappa_3 n^{1/5})$, the prediction error is upper- and lower-bounded by*

$$\left| \mathrm{R}(\hat{w}) - \sqrt{\frac{\kappa_\sigma}{\log(d/n)}} \right| \lesssim \frac{1}{\log^{3/4}(d/n)} \, ,$$

*with probability at least $1 - c_1 \exp\left(-c_2 \frac{n}{\log^5(d/n)}\right) - c_3 \exp\left(-c_4 \frac{n}{\log n \log^{3/2}(d/n)}\right)$ over the draws of the data set and with $\kappa_\sigma$ a constant only depending on $\mathbb{P}_\sigma$ (see Equation (31) in Appendix C for the definition).*

The proof of the theorem is deferred to Appendix C and an overview is given in Section 4. Furthermore, we refer to Appendix 3.4 for a discussion of the assumptions. We now discuss the implications of the theorem in the following paragraphs.

**Benign overfitting:**  We are the first to show that the prediction error of the max-$\ell_1$-margin classifier vanishes albeit interpolating a constant fraction of (randomly) corrupted labels, and thus exhibits benign overfitting Bartlett et al. (2020). Therefore, our work complements recent work studying maximum $\ell_p$-margin classifiers with $p > 1$ that can achieve polynomial rates (Donhauser et al., 2022).

**Comparison with optimal rates:**  Although vanishing, the rates in Theorem 2 are only of logarithmic orders and therefore far from being min-max optimal. Indeed min-max optimal lower bounds for the noisy setting are of order $\frac{\|w^*\|_0 \log(d)}{\sqrt{n}}$ (Wainwright, 2009; Abramovich and Grinshtein, 2018) and attained by regularized (non-interpolating) classifiers maximizing the average margin under $\ell_1$-norm constraints (see, e.g., (Zhang et al., 2014)). Theorem 2 can therefore also be understood as a negative result showing that the maximum $\ell_1$-margin classifier suffers from overfitting the noise, in the sense that, although consistent, the rates are far from min-max optimal.

### 3.3  Comparison with bounds relying on hyperplane tessellation

We now discuss the limitations of proofs relying on hyperplane tessellation (see e.g. Plan and Vershynin (2014)) – a standard tool to bound the prediction error of linear classifier in high-dimensional settings, e.g. in (Chinot et al., 2021).

First, define the Hamming distance of two vectors $w_1, w_2$ to be the fraction of training samples where the corresponding classifiers differ:

$$d_H(w_1, w_2) = \frac{1}{n} \sum_i \mathbb{1}\{\mathrm{sign}(\langle x_i, w_1 \rangle) \neq \mathrm{sign}(\langle x_i, w_2 \rangle)\}.$$

---

[1] where we make use of Proposition 4 and Lemma 4.1 in Chinot et al. (2021)

Note that $d_H(\hat{w}, w^\star)$ corresponds exactly to the fraction of corrupted labels i.e., $d_H(\hat{w}, w^\star) = \frac{1}{n}\sum_i 1\{\xi_i = -1\}$. The high-level idea of hyperplane tessellation is to bound the directional estimation error (3) (which in turn gives a bound on the prediction error (2)) via the Hamming distance by uniformly bounding the difference between the Euclidean and scaled Hamming distance

$$\sup_{w_1, w_2 \in T} |\lambda d_H(w_1, w_2) - \|w_1 - w_2\|_2|, \qquad (4)$$

over some large enough set $T \subset S^{d-1}$ that contains the normalized classifier $\frac{\hat{w}}{\|\hat{w}\|_2}$ with high probability. Here, $\lambda$ is some universal constant.

Observe that this approach only leads to tight bounds if the difference in Equation (4) is small. This, however, is not the case for the settings studied in our main results. Indeed, for noisy data (Theorem 2), by definition of the interpolating classifier we have that

$$\lambda d_H\left(\frac{\hat{w}}{\|\hat{w}\|_2}, w^*\right) = \Theta(1)$$

while $\|\frac{\hat{w}}{\|\hat{w}\|_2} - w^*\|_2$ vanishes at a logarithmic rate. Furthermore, in the noiseless case (Theorem 1), the Hamming distance $d_H\left(\frac{\hat{w}}{\|\hat{w}\|_2}, w^*\right)$ is zero — meaning that we cannot obtain any lower bounds for the directional estimation error using a hyperplane tessellation argument.

This "weakness" of proofs relying on uniform hyperplane tessellation bounds is also not surprising since such approaches do not take the distributional assumptions of the noise into account — in particular, we cannot distinguish between adversarial and non-adversarial noise. In contrast, the logarithmic rates in Theorem 2 crucially rely on Assumption 1 for the distribution of the corruptions.

In defense of hyperplane tessellation bounds, we finally mention that unlike the proofs presented in this paper (see Section 4), results relying on hyperplane tessellation bounds give guarantees for arbitrary corruptions and can also be generalized to non-Gaussian features (Chinot et al., 2021). Yet, in order to capture the rates in Theorem 1 and 2, new proof techniques are needed.

### 3.4 Discussion of the assumptions in Theorem 1 and 2

In this section we discuss the assumptions in our main theorems on the sparsity of the ground truth and the data distribution and their limitations.

**Sparsity of the ground truth** $w^*$ While the upper bound in Theorem 1 can be generalized at the cost of a logarithmic factor (i.e. as in Chinot et al. (2021)), the lower bound requires a very tight analysis (proof of Proposition 5 in Appendix B.2) and strongly relies on the sparsity of the ground truth. We would like to note at this place that only few high-probability lower bounds are known in the literature (beyond classifiers/regression estimators relying on the $\ell_2$-norm) and leave lower bounds for non-sparse ground truths as an exciting and important future work.

Moreover, we mention that the constraint on the degree of the sparsity of the ground truth in Theorem 2 cannot be relaxed without affecting the upper bound. However, it is an open question whether one can relax the constraint with a soft-sparsity constraint on the ground truth of the form $\|w^*\|_1 \leq \sqrt{\frac{n}{\log(d/n)^4}}$. We note that the bound in Theorem 2 does not depend on the ground truth, assuming that the degree of sparsity is sufficiently small. Morally, this is because the effect of fitting the noise dominates the prediction error, similar to the rates for the prediction error of the minimum-$\ell_1$-norm interpolator (Basis pursuit) in (Wang et al., 2022).

**Gaussian distribution of the data** The assumption that the data is normally distributed is a major limitation of the results presented in Theorem 1 and 2. Attempts to generalize this assumption face the fundamental issue that the analysis needs to be tight including multiplicative constants. However, so far, the only technique that allows us to capture this tightness relies on Gaussian comparison inequalities (e.g.,

the (C)GMT (Gordon, 1988; Thrampoulidis et al., 2015)), and generalizations to non-Gaussian data come at a price of a multiplicative constant.

We further remark that any previous work presenting a tight analysis of min-norm/max-margin interpolators (see, e.g., (Donhauser et al., 2022; Wang et al., 2022; Koehler et al., 2021; Zhou et al., 2022; 2021)) crucially relies on Gaussian input data, with the min-$\ell_2$-norm/max-$\ell_2$-margin interpolators (Bartlett et al., 2020; Muthukumar et al., 2021) as the only exceptions. Deriving tight generalizations of the (C)GMT (Gordon, 1988; Thrampoulidis et al., 2015) to non-Gaussian data is a promising direction for extending our main results, with the first results in this direction in (Han and Shen, 2022).

**Isotropic features** In this paper, we only consider isotropic input features $x_i$. Technically, we believe that our methodology could also be extended to non-isotropic features (see (Koehler et al., 2021; Zhou et al., 2021; 2022) for related works in this direction). However, such an extension comes at the price of technically more involved proofs and theorem statements — and therefore at a cost of readability. We believe that, despite the less general setting, our results already reveal interesting novel insights.

## 4 Proof overview

In this section, we give an overview of the proofs of the main results, Theorem 1 and Theorem 2, and summarize the main tools used in the proof. Both proofs rely on a standard localization/ uniform convergence argument (see e.g., Koehler et al. (2021); Zhou et al. (2021); Wang et al. (2022); Donhauser et al. (2022)), where:

1. *(Localization)* we derive a high-probability upper bound on the $\ell_1$-norm of the maximum $\ell_1$-margin interpolator $\hat{w}$ over the draws of $X$ and $\xi$, by finding $M > 0$ such that

$$\min_{\forall i:\, y_i \langle x_i, w \rangle \geq 1} \|w\|_1 \;=: \Phi_N \;\leq M.$$

2. *(Uniform convergence)* we derive high-probability uniform bounds over $X$ and $\xi$ for all interpolators $w$ with $\|w\|_1 \leq M$. Namely, we find a high-probability lower and upper bound, respectively, for the minimum (maximum) alignment

$$\Phi_- := \min_{\substack{\|w\|_1 \leq M \\ \|w\|_2 \geq \delta}} \frac{\langle w, w^* \rangle}{\|w\|_2} \quad \text{s.t.} \quad \forall i:\, y_i \langle x_i, w \rangle \geq 1,$$

$$\Phi_+ := \max_{\substack{\|w\|_1 \leq M \\ \|w\|_2 \geq \delta}} \frac{\langle w, w^* \rangle}{\|w\|_2} \quad \text{s.t.} \quad \forall i:\, y_i \langle x_i, w \rangle \geq 1$$

   with some $\delta > 0$ arbitrarily small, which in turn gives us high probability bounds for the prediction error using that

$$\mathrm{R}(\hat{w}) = \frac{1}{\pi} \arccos\left( \left\langle \frac{\hat{w}}{\|\hat{w}\|_2}, w^* \right\rangle \right).$$

**Remark 2.** *The constraint $\|w\|_2 \geq \delta$ in the definition of $\Phi_+, \Phi_-$ is only added to ensure the optimization problems are well defined. In particular, we can choose $\delta > 0$ arbitrarily small and, therefore, neglect this constraint in the remainder of the analysis.*

The remainder of this section is structured as follows. We first present in Section 4.1 an application of Gaussian comparison (Proposition 1), which allows us to reduce the optimization problems $\Phi_N, \Phi_-$ and $\Phi_+$ to simpler auxiliary optimization problems $\phi_N, \phi_-$ and $\phi_+$. We then describe in Section 4.2 how these auxiliary optimization problems can be further simplified using the localized Gaussian width (Proposition 2). Finally, in Section 4.3, we give a sketch of the remaining proofs of Theorem 1 and Theorem 2.

**Notation** We define the function $(\cdot)_+ : \mathbb{R} \to \mathbb{R}_+$, $(x)_+ = x \mathbf{1}\{x \geq 0\}$. We denote by $s$ the sparsity ($\ell_0$-norm) of $w^*$ and assume w.l.o.g. that the nonzero entries of $w^*$ are exactly the first $s$-entries. Moreover, we use the following notation for components of the vector $w$: $w_\parallel \in \mathbb{R}^d$ and $w_\perp \in \mathbb{R}^d$ for components parallel and perpendicular to $w^*$, respectively. Furthermore, we use $w_\perp^{(\mathcal{S})} \in \mathbb{R}^s$ for the first $s$-entries of $w_\perp$, and $w_\perp^{(\mathcal{S}^c)} \in \mathbb{R}^{d-s}$ for the last $d - s$ entries of $w_\perp$.

We denote by $B_1, B_2$ unit balls with respect to the $\ell_1$ and $\ell_2$-norms, respectively. We use $\kappa_1, \kappa_2, \ldots$ and $c_1, c_2, \ldots$ for generic universal positive constants independent of $d$, $n$, whose values may change from display to display throughout the derivations. The standard notations $O(\cdot), o(\cdot), \Omega(\cdot), w(\cdot)$ and $\Theta(\cdot)$, as well as $\lesssim, \gtrsim$ and $\asymp$, are utilized to hide universal constants, without any hidden dependence on $d$ or $n$.

### 4.1 Preliminary Step 1: application of the (C)GMT

The proofs of both main results rely on the following application of the Gaussian Minmax Theorem (GMT) (Gordon, 1988) and its convex variant (CGMT) (Thrampoulidis et al., 2015), which is a commonly used tool when studying linear min-norm/max-margin interpolators (see e.g., (Deng et al., 2021; Donhauser et al., 2022; Koehler et al., 2021; Zhou et al., 2021; Wang et al., 2022)).

**Recap: (C)GMT** For completeness, we first summarize the following variant of the (C)GMT.

**Lemma 1.** *(Corollary of (Gordon, 1988; Thrampoulidis et al., 2015)) Let $X_1 \in \mathbb{R}^{n \times d-s}$ be a matrix with i.i.d. $\mathcal{N}(0,1)$ entries and let $g \sim \mathcal{N}(0, I_n)$ and $h \sim \mathcal{N}(0, I_{d-s})$ be independent random vectors. Let $S_w \subset \mathbb{R}^s \times \mathbb{R}^{d-s}$ and $S_v \subset \mathbb{R}^n$ be compact sets, and let $\psi : S_w \times S_v \to \mathbb{R}$ be a continuous function. Then for the following two optimization problems:*

$$\Phi = \min_{(w_1, w_2) \in S_w} \max_{v \in S_v} \langle v, X_1 w_1 \rangle + \psi((w_1, w_2), v)$$

$$\phi = \min_{(w_1, w_2) \in S_w} \max_{v \in S_v} \|w_1\|_2 \langle v, g \rangle + \|v\|_2 \langle w_1, h \rangle + \psi((w_1, w_2), v)$$

*and any $t \in \mathbb{R}$ holds that:*

$$\mathbb{P}(\Phi < t) \leq 2\mathbb{P}(\phi \leq t)$$

*If in addition $\psi$ is a convex-concave function, we also have for any $t \in \mathbb{R}$:*

$$\mathbb{P}(\Phi > t) \leq 2\mathbb{P}(\phi \geq t)$$

*In both inequalities the probabilities in the LHS and RHS are over the draws of $X_1$, and of $g$, $h$, respectively.*

We see that $\phi$ controls the upper and lower tail of $\Phi$. Importantly, the inequality is sharp, including multiplicative constants — a high probability upper (lower) bound for $\phi$ is also a high probability upper (lower) bound for $\Phi$. Moreover, $\phi$ no longer depends on a random matrix $X_1$ but only on two random vectors $g$ and $h$, which substantially simplifies the search for bounds for $\phi$ compared to $\Phi$.

**Application of the (C)GMT** We can now use Lemma 1 to simplify the problem of bounding the maximum norm $\Phi_N$ and the minimum (maximum) alignment $\Phi_-, \Phi_+$. For this, we first define the corresponding auxiliary optimization problems $\phi$. Let $z^{(1)}, z^{(2)} \in \mathbb{R}^n$, $h_1 \in \mathbb{R}^s$, $h_2 \in \mathbb{R}^{d-s}$ be i.i.d. isotropic zero mean unit variance Gaussian random vectors and define the function $f_n : \mathbb{R} \times \mathbb{R}_+ \to \mathbb{R}_+$,

$$f_n(\nu, \eta) = \frac{1}{n} \sum_{i=1}^n (1 - \xi_i \nu |z_i^{(1)}| - z_i^{(2)} \eta)_+^2. \tag{5}$$

Similar to the analysis in (Deng et al., 2022) for the related minimum-$\ell_2$-margin classifier, the key insight is now that we can use Lagrange multipliers to apply Lemma 1 to "replace" the data-dependent interpolation constraint $\forall i : y_i \langle x_i, w \rangle \geq 1$ in $\Phi_N, \Phi_-, \Phi_+$ with the simpler constraint,

$$\frac{(\langle w_\perp^{(\mathcal{S})}, h_1 \rangle + \langle w_\perp^{(\mathcal{S}^c)}, h_2 \rangle)^2}{n} \geq f_n(\langle w_\parallel, w^* \rangle, \|w_\perp\|_2). \tag{6}$$

Analyzing this new constraint will make up the heart of the proofs of Theorem 1 and 2. While it will turn out that the constraint in Equation (6) "captures" the interpolation constraint $\forall i : y_i \langle x_i, w \rangle \geq 1$ very well, there is only very limited geometrical intuition for why this is the case.

Formally, we show Proposition 1 (see Appendix D.1 for the proof) for the auxiliary optimization problems:[2]

$$\phi_N = \min_w \|w\|_1 \ \text{ s.t } \ \text{Eq. (6) holds } \text{ and } \ \langle w_\perp^{(\mathcal{S})}, h_1 \rangle + \langle w_\perp^{(\mathcal{S}^c)}, h_2 \rangle \geq 0$$

$$\phi_+ = \max_{\|w\|_2 \geq \delta} \frac{\langle w_\|, w^* \rangle}{\|w\|_2} \ \text{ s.t } \ w \in \tilde{\Gamma} \quad \text{ and } \quad \phi_- = \min_{\|w\|_2 \geq \delta} \frac{\langle w_\|, w^* \rangle}{\|w\|_2} \ \text{ s.t } \ w \in \tilde{\Gamma}.$$

with set $\tilde{\Gamma} \subset \mathbb{R}^d$,

$$\tilde{\Gamma} = \{w \in \mathbb{R}^d \ \text{ s.t } \ \text{Eq. (6) holds and } \ \|w\|_1 \leq M\}.$$

**Proposition 1.** *For any $t \in \mathbb{R}$ we have:*

$$\mathbb{P}(\Phi_N > t|\xi) \leq 2\mathbb{P}(\phi_N \geq t|\xi)$$
$$\mathbb{P}(\Phi_+ > t|\xi) \leq 2\mathbb{P}(\phi_+ \geq t|\xi)$$
$$\mathbb{P}(\Phi_- < t|\xi) \leq 2\mathbb{P}(\phi_- \leq t|\xi),$$

*where the probabilities in LHS and RHS are over the draws of $X$ and of $z^{(1)}$, $z^{(2)}, h_1, h_2$, respectively.*

## 4.2 Preliminary Step 2: simplification of the auxiliary optimization problems

In a second step, we reduce the auxiliary optimization problems $\phi_N, \phi_-$ and $\phi_+$ to low-dimensional optimization problems. While a similar approach has also been used in other papers studying maximum-margin classifiers based on the (C)GMT (see e.g., (Donhauser et al., 2022; Deng et al., 2022; Zhou et al., 2022)), using the reduction in the mentioned papers would only yield loose bounds (not yielding sharp rates). Instead, we propose a much tighter reduction relying on the localized Gaussian width.

**Part 1: $\phi_-$ and $\phi_+$** In order to reduce the two optimization problems to low-dimensional optimization problems, we relax the constraint in Equation (6) by bounding the stochastic term $\langle w_\perp^{(\mathcal{S})}, h_1 \rangle + \langle w_\perp^{(\mathcal{S}^c)}, h_2 \rangle$ only using the $\ell_1$ and $\ell_2$-norms of $w_\perp^{(\mathcal{S})}$ and $w_\perp^{(\mathcal{S}^c)}$. The first term $\langle w_\perp^{(\mathcal{S})}, h_1 \rangle$ can be simply upper-bounded using Cauchy Schwartz: $\langle w_\perp^{(\mathcal{S})}, h_1 \rangle \leq \|h_1\|_2 \|w_\perp^{(\mathcal{S})}\|_2$ where we recall that $h_1 \in \mathbb{R}^s$. However, doing the same for the second term $\langle w_\perp^{(\mathcal{S}^c)}, h_2 \rangle$ would result in loose bounds since $h_2 \in \mathbb{R}^{d-s}$ and $d \gg s$. In fact, using Hoelders inequality to bound $\langle w_\perp^{(\mathcal{S}^c)}, h_2 \rangle \leq \|w_\perp^{(\mathcal{S}^c)}\|_1 \|h_2\|_\infty$ would still result in loose bounds. Instead, we make use of a more refined (tight) upper bound:

$$\langle w_\perp^{(\mathcal{S}^c)}, h_2 \rangle \leq \|w_\perp^{(\mathcal{S}^c)}\|_1 \ell_{h_2}^* \left( \frac{\|w_\perp^{(\mathcal{S}^c)}\|_2}{\|w_\perp^{(\mathcal{S}^c)}\|_1} B_2 \cap B_1 \right) \tag{7}$$

where we use the localized Gaussian width $\ell_{h_2}^* : [\frac{1}{\sqrt{d}}, 1] \to \mathbb{R}_+$ ,

$$\ell_{h_2}^*(\beta B_2 \cap B_1) := \max_{\substack{\|w\|_2 \leq \beta \\ \|w\|_1 \leq 1}} \langle w, h_2 \rangle .$$

As a result, we can now relax the constraint in Equation (6) occurring in $\tilde{\Gamma}$ to:

$$\frac{\left( \|w_\perp^{(\mathcal{S}^c)}\|_1 \ell_{h_2}^* \left( \frac{\|w_\perp^{(\mathcal{S}^c)}\|_2}{\|w_\perp^{(\mathcal{S}^c)}\|_1} B_2 \cap B_1 \right) + \|h_1\|_2 \|w_\perp^{(\mathcal{S})}\|_2 \right)^2}{n} \geq f_n \left( \langle w_\|, w_*^{(\mathcal{S})} \rangle, \sqrt{\|w_\perp^{(\mathcal{S}^c)}\|_2^2 + \|w_\perp^{(\mathcal{S})}\|_2^2} \right).$$

In particular, we note that the resulting relaxed optimization problems for $\phi_-$ and $\phi_+$ only depend on the $\ell_1$ and $\ell_2$-norms of $w_\perp^{(\mathcal{S})}$ and $w_\perp^{(\mathcal{S}^c)}$ and are therefore low-dimensional.

---

[2]We define $\Phi_N, \Phi_-, \phi_N, \phi_- = \infty$ and $\Phi_+, \phi_+ = -\infty$ if the corresponding optimization problems have no feasible solution.

**Part 2:** $\phi_N$   A similar argument can also be used to convert $\phi_N$ into a low-dimensional optimization problem. However, instead of relaxing the constraint in Equation (6), we now need to tighten it. We can do this by setting $w_\perp^{(\mathcal{S})} = 0$ (which is negligible assuming that $s \ll n$) and choosing $w_\perp^{(\mathcal{S}^c)}$ as a function of $\beta$ to be the optimizer of the optimization problem defining $\ell_{h_2}^*(\beta B_2 \cap B_1)$ for which Equation (7) holds with equality.

**Reduction to low dimensional problems**   It will be useful throughout the analysis to slightly change the parameterization: Instead of directly using the localized Gaussian width $\ell_{h_2}^*(\beta B_2 \cap B_1)$, we will use the following (equivalent) curve $\gamma : \alpha \in [1, \alpha_{\max}] \mapsto \gamma(\alpha) \in \mathbb{R}^{d-s}$,

$$\gamma(\alpha) = \arg\min_w \|w\|_2^2 \quad \text{s.t} \quad \begin{cases} \langle w, |h_2| \rangle \geq \|h_2\|_\infty \\ w \geq 0 \\ \|w\|_1 = \alpha \end{cases}, \tag{8}$$

with $\alpha_{\max} = (d-s)\frac{\|h_2\|_\infty}{\|h_2\|_1}$. By Lagrange duality, it is then straightforward to show that for any $\beta \in [\frac{1}{\sqrt{d}}, 1]$, there exists $\alpha \in [1, \alpha_{\max}]$ such that $\frac{\gamma(\alpha)}{\alpha}$ is an optimal solution for the optimization problem that defines $\ell_{h_2}^*(\beta B_2 \cap B_1)$ (see Wang et al. (2022)).

In summary, we obtain the upper (lower) bounds in Proposition 2, where we use the following notation; define $\nu := \langle w_\|, w^* \rangle$, $\eta_{\mathcal{S}^c} := \|w_\perp^{(\mathcal{S}^c)}\|_2$, $\eta_{\mathcal{S}} := \|w_\perp^{(\mathcal{S})}\|_2$, $\eta := \|w_\perp\|_2 = \sqrt{\eta_{\mathcal{S}^c}^2 + \eta_{\mathcal{S}}^2}$ and $b = \frac{\|w_\perp^{(\mathcal{S}^c)}\|_1}{\alpha}$.

**Proposition 2.** *Let $s_{\max} \in \mathbb{N}_+$ and let $w^*$ be any $s$-sparse vector with $s \leq s_{\max}$. Then, the optimization problems $\phi_N, \phi_+$ and $\phi_-$ can be bounded by:*

$$\phi_N \leq \left[ \min_{\nu, b \geq 0, \alpha \in [1, \alpha_{\max}]} |\nu| \|w^*\|_1 + b \|\gamma(\alpha)\|_1 \quad s.t \quad \frac{1}{n} b^2 \|h_2\|_\infty^2 \geq f_n(\nu, b\|\gamma(\alpha)\|_2) \right]$$

$$\phi_+ \leq \max_{(\nu, b, \alpha, \eta_{\mathcal{S}}) \in \Gamma} \frac{\nu}{\sqrt{\nu^2 + b^2 \|\gamma(\alpha)\|_2^2 + \eta_{\mathcal{S}}^2}}$$

$$\phi_- \geq \min_{(\nu, b, \alpha, \eta_{\mathcal{S}}) \in \Gamma} \frac{\nu}{\sqrt{\nu^2 + b^2 \|\gamma(\alpha)\|_2^2 + \eta_{\mathcal{S}}^2}}$$

*where the last two inequalities hold with probability at least $1 - 2\exp(-c_1 s_{\max})$, with universal constant $c_1$, and constraint set $\Gamma$ defined by:*

$$\Gamma = \Big\{ (\nu, b, \alpha, \eta_{\mathcal{S}}) \quad s.t \quad \eta_{\mathcal{S}} \geq 0, b \geq 0, \alpha \in [1, \alpha_{\max}]$$

$$\text{and} \quad \frac{(2\sqrt{s_{\max}}\eta_{\mathcal{S}} + b\|h_2\|_\infty)^2}{n} \geq f_n(\nu, \sqrt{b^2\|\gamma(\alpha)\|_2^2 + \eta_{\mathcal{S}}^2})$$

$$\text{and} \quad \max\left\{ |\nu|\|w^*\|_1 - \sqrt{s}\eta_{\mathcal{S}}, 0 \right\} + b\alpha \leq M \Big\}. \tag{9}$$

The proof follows from the above discussion and by applying Gaussian concentration to control the tail of the term $\|h_1\|_2$.

## 4.3   Proof sketch for bounding the auxiliary optimization problems

We now describe how we obtain the desired bounds in Theorem 1 and 2. Recall that by Proposition 1, it suffices to find high probability bounds for $\phi_N, \phi_-, \phi_+$ using the low-dimensional relaxations in Proposition 2. We now present the main idea for the proof which is rigorously described in Appendices B and C. We only discuss lower bounding $\phi_-$.

**Step 1: reducing the problem to bounding the set** $\Gamma$ We first reduce the problem of bounding $\phi_-$ to one bounding $\Gamma$ in Equation (9) (where we use Proposition 2):

$$\phi_- \geq \left[ 1 + \frac{\max\limits_{(b,\alpha)\in\Gamma} b^2 \|\gamma(\alpha)\|_2^2 + \max\limits_{\eta_\mathcal{S}\in\Gamma} \eta_\mathcal{S}^2}{\min\limits_{\nu\in\Gamma} \nu^2} \right]^{-1/2}. \tag{10}$$

Hence, it suffices to bound the maximum (minimum) of the variables $b^2 \|\gamma(\alpha)\|_2^2$, $\eta_\mathcal{S}^2$ and $\nu^2$. Perhaps surprisingly, this seemingly loose lower bound will turn out to be tight.

**Step 2: controlling** $f_n$ One of the main contributions to the analysis in this paper arises from controlling the function $f_n$ (Equation (5)). To do so, we first show that $\Gamma$ (from Equation (4.1)) is contained in a sufficiently small set. We can then carefully apply concentration arguments to show uniform convergence of $f_n \to \mathbb{E}f_n$. The key insight is then that, using a series expansion, the expectation $\mathbb{E}f_n$ can be approximated by the terms in the following equation:

$$\text{noiseless} \quad \mathbb{E}f_n(\nu,\eta) \approx \frac{\sqrt{2}}{3\sqrt{\pi}}\frac{1}{\nu} + \sqrt{\frac{2}{\pi}}\frac{\eta^2}{\nu} \tag{11}$$

$$\text{noisy} \quad \mathbb{E}f_n(\nu,\eta) \approx \zeta_f + \frac{1}{2}\zeta_{\eta\eta}\eta^2 + \frac{1}{2}\zeta_{\nu\nu}\triangle\nu^2 \tag{12}$$

where $\triangle\nu = \nu - \nu_f$ and $\nu_f, \zeta_{\eta\eta}, \zeta_{\nu\nu}$ are constants arising from the series expansion (only depending on $\mathbb{P}_\sigma$). Moreover, by definition $\eta^2 := b^2\|\gamma(\alpha)\|_2^2 + \eta_\mathcal{S}^2 = \|w_\perp^{(\mathcal{S}^c)}\|_2^2 + \eta_\mathcal{S} := \|w_\perp^{(\mathcal{S})}\|_2^2$.

While the dependency in $\eta$ is quadratic in both cases, the dependency in $\nu$ strongly differs between the noiseless case (Equation (11)) and the noisy case (Equation (12)). To give an intuitive explanation, note that the expectation $\mathbb{E}f_n$ is

$$\mathbb{E}f_n = \mathbb{E}(1 - \xi\nu|z^1| - z^2\eta)_+^2.$$

In the noisy case, by assumption, we have that both $\xi = 1$ and $\xi = -1$ occur with constant (nonvanishing) probability. Therefore, we can lower the bound with a quadratic $\mathbb{E}(1 - \xi\nu|z^1| - z^2\eta)_+^2 \gtrsim (1 + \nu^2 + \eta^2)$. In contrast, in the noiseless case, we have $\xi_i = 1$ a.s. Instead, we lower-bound $(1 - \nu|z^1| - z^2\eta)_+^2 \gtrsim (1 + \eta^2)$ on the event $|z^1| \leq 1/\nu$, which happens with probability inversely dependent on $\nu$. For more details, we refer the reader to the proofs of Lemma 6 and Proposition 8.

**Step 3: bounding the set** $\Gamma$ In the noisy case (Theorem 2), the quadratic approximation from Equation (12) allows us to utilize parts of the analysis in (Wang et al., 2022) for the minimum-$\ell_1$-norm interpolator in regression. For example, we can bound the term $\max_{b,\alpha\in\Gamma} b^2\|\gamma(\alpha)\|_2^2$ from Equation (10) as follows: we can relax the set $\Gamma$ in Equation (9) by replacing the third condition by $b\alpha \leq M$ and using the quadratic form from Equation (12) for the second condition. We then obtain

$$\Gamma \subset \{(\nu, b, \alpha, \eta_\mathcal{S}) \text{ s.t } b\alpha \leq M \quad \text{and} \quad \frac{b^2\|h_2\|_\infty^2}{n} \geq \zeta_f + \frac{1}{2}\zeta_{\eta\eta}(\eta_{\mathcal{S}^c}^2 + b^2\|\gamma(\alpha)\|_2^2) + \frac{1}{2}\zeta_{\nu\nu}\triangle\nu^2\},$$

which resembles the term in Equation (4) in (Wang et al., 2022). In the noiseless case (Theorem 1) such a simplification is not applicable due to the inverse dependency of $\mathbb{E}f_n$ on $\nu$ from Equation (11). In fact, we would only obtain a trivial (loose) bound when again using the relaxation $b\alpha \leq M$ for the third equation in Equation (9). Instead, we need to simultaneously control $(b, \alpha)$ and $\nu$ by iteratively bounding either of them (Appendix B.2), which is the second major technical contribution of the paper.

## 5 Related Work

In this section, we discuss related work on existing bounds for the prediction error of linear maximum-margin classifiers, as well as tools that have so far been used to bound it.

**Related work on error bounds for maximum-margin classifiers** Existing non-asymptotic upper bounds for the maximum $\ell_1$-margin classifier in high-dimensional settings hold for arbitrary (adversarial) corruptions and are discussed in detail in Section 3.3. Furthermore, complementary work (Liang and Sur, 2022) studies asymptotic proportional regimes ($n, d \to \infty$ and $\frac{d}{n} \to c$) where the prediction error does not vanish.

Beyond the $\ell_1$ norm, several works present non-asymptotic bounds for the related maximum $\ell_p$-margin classifiers for $p > 1$. The paper (Donhauser et al., 2022) studies the case where $p \in (1, 2)$ for 1-sparse ground truths and shows that the prediction error can even vanish at polynomial rates close to the min-max lower bounds when trained on a noisy dataset. Furthermore, the papers (Muthukumar et al., 2021; Wang et al., 2021; Shamir, 2022) present bounds for the case where $p = 2$ based on specific proof techniques relying on the geometry of the Euclidean $\ell_2$ norm. However, they only obtain vanishing rates, i.e. achieve benign overfitting, when assuming that the covariance matrix is spiked (i.e., for non-isotropic features).

**Related work on proof techniques** The proofs in this paper rely on Gaussian comparison results (Gordon, 1988; Thrampoulidis et al., 2015) described in detail in Section 4 and popularized for non-asymptotic bounds for linear interpolators in (Koehler et al., 2021). This technique has also recently been used in the paper (Donhauser et al., 2022) to bound the prediction error of the maximum $\ell_p$-margin classifier when $p \in (1, 2)$. However, the analysis presented in the mentioned paper would yield loose bounds when $p = 1$ and is limited to noisy regimes and 1-sparse ground truths.

Other common proof techniques for bounding the prediction error of interpolating linear classifiers include hyperplane tessellation bounds (Plan and Vershynin, 2014; Chinot et al., 2021), discussed in detail in Section 3.3, and proliferation of support vector results (Muthukumar et al., 2021; Hsu et al., 2021; Wang et al., 2021; Ardeshir et al., 2021). The idea of the latter approach is essentially to reduce the maximum-margin classifier to an (approximately) equivalent minimum-norm interpolating classifier. The resulting "simpler" classifier can then be analyzed using tools from regression (Muthukumar et al., 2021; Bartlett et al., 2020). However, so far, such an approach only exists for the maximum $\ell_2$-margin classifiers, and it is an open conjecture to prove that proliferation of support vector results also apply to the maximum $\ell_1$-margin classifier (Ardeshir et al., 2021).

# 6 Future work

In this section, we discuss potentially interesting avenues for future work.

**Early stopped coordinate descent** The bounds presented in this paper imply that the maximum $\ell_1$-margin classifier are not only only sub-optimal in noisy settings (Theorem 2), but also for noiseless data (Theorem 1). As discussed in Section 3.1, this is because the classifier overfits on samples close to the decision boundary. In contrast, $\ell_1$-norm penalized classifiers which maximize the average margin (Zhang et al., 2014) achieve much faster rates than $\|w^*\|_1^{2/3} n^{-1/3}$. An interesting question for future work is whether these faster rates can be obtained for early stopped coordinate descent on exponential losses, where we recall that the solutions of these algorithms converge (after infinite steps) to the maximum $\ell_1$-margin classifier (Telgarsky, 2013).

**Future work on "better" implicit biases** When samples in the training data have a small margin to the ground truth (see discussion in Section 3.1), our results in this paper suggest that the implicit bias of boosting methods with exponential loss functions and coordinate descent is suboptimal. Indeed, the maximum $\ell_1$-margin classifier which is obtained at convergence (Telgarsky, 2013) only achieves suboptimal rates even in the noiseless setting (see Theorem 1 and subsequent discussion). An interesting direction for future work is therefore to investigate whether the implicit bias of the mentioned iterative training algorithms with other loss functions such as polynomial losses would yield faster rates.

## 7 Conclusion

In our main results, Theorems 1 and 2, we present tight matching non-asymptotic upper and lower bounds for the prediction error of the maximum $\ell_1$-margin classifier, both in noiseless and noisy regimes. We thereby answer two open problems in the literature: perhaps surprisingly, as a first result (Theorem 1), we show that the classifier is not adaptive to sparsity in a standard (noiseless) discriminate data model. Furthermore, as a second result (Theorem 2), we show that the prediction error vanishes at a logarithmic rate despite interpolating a constant fraction of (randomly) corrupted labels, and thus that the classifier attains benign overfitting.

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

## A   Preliminary technical tools

The purpose of this section is to cite existing technical tools and simple corollaries of these results. In subsection A.1 we give some properties of the parametric path $\gamma(\alpha)$ introduced in Wang et al. (2022), which we used for reparameterization of optimization problems in preliminary step 2 in Section 4.2. Afterwards, in subsection A.2 we recall some concentration results, which we make use of when proving the localization and uniform convergence propositions (see section B and C) of Theorems 1 and 2.

### A.1  A few helpful properties of $\gamma(\alpha)$

First, recall from Section 4.1 that $h_2 \in \mathbb{R}^{d-s}$ contains samples of i.i.d. standard Gaussian random variables, and for the sake of brevity of notation, we define $h := |h_2|$. Moreover, recall the definition of the function $\gamma(\alpha) : \mathbb{R} \to \mathbb{R}^{d-s}$ from Equation (8):

$$\gamma(\alpha) = \arg\min_{w} \|w\|_2^2 \quad \text{s.t} \quad \begin{cases} \langle w, h \rangle \geq \|h\|_\infty \\ w \geq 0 \\ 1^\top w = \|w\|_1 = \alpha \end{cases}$$

for some scalar variables $b \geq 0$, $\alpha \in [1, (d-s)\frac{\|h\|_\infty}{\|h\|_1}]$. Without loss of generality, we can assume that $h_i > h_j$ for all $i > j$ (see also Wang et al. (2022)). Furthermore, the results of the main theorems do not change by considering $\gamma(\alpha) : \mathbb{R} \to \mathbb{R}^d$ since by our assumptions on the sparsity $s$, we have $d - s \approx d$. Therefore, in all discussion that follows, we will assume that $\gamma(\alpha) : \mathbb{R} \to \mathbb{R}^d$.

In order to study the optimization problem in Proposition 2, we make use of the following three properties of the path $\gamma(\alpha)$:

**Concentration of $\|\gamma(\alpha)\|_1$ and $\|\gamma(\alpha)\|_2$.**  As proven in Section 3.4 in Wang et al. (2022) the path $\gamma(\alpha)$ is a piecewise linear with breakpoints at $\alpha_m$ for integers $m = 2, \ldots, d$, with

$$\alpha_m = \frac{(\|h_{[m]}\|_1 - m h_m) \|h\|_\infty}{\|h_{[m]}\|_2^2 - \|h_{[m]}\|_1 h_m}$$

where $h_{[m]} \in \mathbb{R}^d$ denotes vector which is equal to $h \in \mathbb{R}^d$ on first $m$ components and zero elsewhere. Furthermore, the following concentration result holds as shown in Proposition 4 in Wang et al. (2022).

**Proposition 3.** *Let $t_m$ be given by $2\Phi^{\complement}(t_m) = m/d$. There exist universal positive constants $c_1, c_2, c_3, c_4 > 0$ such that for any $m, d$ with $m \geq c_1$ and $c_2 m \leq d \leq \exp(c_3 m^{1/5})$ we have that:*

$$\left| \frac{\|\gamma(\alpha_m)\|_1}{\|h\|_\infty} - \left( \frac{1}{t_m} - \frac{2}{t_m^3} \right) \right| \leq \frac{c_4}{t_m^5} \qquad \text{and} \qquad \left| \frac{\|\gamma(\alpha_m)\|_2^2}{\|h\|_\infty^2} - \frac{2}{m t_m^2} \right| \leq \frac{c_4}{m t_m^4},$$

*with probability at least $1 - 6\exp\left( -\frac{2m}{\log^5(d/m)} \right)$ over the draws of $h$.*

**Convexity and monotonicity of $\gamma(\alpha)$.**  According to Lemma 4 in Wang et al. (2022) the mapping $\alpha \mapsto \|\gamma(\alpha)\|_2^2$ is convex over $[1, \alpha_{\max}]$, decreasing over $[1, \alpha_{d+1/2}]$ and increasing over $[\alpha_{d+1/2}, \alpha_{\max}]$ where $\alpha_{d+1/2} := \frac{\|h\|_1 \|h\|_\infty}{\|h\|_2^2}$ satisfies $\alpha_d < \alpha_{d+1/2} < \alpha_{d+1}$. Furthermore the map $\alpha \mapsto \frac{\|\gamma(\alpha)\|_2^2}{\|\gamma(\alpha)\|_1^2} = \frac{\|\gamma(\alpha)\|_2^2}{\alpha^2}$ is monotonically decreasing.

**Inequality constraint at optimal point.**  According to Claim 3 in Wang et al. (2022) the inequality constraint in the definition of $\gamma(\alpha)$ is tight for the optimal solution, i.e., $\langle \gamma(\alpha), h \rangle = \|h\|_\infty$.

Furthermore we define $t_m$ as solution to equation

$$2\Phi^{\complement}(t_m) = m/d \tag{13}$$

for some integer $m \in [2, d]$ where $\Phi^{\complement}(.) = \mathbb{P}(Z \geq .)$ with $Z \sim \mathcal{N}(0, 1)$ is the complementary cumulative distribution function. We use the following two characterizations of $t_m$:

**Approximation of $t_m$.**  From Remark 2 in Wang et al. (2022) there exists universal constant $\kappa$ such that, for all $m \leq d/\kappa$ it holds that

$$t_m^2 = 2\log(d/m) - \log\log(d/m) - \log(\pi) + \frac{\log\log(d/m)}{2\log(d/m)} + O\left( \frac{1}{\log(d/m)} \right).$$

**Upper and lower bounds of** $t_m$**.** Following the same argument as in Claim 7 and Claim 9 in Wang et al. (2022), we can prove the following lemma:

**Lemma 2.** *Let $m_*$ be fixed and assume $\kappa_3 m_* \leq d$. Let any fixed constant $\kappa > 0$ and assume that parameter $\lambda$ satisfies $0 < \lambda \leq (\log(\kappa_3))^{\kappa/2}$, and let $\underline{m}_*$ be the largest integer $\widehat{m}$ such that $t^2_{\widehat{m}} \geq t^2_{m_*} + \frac{\lambda}{t^\kappa_{m_*}}$. Then,*

$$\underline{m}_* = m_* \exp\left(-\frac{\lambda}{2t^\kappa_{m_*}}\right)\left(1 + O\left(\frac{1}{t^2_{m_*}}\right)\right) \quad \text{and} \quad \left|t^2_{\underline{m}^*} - \left(t^2_{m_*} + \frac{\lambda}{t^\kappa_{m_*}}\right)\right| \leq O\left(\frac{1}{m_*}\right).$$

*Moreover, let $\overline{m}_*$ be the smallest integer $\widehat{m}$ such that $t^2_{\widehat{m}} \leq t^2_{m_*} - \frac{\lambda}{t^\kappa_{m_*}}$. Then,*

$$\overline{m}_* = m_* \exp\left(\frac{\lambda}{2t^\kappa_{m_*}}\right)\left(1 + O\left(\frac{1}{t^2_{m_*}}\right)\right) \quad \text{and} \quad \left|t^2_{\overline{m}^*} - \left(t^2_{m_*} - \frac{\lambda}{t^\kappa_{m_*}}\right)\right| \leq O\left(\frac{1}{m_*}\right).$$

Furthermore, analogously as in proof of Claim 8 in Wang et al. (2022) we get:

$$\frac{t^2_{m_*}}{t^2_{\underline{m}^*}} = \frac{1}{1 + \frac{\lambda}{t^{2+\kappa}_{m_*}} + O\left(\frac{1}{t^2_{m_*}m_*}\right)} = 1 - \frac{\lambda}{t^{2+\kappa}_{m_*}} + O\left(\frac{1}{t^2_{m_*}m_*}\right) + O\left(\frac{\lambda^2}{t^{4+2\kappa}_{m_*}}\right).$$

A similar result holds for $t_{\overline{m}}$.

## A.2   Concentration results

### Pointwise convergence

Lemmas in this section are used in the proofs of Propositions 4 and  6 (localization step). We recall two standard lemmas for pointwise convergence of functions of random variables to their expectation:

**Lemma 3** (Concentration of Lipschitz functions, Ledoux (1992); Wainwright (2019))**.** *Let $X = (X_1, \ldots, X_n)$ be a vector of i.i.d. $\mathcal{N}(0,1)$ random variables and let $f : \mathbb{R}^n \to \mathbb{R}$ be Lipschitz continuous with Lipschitz constant $L$. Then*

$$\mathbb{P}\left(|f(X) - \mathbb{E}f(X)| \geq \epsilon\right) \leq 2\exp\left(-\frac{\epsilon^2}{2L^2}\right)$$

*for any $\epsilon \geq 0$.*

**Lemma 4** (Bernstein's inequality for sub-exponentials, Vershynin (2018))**.** *Let $X_1, \ldots, X_n$ be mean zero i.i.d. random variables with sub-exponential norm $\kappa = \|X\|_{\psi_1}$. Then for any $\epsilon \geq 0$*

$$\mathbb{P}\left(\left|\frac{1}{n}\sum_{i=1}^n X_i\right| \geq \epsilon\right) \leq 2\exp\left(-cn\min\left\{\frac{\epsilon}{\kappa}, \frac{\epsilon^2}{\kappa^2}\right\}\right)$$

*for some universal constant $c > 0$.*

### Uniform convergence

Results from this section are used in the proofs of Propositions 5 and  7 (uniform convergence), and more specifically, for proving Propositions 9 and  10.

Let $X_1, \ldots, X_n$ be real i.i.d. random variables with continuous distribution function $F$ and let $F_n$ be the empirical distribution function defined by $F_n(x) = \frac{1}{n}\sum_{i=1}^n 1\{X_i \leq x\}$. Then we have:

**Lemma 5** (Dvoretzky-Kiefer-Wolfowitz inequality, Dvoretzky et al. (1956); Massart (1990))**.** *For any $\epsilon > 0$ holds:*

$$\mathbb{P}\left(\sup_x |F_n(x) - F(x)| > \frac{\epsilon}{\sqrt{n}}\right) \leq 2\exp(-2\epsilon^2)$$

Before we recall a result about uniform convergence of functions from a parametrized set, let us introduce an additional notation. Let $\mathcal{G}$ be a countable class of functions $g : \mathbb{R} \to \mathbb{R}$. For a function $g \in \mathcal{G}$ we write $Pg = \mathbb{E}g(X)$, and $P_n g = \frac{1}{n} \sum_{i=1}^{n} g(x_i)$. Moreover, define $\|P_n - P\|_{\mathcal{G}} := \sup_{g \in \mathcal{G}} |(P_n - P)g|$.

Let $\epsilon_1, \ldots, \epsilon_n$ be independent Rademacher random variables. Define $P_n^{\epsilon} g = \frac{1}{n} \sum_{i=1}^{n} \epsilon_i g(x_i)$ and $\|P_n^{\epsilon}\|_{\mathcal{G}} = \sup_{g \in \mathcal{G}} |P_n^{\epsilon} g|$. We also recall the definition of the Orlicz norm $\|\cdot\|_{\Psi_{\alpha}}$. Let $\alpha > 0$ and define the Orlicz function $\psi_{\alpha} : \mathbb{R}_+ \to \mathbb{R}_+$ by $\psi_{\alpha}(x) = \exp(x^{\alpha}) - 1$. The Orlicz norm of the random variable $X$ is given by:

$$\|X\|_{\Psi_{\alpha}} := \inf\{\lambda > 0 : \mathbb{E}\psi_{\alpha}(|X|/\lambda) \leq 1\}$$

For the setting defined in this section we have:

**Theorem 3** (Corollary of Theorem 4 in Adamczak (2008))**.** *For any $0 < t < 1$, $\delta > 0$, $\alpha \in (0,1]$ there exists a constant $C = C(\alpha, t, \delta)$ such that*

$$\mathbb{P}\left(\|P_n - P\|_{\mathcal{G}} \geq (1+t)\mathbb{E}\|P_n - P\|_{\mathcal{G}} + \epsilon\right) \leq \exp\left(-\frac{n\epsilon^2}{2(1+\delta)\sigma_{\mathcal{G}}^2}\right) + 3\exp\left(-\left(\frac{\epsilon}{C\psi_{\mathcal{G}}}\right)^{\alpha}\right)$$

*with*

$$\sigma_{\mathcal{G}}^2 = \sup_{g \in \mathcal{G}} \mathrm{Var}[g(X)] \qquad \text{and} \qquad \psi_{\mathcal{G}} = \left\|\max_{1 \leq i \leq n} \sup_{g \in \mathcal{G}} \frac{1}{n}\left|g(x_i) - \mathbb{E}_X[g(X)]\right|\right\|_{\Psi_{\alpha}}$$

## B   Proof of Theorem 1

In this section, we present the proof of Theorem 1. By Proposition 1, in order to give bounds for prediction error, it suffices to bound $\phi_N, \phi_+$ and $\phi_-$ (defined in Section 4.1). Furthermore, we make use of the simplifications in Proposition 2, which allow us to study low-dimensional stochastic optimization problems. In a first step (localization), we derive an upper bound for $\phi_N$:

**Proposition 4.** *Let the assumptions of Theorem 1 hold, and let $\kappa_M = 3(72\pi)^{-1/6}$ and $m_n$ be the solution of equation*

$$m_n = \sqrt{\frac{2}{\pi}}(72\pi)^{1/6}(nt_{m_n}\|w^*\|_1)^{2/3}, \tag{14}$$

*where $t_{m_n}$ is defined as in Equation (13) in Appendix A.1. There exists universal positive constants $c_1, c_2, c_3$ such that*

$$\phi_N \leq \kappa_M\left(\frac{n}{t_{m_n}^2}\|w^*\|_1\right)^{1/3}\left(1 - \frac{2}{3}\frac{1}{t_{m_n}^2} + \frac{c_1}{t_{m_n}^4}\right) =: M$$

*holds with probability at least $1 - c_2 \exp\left(-c_3 \frac{n^{1/3}}{\log^{10/3}(d/m_n)}\right)$ over the draws of $h_1, h_2, z^{(1)}, z^{(2)}$.*

The proof of the proposition is deferred to Appendix B.1. The second step (uniform convergence) gives the following bounds on the elements of the set $\Gamma$ from Proposition 2:

**Proposition 5.** *Let the assumptions of Theorem 1 hold. Let $\Gamma_0$ be a set of all $(\nu, b, \alpha, \eta_{\mathcal{S}})$ that satisfy:*

$$\left|\nu^2 - \frac{(288\pi)^{-1/3}n^{2/3}}{\|w^*\|_1^{4/3}\log^{2/3}(d/m_n)}\right| \lesssim \frac{n^{2/3}}{\|w^*\|_1^{4/3}\log(d/m_n)} \quad \text{and} \quad \eta_{\mathcal{S}}^2 \lesssim \frac{1}{\log^{7/6}(d/m_n)}$$

$$\text{and} \quad \left|b^2\|\gamma(\alpha)\|_2^2 - \frac{1}{3}\frac{1}{\log(d/m_n)}\right| \lesssim \frac{1}{\log^{7/6}(d/m_n)}$$

*where $m_n$ is the solution of Equation (14). Then there exist positive universal constants $c_1, c_2, c_3$ such that $\Gamma \subset \Gamma_0$ with probability at least $1 - c_1 d^{-1} - c_2 \exp\left(-c_3 \frac{n^{1/3}}{\log^4(d/m_n)}\right)$ over the draws of $h_1, h_2, z^{(1)}, z^{(2)}$.*

The proof is deferred to Appendix B.2. From the Propositions 2 and 5 and using that $\eta_{\mathcal{S}}^2 \geq 0$, we get the following bounds on $\phi_+$ and $\phi_-$:

$$\phi_+ \leq \left[ 1 + \frac{\min\limits_{(b,\alpha)\in\Gamma_0} b^2 \left\| \gamma(\alpha) \right\|_2^2 + \min\limits_{\eta_{\mathcal{S}}\in\Gamma_0} \eta_{\mathcal{S}}^2}{\max\limits_{\nu\in\Gamma_0} \nu^2} \right]^{-1/2} \leq 1 - \frac{4 \left\| w^* \right\|_1^2}{m_n} \left( 1 - \frac{c}{\log^{1/6}(d/m_n)} \right)$$

$$\phi_- \geq \left[ 1 + \frac{\max\limits_{(b,\alpha)\in\Gamma_0} b^2 \left\| \gamma(\alpha) \right\|_2^2 + \max\limits_{\eta_{\mathcal{S}}\in\Gamma_0} \eta_{\mathcal{S}}^2}{\min\limits_{\nu\in\Gamma_0} \nu^2} \right]^{-1/2} \geq 1 - \frac{4 \left\| w^* \right\|_1^2}{m_n} \left( 1 + \frac{c}{\log^{1/6}(d/m_n)} \right),$$

and the statement of Theorem 1 follows straightforwardly when applying Proposition 1 and using that

$$\mathrm{R}(\hat{w}) = \frac{1}{\pi} \arccos \left( \left\langle \frac{\hat{w}}{\left\| \hat{w} \right\|_2}, w^* \right\rangle \right) = \frac{1}{\pi} \sqrt{2 \left( 1 - \left\langle \frac{\hat{w}}{\left\| \hat{w} \right\|_2}, w^* \right\rangle \right)} + O \left( 1 - \left\langle \frac{\hat{w}}{\left\| \hat{w} \right\|_2}, w^* \right\rangle \right)^{3/2}. \quad (15)$$

### B.1 Proof of Localization Proposition 4

Recall the upper bound of $\phi_N$ from Proposition 2, and note that to upper bound $\phi_N$ it is sufficient to find a feasible point $(\tilde{\nu}, \tilde{b}, \tilde{\alpha})$ which satisfies the constraint, i.e. we have:

$$\phi_N \leq \tilde{\nu} \left\| w^* \right\|_1 + \tilde{b}\tilde{\alpha} \quad \text{if} \quad \frac{1}{n}\tilde{b}^2 \left\| h \right\|_\infty^2 \geq f_n(\tilde{\nu}, \tilde{b}\|\gamma(\tilde{\alpha})\|_2) \quad (16)$$

holds with high probability for some $\tilde{\nu} > 0$. We further recall that in the noiseless setting we have

$$f(\nu, \eta) = \mathbb{E}f_n(\nu, \eta) = \mathbb{E} \left( 1 - \nu|Z^{(1)}| - Z^{(2)}\eta \right)_+^2.$$

with $f_n$ from Equation (5). Next, note that the random variable $(1 - \nu|Z^{(1)}| - \eta Z^{(2)})_+^2$ for fixed $(\nu, \eta)$ is a sub-exponential random variable. Furthermore, since $(1 - \nu|Z^{(1)}| - \eta Z^{(2)})_+^2 \lesssim 1 + \eta^2 (Z^{(2)})^2$ we see that the subexponential norm of this random variable is bounded by a constant for $\eta \leq c$. We can therefore apply Lemma 4 to show that for fixed $\nu$, $\eta \leq c$ and $m_n$ given in Equation (14) we have

$$\mathbb{P} \left( |f_n(\nu, \eta) - \mathbb{E}f_n(\nu, \eta)| \lesssim \frac{1}{\nu t_{m_n}^4} \right) \geq 1 - 2\exp \left( -c_1 \frac{n}{\nu^2 t_{m_n}^8} \right).$$

Since $f$ is an infinitely differentiable function, we can use the Taylor expansion of the function $f = \mathbb{E}f_n$ around $\eta = 0$ from Equation (56) which holds for $\nu$ large. Combining the last two results we obtain that with probability $1 - 2\exp \left( -c_1 \frac{n}{\nu^2 t_{m_n}^8} \right)$ holds:

$$f_n(\nu, b\|\gamma(\alpha)\|_2) \leq \frac{\sqrt{2}}{3\sqrt{\pi}} \frac{1}{\nu} + \sqrt{\frac{2}{\pi}} \frac{b^2 \left\| \gamma(\alpha) \right\|_2^2}{\nu} + \mathcal{O}_t + \mathcal{O}_c \quad (17)$$

where $\mathcal{O}_t := O \left( \frac{1}{\nu^3}, \frac{b^4 \|\gamma(\alpha)\|_2^4}{\nu^3} \right)$ and $\mathcal{O}_c := O \left( \frac{1}{\nu t_{m_n}^4} \right)$.

We claim that for our choice of point $(\tilde{\nu}, \tilde{b}, \tilde{\alpha})$ we get $\mathcal{O}_t + \mathcal{O}_c = \frac{1}{\tilde{\nu}}O \left( \frac{1}{t_{m_n}^4} \right)$. Once we have established inequality (17), the claim that the point $(\tilde{\nu}, \tilde{b}, \tilde{\alpha})$ satisfies constraint from (16) is implied by proving the following inequality:

$$\frac{1}{n}\tilde{b}^2 \left\| h \right\|_\infty^2 \geq \frac{\sqrt{2}}{3\sqrt{\pi}} \frac{1}{\tilde{\nu}} + \sqrt{\frac{2}{\pi}} \frac{\tilde{b}^2 \left\| \gamma(\tilde{\alpha}) \right\|_2^2}{\tilde{\nu}} + \frac{1}{\tilde{\nu}}O \left( \frac{1}{t_{m_n}^4} \right) \quad (18)$$

Defining $\widetilde{b_\alpha} = \tilde{b}\tilde{\alpha}$ and rearranging the terms in Equation (18) we obtain the following lower bound for $\widetilde{b_\alpha}$:

$$\widetilde{b_\alpha}^2 \geq \frac{n\tilde{\alpha}^2}{\|h\|_\infty^2} \frac{\frac{\sqrt{2}}{3\sqrt{\pi}}\frac{1}{\tilde{\nu}}\left(1 + O\left(\frac{1}{t_{m_n}^4}\right)\right)}{1 - \sqrt{\frac{2}{\pi}\frac{n}{\tilde{\nu}}\frac{\|\gamma(\tilde{\alpha})\|_2^2}{\|H\|_\infty^2}}}$$

From Section A.1 we have that $\gamma(\alpha)$ is a piecewise linear function with breakpoints at $\alpha_m$ for $m = 2, \ldots, d$, and thus, we can optimize over integers $m$ instead of $\alpha$. Using concentration results from Proposition 3 we get the following result:

$$\widetilde{b_\alpha}^2 \geq n\frac{1}{t_m^2}\left(1 - \frac{4}{t_m^2} + O\left(\frac{1}{t_m^4}\right)\right)\frac{\frac{\sqrt{2}}{3\sqrt{\pi}}\frac{1}{\tilde{\nu}}\left(1 + O\left(\frac{1}{t_{m_n}^4}\right)\right)}{1 - \sqrt{\frac{2}{\pi}\frac{n}{\tilde{\nu}}\frac{2}{mt_m^2}}\left(1 + O\left(\frac{1}{t_m^2}\right)\right)} \tag{19}$$

with probability at least $1 - 6\exp\left(-\frac{2m}{\log^5(d/m)}\right)$. Similarly, as in Remark 1 in Wang et al. (2022), we choose $m$, which approximately minimizes the expression above, i.e. to maximize:

$$t_m^2\left(1 - \sqrt{\frac{2}{\pi}\frac{n}{\tilde{\nu}}\frac{2}{mt_m^2}}\right) \approx 2\log\left(\frac{d}{m}\right) - 2\sqrt{\frac{2}{\pi}\frac{n}{\tilde{\nu}m}}$$

This gives $m = m_n(\tilde{\nu}) := \sqrt{\frac{2}{\pi}\frac{n}{\tilde{\nu}}}$. We claim that for our choice of $\tilde{\nu}$ we can set $m_n$ as the solution of equation $m_n = \sqrt{\frac{2}{\pi}}(72\pi)^{1/6}(nt_{m_n}\|w^*\|_1)^{2/3}$ which is exactly $m_n$ given in Equation (14). For such $m = m_n$ we have from Equation (19):

$$\widetilde{b_\alpha}^2 \geq \frac{\sqrt{2}}{3\sqrt{\pi}}\frac{n}{\tilde{\nu}t_{m_n}^2}\left(1 - \frac{2}{t_{m_n}^2} + O\left(\frac{1}{t_{m_n}^4}\right)\right)$$

So we let:

$$\widetilde{b_\alpha}(\tilde{\nu}) := \sqrt{\frac{\sqrt{2}}{3\sqrt{\pi}}\frac{n}{\tilde{\nu}t_{m_n}^2}\left(1 - \frac{2}{t_{m_n}^2} + O\left(\frac{1}{t_{m_n}^4}\right)\right)}$$

Now we choose $\tilde{\nu}$ which minimizes the upper bound on $\phi_N$ in Equation (16) as follows:

$$\tilde{\nu} := \underset{\nu>0}{\arg\min}\, \nu\|w^*\|_1 + \widetilde{b_\alpha}(\nu) = \underset{\nu>0}{\arg\min}\, \nu\|w^*\|_1 + \sqrt{\frac{\sqrt{2}}{3\sqrt{\pi}}\frac{n}{\nu t_{m_n}^2}\left(1 - \frac{2}{t_{m_n}^2} + O\left(\frac{1}{t_{m_n}^4}\right)\right)}$$

After minimization, we get that $\tilde{\nu}$ is given by:

$$\tilde{\nu} = (72\pi)^{-1/6}\|w^*\|_1^{-2/3}\left(\frac{n}{t_{m_n}^2}\right)^{1/3}\left(1 - \frac{2}{t_{m_n}^2} + O\left(\frac{1}{t_{m_n}^4}\right)\right) > 0$$

Note that indeed $m_n(\tilde{\nu}) = m_n$ for this choice of $\tilde{\nu}$. Returning to $\widetilde{b_\alpha}$, we obtain the following:

$$\widetilde{b_\alpha} := \widetilde{b_\alpha}(\tilde{\nu}) = 2(72\pi)^{-1/6}\|w^*\|_1^{1/3}\left(\frac{n}{t_{m_n}^2}\right)^{1/3}\left(1 + O\left(\frac{1}{t_{m_n}^4}\right)\right)$$

Summing up the two terms, we obtain a bound from the proposition. Also, note that for $m = m_n$ we get:

$$\tilde{b}\|\gamma(\tilde{\alpha})\|_2 = \widetilde{b_\alpha}\frac{\|\gamma(\tilde{\alpha})\|_2}{\tilde{\alpha}} = \sqrt{\frac{2}{3}}\frac{1}{t_{m_n}}\left(1 + O\left(\frac{1}{t_{m_n}^2}\right)\right)$$

So, we have $O_t = O\left(\frac{1}{\nu^3}, \frac{\eta^4}{\nu^3}\right) = o\left(\frac{1}{\nu t_{m_n}^4}\right)$ as we assumed at the beginning of the proof. Thus, the point $(\tilde{\nu}, \tilde{b}, \tilde{\alpha})$ indeed satisfies the inequality (16) with high probability, and we define the upper bound $M := \tilde{\nu} + \tilde{b}\tilde{\alpha} \geq \phi_N$.

### B.2 Proof of Uniform Convergence Proposition 5

For the sake of completeness, let us recall the definition of set $\Gamma$ from Proposition 2:

$$\Gamma = \Big\{(\nu, b, \alpha, \eta_{\mathcal{S}}) \ \ \text{s.t} \ \ \eta_{\mathcal{S}} \geq 0, b \geq 0, \alpha \in [1, \alpha_{\max}]$$

$$\text{and} \ \ \frac{1}{n}(2\sqrt{s_{\max}}\eta_{\mathcal{S}} + b\|h\|_\infty)^2 \geq f_n(\nu, \sqrt{b^2\|\gamma(\alpha)\|_2^2 + \eta_{\mathcal{S}}^2})$$

$$\text{and} \ \ \max\big\{|\nu|\|w^*\|_1 - \sqrt{s}\eta_{\mathcal{S}}, 0\big\} + b\|\gamma(\alpha)\|_1 \leq M\Big\}$$

with $M$ given in Proposition 4 and $s_{\max} = \Theta(n^{2/3}\log^{-14/3} d)$. We further recall the notation $\eta_{\mathcal{S}^c} = b\|\gamma(\alpha)\|_2$ and $\eta = \sqrt{\eta_{\mathcal{S}^c}^2 + \eta_{\mathcal{S}}^2}$ in Section 4.2.

The proof consists of three steps where we iteratively bound the set $\Gamma$: for every step, we use different approximations of $f_n$, and based on them, we develop tighter bounds for $\nu, \eta_{\mathcal{S}^c}, \eta_{\mathcal{S}}$. Finally, the statement of the proposition follows from the last, tightest bound. We start with the following bound:

**Bound 1:** $\nu\|w^*\|_1 \lesssim M, \nu \gtrsim \frac{n^{1/3}}{s_{\max}^{1/3}\log d}$

In order to derive the bounds in this section, we first need to simplify the constraints from the definition of the set $\Gamma$. First, note that we can relax the second constraint to the following two constraints: $b\alpha \leq M$ and $\nu\|w^*\|_1 \leq M + \sqrt{s}\eta_{\mathcal{S}}$. Then, the first constraint is simplified by deriving an upper bound on the term from the LHS as follows. By using simple quadratic inequality, we have that for any $(\nu, b, \alpha, \eta_{\mathcal{S}}) \in \Gamma$ it holds that:

$$\frac{1}{n}(2\sqrt{s_{\max}}\eta_{\mathcal{S}} + b\|H\|_\infty)^2 \leq \frac{2}{n}b^2\|h\|_\infty^2 + \frac{8}{n}s_{\max}\eta_{\mathcal{S}}^2 \tag{20}$$

Now, recall that $t_{m_n}^2 \gtrsim \log(d/m_n) \geq \log\kappa_2$ and $\alpha \geq 1$, both from Section A.1. We can further bound the first term from Equation (20) with probability $\geq 1 - \frac{1}{d}$ as follows:

$$\frac{2}{n}b^2\|h\|_\infty^2 \leq \max_{(\nu,b,\alpha,\eta_{\mathcal{S}})\in\Gamma}\frac{2}{n}b^2\|h\|_\infty^2 \leq \max_{\alpha\in\Gamma}\frac{2}{n}\frac{M^2}{\alpha^2}\|h\|_\infty^2 = \frac{2}{n}M^2\|h\|_\infty^2$$

$$\lesssim \frac{1}{n}\left(\frac{n}{t_{m_n}^2}\|w^*\|_1\right)^{2/3}\log d \lesssim \frac{\|w^*\|_1^{2/3}}{n^{1/3}}\log d,$$

where we used the concentration of the maximum of i.i.d. Gaussian random variables in the second line. We can now define the following (larger) set:

$$\Gamma_1 = \Big\{(\nu, b, \alpha, \eta_{\mathcal{S}}) \ \ \text{s.t} \ \ \eta_{\mathcal{S}}^2\frac{s_{\max}}{n} + \frac{\|w^*\|_1^{2/3}}{n^{1/3}}\log d \gtrsim f_n(\nu, \sqrt{b^2\|\gamma(\alpha)\|_2^2 + \eta_{\mathcal{S}}^2})$$

$$\text{and} \ \ b\alpha \leq M \ \ \text{and} \ \ \nu\|w^*\|_1 \leq M + \sqrt{s}\eta_{\mathcal{S}}\Big\}.$$

From our discussion above it follows that $\Gamma \subset \Gamma_1$ with high probability. The goal of this first step is to show that the bounds $\nu\|w^*\|_1 \lesssim M$ and $\nu \gtrsim \frac{n^{1/3}}{s_{\max}^{1/3}\log d}$ hold uniformly over all $\nu \in \Gamma_1$, implying that they also hold uniformly for all $\nu \in \Gamma$ with high probability.

**Step 1.1: Upper bound $\nu\|w^*\|_1 \lesssim M$.** In all of Step 1.1 we assume that $(\nu, b, \alpha, \eta_{\mathcal{S}}) \in \Gamma_1$ that is, we bound these variables only if they are contained in $\Gamma_1$. Since by the last constraint of $\Gamma_1$ it holds that $\nu\|w^*\|_1 \leq M + \sqrt{s}\eta_{\mathcal{S}}$, showing that $\sqrt{s}\eta_{\mathcal{S}} \leq cM$ for some universal constant $c > 0$ is sufficient to deduce that $\nu\|w^*\|_1 \leq (c+1)M$.

Assume by contradiction that $\sqrt{s}\eta_{\mathcal{S}} > cM$ for any constant $c > 0$. Then, we can relax the first constraint of $\Gamma_1$ as follows:

$$\eta_{\mathcal{S}}^2 \frac{s_{\max}}{n} + \frac{\|w^*\|_1^{2/3}}{n^{1/3}} \log d \gtrsim f_n(\nu, \sqrt{b^2 \|\gamma(\alpha)\|_2^2 + \eta_{\mathcal{S}}^2}) = \frac{1}{n}\sum_{i=1}^n (1 - \nu|z_i^{(0)}| - z_i^{(1)}\sqrt{b^2\|\gamma(\alpha)\|_2^2 + \eta_{\mathcal{S}}^2})_+^2$$

$$\geq \frac{1}{n}\sum_{i=1}^n (1 - \nu|z_i^{(0)}| - z_i^{(1)}\sqrt{b^2\|\gamma(\alpha)\|_2^2 + \eta_{\mathcal{S}}^2})_+^2 1\{z_i^{(2)} \leq -c_1\}$$

$$\geq \frac{1}{n}\sum_{i=1}^n (-\nu|z_i^{(0)}| + c_1\eta_{\mathcal{S}})_+^2 1\{z_i^{(2)} \leq -c_1\}$$

$$\geq \frac{1}{n}\sum_{i=1}^n \left(-\frac{M + \sqrt{s}\eta_{\mathcal{S}}}{\|w^*\|_1}|z_i^{(0)}| + c_1\eta_{\mathcal{S}}\right)_+^2 1\{z_i^{(2)} \leq -c_1\}$$

$$\geq \frac{1}{n}\sum_{i=1}^n \left(-(1+c^{-1})\frac{\sqrt{s}\eta_{\mathcal{S}}}{\|w^*\|_1}|z_i^{(0)}| + c_1\eta_{\mathcal{S}}\right)_+^2 1\left\{|z_i^{(1)}| \leq \frac{c_1\|w^*\|_1}{2(1+c^{-1})\sqrt{s}}\right\} 1\{z_i^{(2)} \leq -c_1\}$$

$$\gtrsim \eta_{\mathcal{S}}^2 \frac{1}{n}\sum_{i=1}^n 1\left\{|z_i^{(1)}| \leq \frac{c_1\|w^*\|_1}{2(1+c^{-1})\sqrt{s}}\right\} 1\{z_i^{(2)} \leq -c_1\}$$

where in the fourth line we used that $\nu \leq \frac{M+\sqrt{s}\eta_{\mathcal{S}}}{\|w^*\|_1}$, and in fifth that $M < c^{-1}\sqrt{s}\eta_{\mathcal{S}}$. Next, we use that

$$\mathbb{P}(|Z^{(1)}| \leq \frac{c_1\|w^*\|_1}{2(1+c^{-1})\sqrt{s}}) \gtrsim \frac{\|w^*\|_1}{\sqrt{s}} \gtrsim \frac{\|w^*\|_1}{\sqrt{s_{\max}}}$$

and $\mathbb{P}(Z^{(2)} \leq -c_1) \geq c_2$ and thus, from Lemma 5 with $\epsilon = c\sqrt{\frac{n}{s_{\max}}}$, we obtain the following inequality holds with probability $\geq 1 - \exp\left(-c_3 \frac{n}{s_{\max}}\right)$:

$$\eta_{\mathcal{S}}^2 \frac{s_{\max}}{n} + \frac{\|w^*\|_1^{2/3}}{n^{1/3}} \log d \geq c_4 \eta_{\mathcal{S}}^2 \frac{\|w^*\|_1}{\sqrt{s_{\max}}} \tag{21}$$

First note that $s_{\max} = o((n\|w^*\|_1)^{2/3})$ and thus $\eta_{\mathcal{S}}^2 \frac{s_{\max}}{n} < \frac{c_4}{2}\eta_{\mathcal{S}}^2 \frac{\|w^*\|_1}{\sqrt{s_{\max}}}$. Thus in order for inequality (21) to hold, we need that $\frac{\|w^*\|_1^{2/3}}{n^{1/3}} \log d \geq \frac{c_4}{2}\eta_{\mathcal{S}}^2 \frac{\|w^*\|_1}{\sqrt{s_{\max}}}$ or equivalently $\eta_{\mathcal{S}}^2 \lesssim \frac{\sqrt{s_{\max}} \log d}{n^{1/3}\|w^*\|_1^{1/3}} \leq \frac{\sqrt{s_{\max}}}{n^{1/3}} \log d$. But then $\sqrt{s}\eta_{\mathcal{S}} \lesssim s_{\max}^{3/4} n^{-1/6}\sqrt{\log d}$, which is in contradiction with our assumption that $\sqrt{s}\eta_{\mathcal{S}} > cM$, since $s_{\max}^{3/4} n^{-1/6}\sqrt{\log d} \lesssim \left(\frac{n}{\log d}\right)^{1/3} \lesssim M$ for $s_{\max} = \Theta(n^{2/3}\log^{-14/3} d)$.

Hence we conclude that $\sqrt{s}\eta_{\mathcal{S}} \leq cM$, and furthermore $\nu\|w^*\|_1 \leq \tilde{c}M$ for some universal constants $c, \tilde{c} > 0$, which is exactly what we wanted to show in this step.

**Step 1.2: Lower bound $\nu \gtrsim \frac{n^{1/3}}{s_{\max}^{1/3}\log d}$.**

In order to show this lower bound, we first lower bound the function $f_n$ for any $\nu, \eta$ as follows:

$$f_n(\nu, \eta) = \frac{1}{n}\sum_{i=1}^n (1 - \nu|z_i^{(1)}| - \eta z_i^{(2)})_+^2 \geq \frac{1}{n}\sum_{i=1}^n (1 - \nu|z_i^{(1)}| - \eta z_i^{(2)})_+^2 1\{z_i^{(2)} \leq 0\}$$

$$\gtrsim \frac{1}{n}\sum_{i=1}^n (1 - \nu|z_i^{(1)}|)_+^2$$

with probability $\geq 1 - \exp(-c_1 n)$ for some positive universal constant $c_1$. Combining this inequality with the first constraint of $\Gamma_1$ we have that any $(\nu, b, \alpha, \eta_\mathcal{S}) \in \Gamma_1$ must satisfy with high probability that:

$$\frac{1}{n}\sum_{i=1}^{n}(1 - \nu|z_i^{(1)}|)_+^2 \lesssim f_n(\nu, \sqrt{b^2\|\gamma(\alpha)\|_2^2 + \eta_\mathcal{S}^2}) \lesssim \eta_\mathcal{S}^2 \frac{s_{\max}}{n} + \frac{\|w^*\|_1^{2/3}}{n^{1/3}}\log d$$

$$\lesssim \max\left\{\frac{s_{\max}^{1/3}}{n^{1/3}}, \frac{s_{\max}^{1/3}}{n^{1/3}}\log d\right\} \lesssim \frac{s_{\max}^{1/3}}{n^{1/3}}\log d \tag{22}$$

where in the second line we used that $s_{\max}\eta_\mathcal{S}^2 \leq c^2 M^2 \lesssim (n\sqrt{s_{\max}})^{2/3}$ shown in the previous step, and that $\|w^*\|_1^{2/3} \leq (s_{\max})^{1/3}$.

Now, define $F_n(\frac{1}{2\nu}) := \frac{1}{n}\sum_{i=1}^{n}1\{|z_i^{(1)}| \leq \frac{1}{2\nu}\}$ and $F(\frac{1}{2\nu}) := \mathbb{P}(|Z^{(1)}| \leq \frac{1}{2\nu}) = \mathrm{erf}(\frac{1}{2\sqrt{2}\nu})$ by the definition of the error function. We can further simplify inequality (22) as follows:

$$\frac{s_{\max}^{1/3}}{n^{1/3}}\log d \gtrsim \frac{1}{n}\sum_{i=1}^{n}(1 - \nu|z_i^{(1)}|)_+^2 \geq \frac{1}{n}\sum_{i=1}^{n}(1 - \nu|z_i^{(1)}|)^2 1\{1 - 2\nu|z_i^{(1)}|\}$$

$$\geq \frac{nF_n(\frac{1}{2\nu})}{n}\left(\frac{1}{2}\right)^2 \gtrsim F_n\left(\frac{1}{2\nu}\right)$$

where we used that the number of activated indicators of the set $\{1\{1 - 2\nu|z_i^{(1)}|\}\}_{i=1}^{n}$ is equal to $nF_n(\frac{1}{2\nu})$ and that $(1 - \nu|z_i^{(1)}|)_+ \geq \frac{1}{2}$ when $1 - 2\nu|z_i^{(1)}| \geq 0$. Then, according to the Dvoretzky-Kiefer-Wolfowitz inequality from Lemma 5 we have with probability at least $1 - 2\exp(-cn^{1/3}s_{\max}^{2/3}\log^2 d)$ that

$$\sup_\nu\left|F_n\left(\frac{1}{2\nu}\right) - F\left(\frac{1}{2\nu}\right)\right| = \sup_\nu\left|F_n\left(\frac{1}{2\nu}\right) - \mathrm{erf}\left(\frac{1}{2\sqrt{2}\nu}\right)\right| \lesssim \frac{s_{\max}^{1/3}}{n^{1/3}}\log d$$

Thus we can use the Taylor series approximation of $\mathrm{erf}(\cdot)$ around zero to show that $\nu \gtrsim \frac{n^{1/3}}{s_{\max}^{1/3}\log d}$, as we wanted to show.

**Bound 2:** $\eta_{\mathcal{S}^c}, \eta_\mathcal{S} = O(1)$, $\nu\|w^*\|_1 \geq \underline{\kappa}M$

For this bound we use results from the previous steps. Restricting to the set where $\nu \gtrsim \frac{n^{1/3}}{s_{\max}^{1/3}\log d}$, and $\nu \lesssim \frac{M}{\|w^*\|_1} \lesssim n^{1/3}$, we can use the lower bound from Proposition 8:

$$f_n(\nu, \eta) \geq \kappa_1\frac{1}{\nu} + \kappa_2\frac{\eta^2}{\nu} \tag{23}$$

which holds with probability $\geq 1 - 2\exp(-c_2 n^{1/3})$.

Now we further simplify the LHS of the first constraint in the definition of set $\Gamma_1$. Combining the upper bound from Equation (20) with the lower bound (23), we have

$$\frac{2}{n}b^2\|h\|_\infty^2 + \frac{8}{n}s_{\max}\eta_\mathcal{S}^2 \geq f_n(\nu, \eta) \geq \kappa_1\frac{1}{\nu} + \kappa_2\frac{b^2\|\gamma(\alpha)\|_2^2 + \eta_\mathcal{S}^2}{\nu} \tag{24}$$

As before, we have $\frac{s_{\max}}{n} = \Theta(\frac{1}{n^{1/3}\log^{14/3}d})$ from the definition of $s_{\max}$, and, as noted above, we have that $\frac{1}{\nu} \gtrsim \frac{1}{n^{1/3}}$. Thus, for $n \geq c$ we have that $\frac{8s_{\max}}{n} \leq \frac{\kappa_2}{2\nu}$, and hence:

$$\frac{2}{n}b^2\|h\|_\infty^2 \geq \kappa_1\frac{1}{\nu} + \kappa_2\frac{b^2\|\gamma(\alpha)\|_2^2}{\nu} + \eta_\mathcal{S}^2\left(\frac{\kappa_2}{\nu} - \frac{8s_{\max}}{n}\right) \geq \kappa_1\frac{1}{\nu} + \kappa_2\frac{b^2\|\gamma(\alpha)\|_2^2}{\nu} + \kappa_3\frac{\eta_\mathcal{S}^2}{\nu}$$

where we set $\kappa_3 = \frac{\kappa_2}{2}$. Since the above inequalities hold with high probability, we define a set $\Gamma_2$ as the set of all $(\nu, b, \alpha, \eta_{\mathcal{S}})$ that satisfy:

$$\frac{2}{n} b^2 \|h\|_\infty^2 \geq \kappa_1 \frac{1}{\nu} + \kappa_2 \frac{b^2 \|\gamma(\alpha)\|_2^2}{\nu} + \kappa_3 \frac{\eta_{\mathcal{S}}^2}{\nu} \quad \text{and} \quad b\alpha \leq M \quad \text{and} \quad \frac{n^{1/3}}{s_{\max}^{1/3} \log d} \lesssim \nu \lesssim \frac{M}{\|w^*\|_1}$$

and by the above discussion we have that with high probability, $\Gamma \subset \Gamma_2$. Hence, by bounding the variables $\nu, \eta$ from $\Gamma_2$, we will also obtain valid upper bounds in $\Gamma$ as well.

**Step 2.1: Upper bound $\eta_{\mathcal{S}^c} = O(1)$.** Recall that we use the parameterization of $\eta_{\mathcal{S}^c}$ such that $\eta_{\mathcal{S}^c} = b \|\gamma(\alpha)\|_2$. Thus, we bound $\eta_{\mathcal{S}^c}$ as follows:

$$\eta_{\mathcal{S}^c}^2 \leq \max_{(\nu, b, \alpha, \eta_{\mathcal{S}}) \in \Gamma_2} b^2 \|\gamma(\alpha)\|_2^2$$

$$\leq \max_{\nu, b, \alpha} \left[ b^2 \|\gamma(\alpha)\|_2^2 \quad \text{s.t} \quad \|\gamma(\alpha)\|_2^2 \leq \frac{2}{\kappa_2 n} \nu \|h\|_\infty^2 \quad \text{and} \quad b\alpha \leq M \quad \text{and} \quad \nu \|w^*\|_1 \leq cM \right]$$

$$= \max_\alpha \left[ M^2 \frac{\|\gamma(\alpha)\|_2^2}{\alpha^2} \quad \text{s.t} \quad \|\gamma(\alpha)\|_2^2 \leq \frac{2c}{\kappa_2 n} \frac{M}{\|w^*\|_1} \|h\|_\infty^2 \right]. \tag{25}$$

As we mentioned in Section A.1, the function $\frac{\|\gamma(\alpha)\|_2^2}{\alpha^2}$ is a monotonically decreasing function in $\alpha$, while $\|\gamma(\alpha)\|_2^2$ is a convex function. Thus, similarly to the proofs in Wang et al. (2022), it is sufficient to find $\alpha_{\underline{m}} < \alpha_{m_n}$, such that

$$\frac{\|\gamma(\alpha_{\underline{m}})\|_2^2}{\|h\|_\infty^2} > \frac{2c}{\kappa_2 n} \frac{M}{\|w^*\|_1}$$

to obtain an upper bound on $\frac{\|\gamma(\alpha)\|_2^2}{\alpha^2}$ (where we implicitly make use of the fact that the set $\Gamma$ contains the point $(\tilde{\nu}, \tilde{b}, \tilde{\alpha} = \alpha_{m_n}, 0)$ from Proposition 5). Using the concentration results from Section A.1 we can rewrite the above inequality as follows:

$$\frac{2}{\underline{m} t_{\underline{m}}^2} \left( 1 + O\left( \frac{1}{t_{\underline{m}}^2} \right) \right) > \frac{2c}{\kappa_2 n} \frac{\kappa_M}{\|w^*\|_1^{2/3}} \left( \frac{n}{t_{m_n}^2} \right)^{1/3} \left( 1 + O\left( \frac{1}{t_{m_n}^2} \right) \right)$$

After recalling that $t_m^2 = 2 \log(d/m) + O(\log\log(d/m))$ from Section A.1, it is straightforward to show that we can choose $\underline{m} = \lambda \left( \frac{n \|w^*\|_1}{t_{m_n}^2} \right)^{2/3}$ with sufficiently small universal constant $\lambda > 0$. We finish this step by substituting this choice of $\underline{m}$ into the upper bound from Equation (25) to get:

$$\eta_{\mathcal{S}^c}^2 \leq M^2 \frac{\|\gamma(\alpha_{\underline{m}})\|_2^2}{\alpha_{\underline{m}}^2} = \kappa_M^2 \left( \frac{n \|w^*\|_1}{t_{m_n}^2} \right)^{2/3} \left( 1 + O\left( \frac{1}{t_{m_n}^2} \right) \right) \frac{2}{\underline{m}} \left( 1 + O\left( \frac{1}{t_{\underline{m}}^2} \right) \right) =: B_{\eta_{\mathcal{S}^c}}^2 = O(1)$$

**Step 2.2: Upper bound $\eta_{\mathcal{S}} = O(1)$.** Similarly as in the previous step we use the relaxations of the constraints from the set $\Gamma_2$ to bound $\eta_{\mathcal{S}}$ as follows:

$$\eta_{\mathcal{S}}^2 \leq \max_{(\nu, b, \alpha, \eta_{\mathcal{S}}) \in \Gamma_2} \eta_{\mathcal{S}}^2$$

$$\leq \max_{\nu, b, \alpha, \eta_{\mathcal{S}}} \left[ \eta_{\mathcal{S}}^2 \quad \text{s.t} \quad \eta_{\mathcal{S}}^2 \leq \frac{2}{\kappa_3 n} \nu b^2 \|h\|_\infty^2 \quad \text{and} \quad \|\gamma(\alpha)\|_2^2 \leq \frac{2}{\kappa_2 n} \nu \|h\|_\infty^2 \right.$$

$$\left. \text{and} \quad b\alpha \leq M \quad \text{and} \quad \nu \|w^*\|_1 \leq cM \right]$$

$$\leq \max_{\nu, b, \alpha} \left[ \frac{2}{\kappa_3 n} \nu b^2 \|h\|_\infty^2 \quad \text{s.t} \quad \|\gamma(\alpha)\|_2^2 \leq \frac{2}{\kappa_2 n} \nu \|h\|_\infty^2 \quad \text{and} \quad b \leq \frac{M}{\alpha} \quad \text{and} \quad \nu \leq c \frac{M}{\|w^*\|_1} \right]$$

$$\leq \frac{2}{\kappa_3 n} \frac{cM}{\|w^*\|_1} M^2 \|h\|_\infty^2 \max_\alpha \left[ \frac{1}{\alpha^2} \quad \text{s.t} \quad \|\gamma(\alpha)\|_2^2 \leq \frac{2c}{\kappa_2 n} \frac{M}{\|w^*\|_1} \|h\|_\infty^2 \right]$$

Now note that $\frac{1}{\alpha^2}$ is monotonically decreasing function, while the last constraint is identical to constraint from Equation (25). Thus using exactly the same arguments as in the previous step we upper bound $\eta_{\mathcal{S}}^2$ as follows:

$$\eta_{\mathcal{S}}^2 \leq \frac{2c}{\kappa_3 n} \frac{M^3}{\|w^*\|_1} \frac{\|h\|_\infty^2}{\alpha_{\underline{m}}^2} = \frac{2c}{\kappa_3 n} \kappa_M^3 \frac{n}{t_{m_n}^2} t_{\underline{m}}^2 (1 + O\left(\frac{1}{t_{m_n}^2}, \frac{1}{t_{\underline{m}}^2}\right)) =: B_{\eta_{\mathcal{S}}}^2 = O(1)$$

where we again used concentration results from Proposition 3, and approximation $t_m^2 = 2\log(d/m) + O(\log\log(d/m))$ from Section A.1.

**Step 2.3: Lower bound $\nu \|w^*\|_1 \geq \underline{\kappa} M$.** This bound follows the same reasoning as the previous two steps. Namely, we find a lower bound on $\nu$ as follows:

$$\nu \geq \min_{(\nu, b, \alpha, \eta_{\mathcal{S}}) \in \Gamma_2} \nu$$

$$\geq \min_{\nu, b, \alpha} \left[ \nu \ \text{ s.t } \ \nu \geq \frac{\kappa_1}{2} \frac{n}{b^2 \|h\|_\infty^2} \ \text{ and } \ \|\gamma(\alpha)\|_2^2 \leq \frac{2}{\kappa_2 n} \nu \|h\|_\infty^2 \ \text{ and } \ b\alpha \leq M \ \text{ and } \ \nu \|w^*\|_1 \leq cM \right]$$

$$\geq \min_{\nu, b, \alpha} \left[ \frac{\kappa_1}{2} \frac{n}{b^2 \|h\|_\infty^2} \ \text{ s.t } \ \|\gamma(\alpha)\|_2^2 \leq \frac{2}{\kappa_2 n} \nu \|h\|_\infty^2 \ \text{ and } \ b \leq \frac{M}{\alpha} \ \text{ and } \ \nu \|w^*\|_1 \leq cM \right]$$

$$= \frac{\kappa_1 n}{2M^2 \|h\|_\infty^2} \min_\alpha \left[ \alpha^2 \ \text{ s.t } \ \|\gamma(\alpha)\|_2^2 \leq \frac{2c}{\kappa_2 n} \frac{M}{\|w^*\|_1} \|h\|_\infty^2 \right]$$

Similarly as in the previous two steps, since $\alpha^2$ is monotonically increasing function, the minimum is lower bounded by $\alpha^2 \geq \alpha_{\underline{m}}^2$ and after substitution of $\underline{m}$ as defined above, we have:

$$\nu \geq \frac{\kappa_1 n}{2M^2} \frac{\alpha_{\underline{m}}^2}{\|h\|_\infty^2} = \frac{\kappa_1}{2\kappa_M^2} \|w^*\|_1^{-2/3} \left(\frac{n}{t_{m_n}^2}\right)^{1/3} \frac{t_{m_n}^2}{t_{\underline{m}}^2} (1 + O\left(\frac{1}{t_{m_n}^2}\right)) =: \underline{\kappa} \frac{M}{\|w^*\|_1}$$

where once again we applied Proposition 3, and used that $t_m^2 = 2\log(d/m) + O(\log\log(d/m))$ from Section A.1. After noting that we have shown $\nu \|w^*\|_1 \geq \underline{\kappa} M$ with high probability, we conclude this part of the proof.

**Bound 3: Tight bounds**

From Step 2.2 in the previous bound we have $\eta_{\mathcal{S}} = O(1)$ and thus $\sqrt{s}\eta_{\mathcal{S}} \leq \sqrt{s_{\max}}\eta_{\mathcal{S}} = O(n^{1/3}\log^{-7/3} d)$. Combining this bound with the lower bound $M \gtrsim \left(\frac{n}{\log d}\right)^{1/3}$, we obtain that:

$$\nu \|w^*\|_1 \leq M + \sqrt{s}\eta_{\mathcal{S}} \leq M\left(1 + \frac{c_1}{\log^2 d}\right) \leq \kappa_M \left(\frac{n}{t_{m_n}^2}\|w^*\|_1\right)^{1/3}\left(1 - \frac{2}{3}\frac{1}{t_{m_n}^2} + \frac{c_2}{t_{m_n}^4}\right) =: \widetilde{M} \quad (26)$$

for some fixed universal constant $c_1, c_2 > 0$. Moreover, in Step 2.3 of the previous bound we have shown that $\nu \|w^*\|_1 \geq \underline{\kappa} M$, and thus $\nu \|w^*\|_1 \geq \widetilde{\underline{\kappa}}\widetilde{M}$ for some $0 < \widetilde{\underline{\kappa}} \leq \underline{\kappa}$. Combining both results, we have $\nu \|w^*\|_1 \in [\widetilde{\underline{\kappa}}, 1]\widetilde{M}$.

Now we show how we can relax and simplify the first constraint of the set $\Gamma$. Recall Equation (24) and note that it implies $\frac{2}{n}b^2\|h\|_\infty^2 + \frac{8}{n}s_{\max}\eta_{\mathcal{S}}^2 \geq \kappa_1 \frac{1}{\nu}$. Moreover, since $\nu \|w^*\|_1 \leq \widetilde{M}$, and $\eta_{\mathcal{S}} \leq B_{\eta_{\mathcal{S}}}$ from Step 2.2, we have:

$$\frac{1}{n}b^2\|h\|_\infty^2 \geq \frac{\kappa_1}{2}\frac{1}{\nu} - \frac{4s_{\max}}{n}B_{\eta_{\mathcal{S}}}^2 \geq \frac{\kappa_1}{2}\frac{\|w^*\|_1}{\widetilde{M}} - 4B_{\eta_{\mathcal{S}}}^2\frac{s_{\max}}{n} \gtrsim \frac{1}{n^{1/3}} - \frac{1}{n^{1/3}\log^{14/3} d} \gtrsim \frac{1}{n^{1/3}}$$

for $n$ large enough, since $s_{\max} = \Theta\left(n^{2/3}\log^{-14/3} d\right)$ and $\frac{\|w^*\|_1}{\widetilde{M}} \geq \frac{1}{2}\frac{\|w^*\|_1}{M} \gtrsim \frac{1}{n^{1/3}}$. Thus, using this lower bound on $b\|h\|_\infty$ and upper bound $\eta_{\mathcal{S}} \leq B_{\eta_{\mathcal{S}}}$ we have:

$$\frac{1}{n}(2\sqrt{s_{\max}}\eta_{\mathcal{S}} + b\|h\|_\infty)^2 = \frac{1}{n}b^2\|h\|_\infty^2\left(1 + \frac{2\sqrt{s_{\max}}\eta_{\mathcal{S}}}{b\|h\|_\infty}\right)^2 \leq \frac{1}{n}b^2\|h\|_\infty^2(1 + \mathcal{O}_b)^2$$

where we defined $\mathcal{O}_b = c \log^{-7/3} d$ for some universal constant $c > 0$. This finishes our relaxation of the LHS of the first constraint from the definition of $\Gamma$.

For the RHS of this constraint, we can apply Corollary 1 with $\epsilon \asymp \frac{1}{n^{1/3} t_{m_n}^4}$ to obtain that the inequality

$$f_n(\nu, \eta) \geq \frac{\sqrt{2}}{3\sqrt{\pi}} \frac{1}{\nu} + \sqrt{\frac{2}{\pi}} \frac{\eta^2}{\nu} - \epsilon$$

holds with probability at least $1 - c_1 \exp\left(-c_2 \frac{n^{1/3}}{t_{m_n}^8}\right)$.

Now we use the derived relaxations of the first constraint of $\Gamma$ to define a new set $\Gamma_3$:

$$\Gamma_3 = \left\{ (\nu, b, \alpha, \eta_{\mathcal{S}}) \text{ s.t } \frac{1}{n} b^2 \|h\|_\infty^2 (1 + \mathcal{O}_b) \geq \frac{\sqrt{2}}{3\sqrt{\pi}} \frac{1}{\nu} + \sqrt{\frac{2}{\pi}} \frac{b^2 \|\gamma(\alpha)\|_2^2 + \eta_{\mathcal{S}}^2}{\nu} - \epsilon \right.$$

$$\left. \text{and } b\alpha + \nu \|w^*\|_1 \leq \widetilde{M} \text{ and } \nu \|w^*\|_1 \in [\widetilde{\underline{\kappa}}, 1]\widetilde{M} \text{ and } b\|\gamma(\alpha)\|_2 + \eta_{\mathcal{S}} \lesssim 1 \right\}.$$

Again, we have that with high probability $\Gamma \subset \Gamma_3$ and in the following four steps we bound variables $\alpha, \nu, \eta_{\mathcal{S}^c}, \eta_{\mathcal{S}}$ such that $(\nu, b, \alpha, \eta_{\mathcal{S}^c}) \in \Gamma_3$. Furthermore, in the following, steps we will use multiple times the fact that:

$$\frac{1}{t_{m_n}^4} \gtrsim \frac{1}{\log^2(d/m_n)} \gtrsim \frac{1}{\log^2 d}$$

which follows from characterization of $t_m^2$ from Section A.1.

In order to derive tight bounds on $\nu, \eta_{\mathcal{S}^c}, \eta_{\mathcal{S}}$ in Steps 3.3, 3.4. and 3.5, respectively, we first need to show an upper and lower bound on $\alpha$ in Steps 3.1 and 3.2, respectively.

**Step 3.1: Upper bound $\alpha \leq \alpha_{\lambda m_n} (\lambda > 1)$.**

We upper bound $\alpha$ uniformly over $\Gamma_3$ as follows:

$$\alpha^2 \leq \max_{(\nu, b, \alpha, \eta_{\mathcal{S}}) \in \Gamma_3} \alpha^2$$

$$\leq \max_{\nu, b, \alpha} \left[ \alpha^2 \text{ s.t } \frac{1}{n} b^2 \|h\|_\infty^2 (1 + \mathcal{O}_b) \geq \frac{\sqrt{2}}{3\sqrt{\pi}} \frac{1}{\nu} - \epsilon \text{ and } b\alpha + \nu \|w^*\|_1 \leq \widetilde{M} \text{ and } \nu \|w^*\|_1 \in [\widetilde{\underline{\kappa}}, 1]\widetilde{M} \right]$$

$$\leq \max_{\nu, \alpha} \left[ \alpha^2 \text{ s.t } \frac{1}{n} \left( \frac{\widetilde{M} - \nu \|w^*\|_1}{\alpha} \right)^2 \|h\|_\infty^2 (1 + \mathcal{O}_b) \geq \frac{\sqrt{2}}{3\sqrt{\pi}} \frac{1}{\nu} - \epsilon \text{ and } \nu \|w^*\|_1 \in [\widetilde{\underline{\kappa}}, 1]\widetilde{M} \right]$$

$$\leq \max_{\nu, \alpha} \left[ \alpha^2 \text{ s.t } \frac{\alpha^2}{\|h\|_\infty^2} \left( 1 - \frac{3\sqrt{\pi}}{\sqrt{2}} \epsilon \nu \right) \leq \frac{3\sqrt{\pi}}{\sqrt{2}} \frac{1}{n} \nu (\widetilde{M} - \nu \|w^*\|_1)^2 (1 + \mathcal{O}_b) \text{ and } \nu \|w^*\|_1 \in [\widetilde{\underline{\kappa}}, 1]\widetilde{M} \right]$$

$$\leq \max_{\alpha} \left[ \alpha^2 \text{ s.t } \frac{\alpha^2}{\|h\|_\infty^2} \leq \frac{12\sqrt{\pi}}{27\sqrt{2}} \frac{1}{n} \frac{\widetilde{M}^3}{\|w^*\|_1} \left( 1 + O\left( \frac{1}{t_{m_n}^4} \right) \right) \right] \tag{27}$$

where in the second line we used the second constraint to upper bound $b$, and in the last line we used that $(1 + \mathcal{O}_b)(1 - \frac{3\sqrt{\pi}}{\sqrt{2}} \epsilon \nu)^{-1} \leq 1 + O(\frac{1}{t_{m_n}^4})$ and that the function $\nu(\widetilde{M} - \nu \|w^*\|_1)^2$ under the constraint $\nu \|w^*\|_1 \in [\widetilde{\underline{\kappa}}, 1]\widetilde{M}$ is maximized for $\nu \|w^*\|_1 = \widetilde{M}/3$. Furthermore, note that $1/3 \in [\widetilde{\underline{\kappa}}, 1]$ since $\Gamma \subset \Gamma_3$ and point $\nu = \frac{M}{3\|w^*\|_1} \in \Gamma$ by arguments from the proof of the localization proposition 4.

Similarly as in the previous bounds, we use that $\alpha^2$ is a monotonically increasing convex function, and thus in order to lower bound $\|\gamma(\alpha)\|_2^2$, it is sufficient to find a point $\alpha_{\overline{m}}$ such that $\alpha_{\overline{m}} \geq \alpha_m$ for which the constraint of Equation (27) does not hold. Now, using concentration result from Proposition 3 and definition of $\widetilde{M}$, we have that $\alpha = \alpha_{\overline{m}}$ does not satisfy the constraint if:

$$\frac{1}{t_{\overline{m}}^2} \left( 1 - \frac{4}{t_{\overline{m}}^2} + O\left( \frac{1}{t_{\overline{m}}^4} \right) \right) > \frac{1}{t_{m_n}^2} \left( 1 - \frac{2}{t_{m_n}^2} + O\left( \frac{1}{t_{m_n}^4} \right) \right)$$

We can choose $\overline{m} = \lambda m_n$ for some constant $\lambda > 1$ since using characterization of $t_m$ from Section A.1 we have:

$$\frac{t_{m_n}^2}{t_{\overline{m}}^2} = 1 + \frac{2 \log \lambda}{t_{m_n}^2} + O\left(\frac{1}{t_{m_n}^4}\right)$$

Thus we finally obtain that $\alpha \leq \alpha_{\overline{m}}$, as we wanted to show.

**Step 3.2: Lower bound** $\alpha \geq \alpha_{\lambda m_n} (\lambda \in (0,1))$

The bound in this step is derived similarly to the bound in Step 3.1. However, in this step we cannot neglect the term $\sqrt{\frac{2}{\pi}} \frac{b^2 \|\gamma(\alpha)\|_2^2}{\nu}$ from the first constraint of $\Gamma_3$, as we did in the previous step. For the sake of shorter equations, we will write only relaxations of constraints that $\alpha$ needs to satisfy and skip writing that we minimize over $\alpha^2$ like we did previously.

We start by rewriting and relaxing the first constraint from $\Gamma_3$ as follows:

$$b^2 \left( \frac{\|h\|_\infty^2}{n}(1 + \mathcal{O}_b) - \sqrt{\frac{2}{\pi}} \frac{\|\gamma(\alpha)\|_2^2}{\nu} \right) \geq \frac{\sqrt{2}}{3\sqrt{\pi}} \frac{1}{\nu} + \sqrt{\frac{2}{\pi}} \frac{\eta_{\mathcal{S}}^2}{\nu} - \epsilon$$

$$\geq \frac{\sqrt{2}}{3\sqrt{\pi}} \frac{1}{\nu} - \epsilon \geq \frac{\sqrt{2}}{3\sqrt{\pi}} \frac{1}{\nu}\left(1 - \mathcal{O}\left(\frac{1}{t_{m_n}^4}\right)\right) \tag{28}$$

where we used that $\epsilon\nu = O(\frac{1}{t_{m_n}^4})$. Now, using the second constraint of $\Gamma_3$, we can further relax the LHS of the previous inequality as follows:

$$b^2 \left( \frac{\|h\|_\infty^2}{n}(1 + \mathcal{O}_b) - \sqrt{\frac{2}{\pi}} \frac{\|\gamma(\alpha)\|_2^2}{\nu} \right) \leq \frac{(\widetilde{M} - \nu \|w^*\|_1)^2}{\alpha^2} \left( \frac{\|h\|_\infty^2}{n}(1 + \mathcal{O}_b) - \sqrt{\frac{2}{\pi}} \frac{\|\gamma(\alpha)\|_2^2}{\nu} \right) \tag{29}$$

Combining inequalities (28) and (29), and plugging in $\nu \|w^*\|_1 = \kappa\widetilde{M}$ for $\kappa \in [\widetilde{\kappa}, 1]$ yields:

$$\frac{\sqrt{2}}{3\sqrt{\pi}} \frac{\|w^*\|_1}{\kappa\widetilde{M}}\left(1 - O\left(\frac{1}{t_{m_n}^4}\right)\right) \leq \frac{\widetilde{M}^2(1 - \kappa)^2}{\alpha^2} \left( \frac{\|h\|_\infty^2}{n}(1 + \mathcal{O}_b) - \sqrt{\frac{2}{\pi}} \frac{\|\gamma(\alpha)\|_2^2 \|w^*\|_1}{\kappa\widetilde{M}} \right)$$

After multiplying the previous inequality by $\frac{\kappa\widetilde{M}}{\|w^*\|_1} \frac{\alpha^2}{\widetilde{M}^2(1-\kappa)^2}$ and rearranging terms, we obtain:

$$\sqrt{\frac{2}{\pi}} \|\gamma(\alpha)\|_2^2 \leq \frac{\|h\|_\infty^2}{n} \frac{\kappa\widetilde{M}}{\|w^*\|_1}(1 + \mathcal{O}_b) - \frac{\sqrt{2}}{3\sqrt{\pi}} \frac{\alpha^2}{\widetilde{M}^2(1 - \kappa)^2}\left(1 - O\left(\frac{1}{t_{m_n}^4}\right)\right) \tag{30}$$

Note that only the right hand side depends on $\nu$ (and thus on $\kappa$). Hence maximizing over $\kappa$ the right hand side we obtain:

$$\kappa = 1 - \left( \frac{2\sqrt{2}}{3\sqrt{\pi}} \frac{n\alpha^2 \|w^*\|_1}{\|h\|_\infty^2 \widetilde{M}^3}\left(1 - O\left(\frac{1}{t_{m_n}^4}\right)\right) \right)^{1/3} \geq 1 - \left( \frac{2\sqrt{2}}{3\sqrt{\pi}} \frac{n\alpha_{\overline{m}}^2 \|w^*\|_1}{\|h\|_\infty^2 \widetilde{M}^3}\left(1 - O\left(\frac{1}{t_{m_n}^4}\right)\right) \right)^{1/3} > \frac{1}{3}$$

where we used that $\alpha \geq \alpha_{\overline{m}}$ derived in the previous step. Moreover, note that $\kappa \in [\widetilde{\kappa}, 1]\widetilde{M}$, by the proof of our localization proposition. Substituting this $\kappa$ into (30) we get the following inequality:

$$\frac{\alpha^{2/3}}{\|h\|_\infty^{2/3}} \left( \frac{9\sqrt{2}}{4\sqrt{\pi}} \right)^{1/3}\left(1 - \mathcal{O}\left(\frac{1}{t_{m_n}^4}\right)\right) + n^{2/3}\sqrt{\frac{2}{\pi}} \frac{\|\gamma(\alpha)\|_2^2}{\|h\|_\infty^2} \|w^*\|_1^{2/3} \leq \frac{1}{n^{1/3}} \frac{\widetilde{M}}{\|w^*\|_1^{1/3}}(1 + \mathcal{O}_b)$$

Now we further relax the constraint by raising the previous inequality to the third power and keeping only the first two terms to get:

$$\frac{\alpha^2}{\|h\|_\infty^2} \frac{9\sqrt{2}}{4\sqrt{\pi}} + 3\sqrt{\frac{2}{\pi}} \frac{\|\gamma(\alpha)\|_2^2}{\|h\|_\infty^2} \|w^*\|_1^{2/3} n^{2/3} \frac{\alpha^{4/3}}{\|h\|_\infty^{4/3}} \left( \frac{9\sqrt{2}}{4\sqrt{\pi}} \right)^{2/3} \leq \frac{1}{n} \frac{\widetilde{M}^3}{\|w^*\|_1}\left(1 + O\left(\frac{1}{t_{m_n}^4}\right)\right)$$

We can further relax this constraint by using the that $\alpha \geq \alpha_{\underline{m}}$ with $\underline{m} = \lambda \left(\frac{n\|w^*\|_1}{t_{m_n}^2}\right)^{2/3}$ as shown in the Bound 2. Then, we have substitute this value of $\alpha$ only in the second term as follows:

$$\frac{\alpha^2}{\|h\|_\infty^2}\frac{9\sqrt{2}}{4\sqrt{\pi}} + 3\sqrt{\frac{2}{\pi}}\frac{\|\gamma(\alpha)\|_2^2}{\|h\|_\infty^2}\|w^*\|_1^{2/3}n^{2/3}\frac{\alpha_{\underline{m}}^{4/3}}{\|h\|_\infty^{4/3}}\left(\frac{9\sqrt{2}}{4\sqrt{\pi}}\right)^{2/3} \leq \frac{1}{n}\frac{\widetilde{M}^3}{\|w^*\|_1}(1 + O\left(\frac{1}{t_{m_n}^4}\right))$$

Now, note that the term on the left hand side is a sum of two convex functions in $\alpha$ and thus is convex. Similarly, as before, we look for $\alpha_{\underline{\underline{m}}} < \alpha_m$ so that the previous inequality is not satisfied. Using concentration results from Proposition 3, we get:

$$3\sqrt{\frac{2}{\pi}}\left(\frac{9\sqrt{2}}{4\sqrt{\pi}}\right)^{2/3}\|w^*\|_1^{2/3}\frac{2n^{2/3}}{\underline{m}t_{\underline{m}}^2}\left(1 + O\left(\frac{1}{t_{\underline{m}}^2}\right)\right)\frac{1}{t_{\underline{m}}^{4/3}}\left(1 - \frac{8}{3}\frac{1}{t_{\underline{m}}^2} + O\left(\frac{1}{t_{\underline{m}}^4}\right)\right)$$

$$+ \frac{9\sqrt{2}}{4\sqrt{\pi}}\frac{1}{t_{\underline{m}}^2}\left(1 - \frac{4}{t_{\underline{m}}^2} + O\left(\frac{1}{t_{\underline{m}}^4}\right)\right) > \frac{9\sqrt{2}}{4\sqrt{\pi}}\frac{1}{t_{m_n}^2}\left(1 - \frac{2}{t_{m_n}^2} + O\left(\frac{1}{t_{m_n}^4}\right)\right)$$

and we can choose $\underline{m} = \lambda m_n$ with $\lambda \in (0,1)$. This gives us a lower bound on $\alpha$ which is tight enough to obtain bounds on $\nu$ with a right multiplicative constant.

**Step 3.3: Tight bounds in $\nu$**

Now consider a set $\Gamma_3^\nu := \Gamma_3 \cap \{(\nu, b, \alpha, \eta_S) \text{ s.t } \alpha \geq \alpha_{\underline{\underline{m}}}\}$ with $\underline{\underline{m}}$ given in the previous step. Furthermore, from the arguments in the previous step it holds that $\widehat{\Gamma} \subset \Gamma_3^\nu$ with high probability.

Now, similarly to Step 3.1 we can relax the first constraint of $\Gamma_3$ to $\frac{1}{n}b^2\|h\|_\infty^2 \geq \frac{\sqrt{2}}{3\sqrt{\pi}}\frac{1}{\nu}(1 - O\left(\frac{1}{t_{m_n}^4}\right))$. Combining this lower bound on $b$ with the second constraint of $\Gamma_3$ we have:

$$\widetilde{M} - \nu\|w^*\|_1 \geq b\alpha \geq \sqrt{\frac{\sqrt{2}}{3\sqrt{\pi}}}\frac{\sqrt{n}}{\|h\|_\infty}\frac{\alpha}{\sqrt{\nu}}(1 - O\left(\frac{1}{t_{m_n}^4}\right))$$

Rearranging the terms we obtain that for any $(\nu, b, \alpha, \eta_S) \in \Gamma_3^\nu$ must hold that:

$$0 \geq \nu^{3/2}\|w^*\|_1 - \widetilde{M}\nu^{1/2} + \sqrt{\frac{\sqrt{2}}{3\sqrt{\pi}}}\sqrt{n}\frac{\alpha}{\|h\|_\infty}(1 - O\left(\frac{1}{t_{m_n}^4}\right))$$

$$\geq \nu^{3/2}\|w^*\|_1 - \widetilde{M}\nu^{1/2} + \sqrt{\frac{\sqrt{2}}{3\sqrt{\pi}}}\sqrt{n}\frac{\alpha_{\underline{\underline{m}}}}{\|h\|_\infty}(1 - O\left(\frac{1}{t_{m_n}^4}\right))$$

where we used that $\alpha \geq \alpha_{\underline{\underline{m}}}$ Thus, the constraint B.2 must hold uniformly for all $\nu \in \Gamma_3^\nu$. Setting $\nu\|w^*\|_1 = \kappa^2\widetilde{M}$ with $\kappa^2 \in [\widetilde{\underline{\kappa}}, 1]$ we obtain the following constraint on $\kappa$:

$$\kappa^3 - \kappa + \sqrt{\frac{\sqrt{2}}{3\sqrt{\pi}}\frac{n\|w^*\|_1}{\widetilde{M}^3}}\frac{\alpha_{\underline{\underline{m}}}}{\|h\|_\infty}(1 - O\left(\frac{1}{t_{m_n}^4}\right)) \leq 0$$

Using definition of $\widetilde{M}$ from Equation (26) and concentration inequality from Proposition 3 we obtain

$$\kappa^3 - \kappa + \frac{2}{3\sqrt{3}}\frac{t_{m_n}}{t_{\underline{\underline{m}}}}\left(1 - \frac{2}{t_{\underline{\underline{m}}}^2} + O\left(\frac{1}{t_{\underline{\underline{m}}}^4}\right)\right)\left(1 + \frac{1}{t_{m_n}^2} + O\left(\frac{1}{t_{m_n}^4}\right)\right) \leq 0$$

and after substituting $\underline{\underline{m}} = \lambda m_n$ with $\lambda < 1$ we get the following:

$$\kappa^3 - \kappa + \frac{2}{3\sqrt{3}} + \frac{2}{3\sqrt{3}}\frac{\log\lambda - 1}{t_{m_n}^2} + O\left(\frac{1}{t_{m_n}^4}\right) \leq 0$$

Thus, we obtain $\kappa^2 \in \left[\frac{1}{3} - \frac{\tilde{\lambda}}{t_{m_n}^{2/3}}, \frac{1}{3} + \frac{\tilde{\lambda}}{t_{m_n}^{2/3}}\right]$ for some positive universal constant $\tilde{\lambda}$, which we can write as $\nu\|w^*\|_1 = \frac{\widetilde{M}}{3}(1 + O(t_{m_n}^{-2/3}))$.

**Step 3.4: Tight bounds on $\eta_{\mathcal{S}^c}$**

Define $\Gamma_3^{\eta_{\mathcal{S}^c}} := \Gamma_3 \cap \left\{ (\nu, b, \alpha, \eta_{\mathcal{S}}) \text{ s.t } \left| \nu \|w^*\|_1 - \frac{\widetilde{M}}{3} \right| \le \frac{\tilde{\lambda}\widetilde{M}}{t_{m_n}^{2/3}} \right\}$. Since inequality (30) holds for $\Gamma_3$, it also holds for $\Gamma_3^{\eta_{\mathcal{S}^c}}$. Multiplying this inequality by $\frac{n\|w^*\|_1}{\widetilde{M}\|h\|_\infty^2}(1-\kappa)^2(1+\mathcal{O}_b)^{-1}$, we get:

$$\sqrt{\frac{2}{\pi}}\frac{\|\gamma(\alpha)\|_2^2}{\|h\|_\infty^2}\frac{n\|w^*\|_1}{\widetilde{M}}(1-\kappa)^2(1+\mathcal{O}_b)^{-1} + \frac{\sqrt{2}}{3\sqrt{\pi}}\frac{\alpha^2}{\|h\|_\infty^2}\frac{n\|w^*\|_1}{\widetilde{M}^3}(1+\mathcal{O}_b)^{-1}\left(1-O\left(\frac{1}{t_{m_n}^4}\right)\right)$$

$$\le \kappa(1-\kappa)^2 \le \frac{4}{27}$$

and using our established bound on $\nu\|w^*\|_1$ we get $(1-\kappa)^2 \ge (1-\frac{1}{3}-\frac{\tilde{\lambda}}{t_{m_n}^{2/3}})^2 = \frac{4}{9}(1-\frac{3\tilde{\lambda}}{t_{m_n}^{2/3}}+O(\frac{1}{t_{m_n}^{4/3}}))$ and hence we obtain:

$$3\sqrt{\frac{2}{\pi}}\frac{\|\gamma(\alpha)\|_2^2}{\|h\|_\infty^2}\frac{n\|w^*\|_1}{\widetilde{M}}\left(1-\frac{3\tilde{\lambda}}{t_{m_n}^{2/3}}+O\left(\frac{1}{t_{m_n}^{4/3}}\right)\right) + \frac{9\sqrt{2}}{4\sqrt{\pi}}\frac{\alpha^2}{\|h\|_\infty^2}\frac{n\|w^*\|_1}{\widetilde{M}^3}\left(1-O\left(\frac{1}{t_{m_n}^4}\right)\right) \le 1$$

Note that the function is convex in $\alpha$. Using concentration, we get for $\alpha = \alpha_m$:

$$2\frac{2\left(\frac{3}{\pi}\right)^{1/3}(nt_{m_n}\|w^*\|_1)^{2/3}}{mt_m^2}\left(1+O\left(\frac{1}{t_m^2}\right)\right)\left(1-\frac{3\tilde{\lambda}}{t_{m_n}^{2/3}}+O\left(\frac{1}{t_{m_n}^{4/3}}\right)\right)$$

$$+\frac{t_{m_n}^2}{t_m^2}\left(1-\frac{4}{t_m^2}+O\left(\frac{1}{t_m^4}\right)\right)\left(1+\frac{2}{t_{m_n}^2}+O\left(\frac{1}{t_{m_n}^4}\right)\right) \le 1$$

Now we claim that the $\underline{m}_* < m_n, \overline{m}_* > m_n$ given in Lemma 2, respectively, with $\kappa = 1/3$ and parameter $\mu$ do not satisfy this inequality for the well-chosen universal constant $\mu$ since

$$2\frac{2\left(\frac{3}{\pi}\right)^{1/3}(nt_{m_n}\|w^*\|_1)^{2/3}}{\underline{m}_* t_{\underline{m}_*}^2}\left(1-\frac{3\tilde{\lambda}}{t_{m_n}^{2/3}}+O\left(\frac{1}{t_{m_n}^{4/3}}\right)\right) + \frac{t_{m_n}^2}{t_{\underline{m}_*}^2}\left(1-\frac{2}{t_{m_n}^2}+O\left(\frac{1}{t_{m_n}^4}\right)\right)$$

$$= 1 - \frac{\mu}{t_{m_n}^{7/3}}\left(1-\frac{t_{m_n}^2}{t_{\underline{m}_*}^2}\right) + \frac{2\mu^2-6\tilde{\lambda}}{t_{\underline{m}_*}^2 t_{m_n}^{2/3}} + O\left(\frac{1}{t_{m_n}^{10/3}}\right) = 1 + \frac{2\mu^2-6\tilde{\lambda}}{t_{m_n}^{8/3}} + O\left(\frac{1}{t_{m_n}^{10/3}}\right) > 1$$

for $\mu > \sqrt{3\tilde{\lambda}}$. Similarly, for $\overline{m}_*$ we get:

$$2\frac{2\left(\frac{3}{\pi}\right)^{1/3}(nt_{m_n}\|w^*\|_1)^{2/3}}{\overline{m}_* t_{\overline{m}_*}^2}\left(1-\frac{3\tilde{\lambda}}{t_{m_n}^{2/3}}+O\left(\frac{1}{t_{m_n}^{4/3}}\right)\right) + \frac{t_{m_n}^2}{t_{\overline{m}_*}^2}\left(1-\frac{2}{t_{m_n}^2}+O\left(\frac{1}{t_{m_n}^4}\right)\right)$$

$$= 1 + \frac{\mu}{t_{m_n}^{7/3}}\left(1-\frac{t_{m_n}^2}{t_{\overline{m}_*}^2}\right) + \frac{2\mu^2-6\tilde{\lambda}}{t_{\overline{m}_*}^2 t_{m_n}^{2/3}} + O\left(\frac{1}{t_{m_n}^{10/3}}\right) = 1 + \frac{2\mu^2-6\tilde{\lambda}}{t_{m_n}^{8/3}} + O\left(\frac{1}{t_{m_n}^{10/3}}\right) > 1$$

In order to bound $\eta_{\mathcal{S}^c}$ we use that $b \le \frac{\widetilde{M}-\nu\|w^*\|_1}{\alpha}$, $\alpha \ge \alpha_{\underline{m}_*}$, and $\nu \ge \widetilde{M}(\frac{1}{3}-\frac{\tilde{\lambda}}{t_{m_n}^{2/3}})$, respectively, to obtain:

$$\eta_{\mathcal{S}^c}^2 \le \max_{(\nu,b,\alpha,\eta_{\mathcal{S}})\in\Gamma_3^{\eta_{\mathcal{S}^c}}} b^2\|\gamma(\alpha)\|_2^2 \le \max_{\nu,\alpha}(\widetilde{M}-\nu\|w^*\|_1)^2\frac{\|\gamma(\alpha)\|_2^2}{\alpha^2}$$

$$\le \widetilde{M}^2\left(1-\frac{1}{3}+\frac{\tilde{\lambda}}{t_{m_n}^{2/3}}\right)^2\frac{\|\gamma(\alpha_{\underline{m}_*})\|_2^2}{\alpha_{\underline{m}_*}^2}$$

and after application of concentration Proposition 3 and definition of $\widetilde{M}$ we obtain:

$$\eta_{\mathcal{S}^c}^2 \le \frac{2}{3}\frac{1}{t_{m_n}^2}\exp\left(\frac{\mu}{2t_{m_n}^{1/3}}\right)\left(1+O\left(\frac{1}{t_{m_n}^{2/3}}\right)\right) = \frac{2}{3}\frac{1}{t_{m_n}^2}\left(1+\frac{\mu}{2t_{m_n}^{1/3}}+O\left(\frac{1}{t_{m_n}^{2/3}}\right)\right)$$

and

$$
\begin{aligned}
\eta_{\mathcal{S}^c}^2 &\geq \min_{(\nu,b,\alpha,\eta_{\mathcal{S}})\in\Gamma_3^{\eta_{\mathcal{S}^c}}} b^2 \left\|\gamma(\alpha)\right\|_2^2 \geq \min_{\nu,\alpha} \frac{\sqrt{2}}{3\sqrt{\pi}} \frac{n}{\nu} \frac{\left\|\gamma(\alpha)\right\|_2^2}{\left\|h\right\|_\infty^2} (1 - O\left(\frac{1}{t_{m_n}^4}\right)) \\
&\geq \frac{\sqrt{2}}{3\sqrt{\pi}} \frac{n\left\|w^*\right\|_1}{\widetilde{M}\left(\frac{1}{3} + \frac{\tilde{\lambda}}{t_{m_n}^{2/3}}\right)} \frac{2}{\overline{m}_* t_{\overline{m}^*}^2} \left(1 + O\left(\frac{1}{t_{\overline{m}^*}^2}\right)\right) \geq \frac{2}{3} \frac{1}{t_{m_n}^2} \exp\left(-\frac{\mu}{2t_{m_n}^{1/3}}\right)\left(1 - O\left(\frac{1}{t_{m_n}^{2/3}}\right)\right) \\
&\geq \frac{2}{3} \frac{1}{t_{m_n}^2} \left(1 - \frac{\mu}{2t_{m_n}^{1/3}} - O\left(\frac{1}{t_{m_n}^{2/3}}\right)\right),
\end{aligned}
$$

which are the upper and lower bound claimed in the Proposition 5.

**Step 3.5: Tight upper bound on $\eta_{\mathcal{S}}$**

Define $\Gamma_3^{\eta_{\mathcal{S}}} := \Gamma_3 \cap \left\{(\nu,b,\alpha,\eta_{\mathcal{S}}) \text{ s.t } \left|\nu\left\|w^*\right\|_1 - \frac{\widetilde{M}}{3}\right| \leq \frac{\tilde{\lambda}\widetilde{M}}{t_{m_n}^{2/3}} \quad \text{and} \quad \alpha \leq \alpha_{\overline{m}_*} \quad \text{and} \quad \alpha \geq \alpha_{\underline{m}_*}\right\}$. In this step we keep the term $\frac{\eta_{\mathcal{S}}^2}{\nu}$ from the first constraint of $\Gamma_3$, and repeat the same steps leading to Equation (30) to obtain constraint:

$$
\begin{aligned}
&\frac{\sqrt{2}}{3\sqrt{\pi}} \frac{n\left\|w^*\right\|_1 \alpha^2}{\widetilde{M}^3 \kappa(1-\kappa)^2 \left\|h\right\|_\infty^2}\left(1 - O\left(\frac{1}{t_{m_n}^4}\right)\right) + \sqrt{\frac{2}{\pi}}\eta_{\mathcal{S}}^2 \frac{\alpha^2 n\left\|w^*\right\|_1}{\left\|h\right\|_\infty^2 \widetilde{M}^3 \kappa(1-\kappa)^2}(1+\mathcal{O}_b)^{-1} \\
&\qquad\qquad + \sqrt{\frac{2}{\pi}}\left\|\gamma(\alpha)\right\|_2^2 \frac{n\left\|w^*\right\|_1}{\left\|h\right\|_\infty^2 \kappa\widetilde{M}}(1+\mathcal{O}_b)^{-1} \leq 1
\end{aligned}
$$

As in the Step 3.3 we have that $\kappa(1-\kappa)^2 \leq \frac{4}{27}$ and $\kappa \leq \frac{1}{3} + \frac{\tilde{\lambda}}{t_{m_n}^{2/3}}$. Plugging these two bounds into the inequality above, we further relax the constraint to:

$$
\begin{aligned}
\eta_{\mathcal{S}}^2 \frac{\alpha^2}{\left\|h\right\|_\infty^2} \frac{n\left\|w^*\right\|_1}{\widetilde{M}^3} &\lesssim 1 - 3\sqrt{\frac{2}{\pi}}\frac{\left\|\gamma(\alpha)\right\|_2^2}{\left\|h\right\|_\infty^2}\frac{n\left\|w^*\right\|_1}{\widetilde{M}}(1 - \frac{3\tilde{\lambda}}{t_{m_n}^{2/3}} + O\left(\frac{1}{t_{m_n}^{4/3}}\right)) \\
&\qquad - \frac{9\sqrt{2}}{4\sqrt{\pi}}\frac{\alpha^2}{\left\|h\right\|_\infty^2}\frac{n\left\|w^*\right\|_1}{\widetilde{M}^3}(1 - O\left(\frac{1}{t_{m_n}^4}\right))
\end{aligned}
$$

At the end we use derived bounds on $\alpha$ to upper bound $\eta_{\mathcal{S}}$ as follows:

$$
\begin{aligned}
\eta_{\mathcal{S}}^2 \lesssim \frac{\widetilde{M}^3 \left\|h\right\|_\infty^2}{\alpha_{\underline{m}_*}^2 n\left\|w^*\right\|_1}\Bigg[&1 - 3\sqrt{\frac{2}{\pi}}\frac{\left\|\gamma(\alpha_{\overline{m}_*})\right\|_2^2}{\left\|h\right\|_\infty^2}\frac{n\left\|w^*\right\|_1}{\widetilde{M}}(1 - \frac{3\tilde{\lambda}}{t_{m_n}^{2/3}} + O\left(\frac{1}{t_{m_n}^{4/3}}\right)) \\
&- \frac{9\sqrt{2}}{4\sqrt{\pi}}\frac{\alpha_{\underline{m}_*}^2}{\left\|h\right\|_\infty^2}\frac{n\left\|w^*\right\|_1}{\widetilde{M}^3}(1 - O\left(\frac{1}{t_{m_n}^4}\right))\Bigg]
\end{aligned}
$$

Finally, after application of concentration Proposition 3 and definitions of $\alpha_{\underline{m}_*}, \alpha_{\overline{m}_*}$ and $\widetilde{M}$ we obtain $\eta_{\mathcal{S}}^2 \lesssim \frac{1}{t_{m_n}^{7/3}}$, which finishes the proof of this proposition.

# C   Proof of Theorem 2

In this section we present the proof of Theorem 2. We begin by recalling some definitions: $f_n(\nu,\eta) = \frac{1}{n}\sum_{i=1}^n (1 - \xi_i\nu|z_i^{(0)}| - z_i^{(1)}\eta)_+^2$ and $f(\nu,\eta) = \mathbb{E}f_n(\nu,\eta) = \mathbb{E}\left(1 - \xi\nu|Z^{(1)}| - Z^{(2)}\eta\right)_+^2$ and $\nu_f := \arg\min f(\nu,0)$. Further, define $\zeta_f = f(\nu_f,0)$, $\zeta_{\eta\eta} = \frac{d^2}{d^2\eta}|_{(\nu_f,0)}f(\nu,\eta)$, $\zeta_{\nu\nu} = \frac{d^2}{d^2\nu}|_{(\nu_f,0)}f(\nu,\eta)$. which are all non-zero positive constants. We define the constant $\kappa_\sigma$ in Theorem 2 by:

$$
\kappa_\sigma = \frac{2\zeta_f}{\zeta_{\eta\eta}\nu_f^2\pi^2}. \tag{31}
$$

In a first localization step, we bound $\Phi_N$. By proposition 1, it suffices to the upper bound $\phi_N$, which by Proposition 2 can be reduced to a low-dimensional stochastic optimization problem. We show:

**Proposition 6.** *Let the assumptions of Theorem 2 hold. Let $t_{m_n}$ (as in Equation* (13) *in Appendix A.1) be such that $2\Phi^{\complement}(t_{m_n}) = m_n/d$ with $m_n = n\zeta_{\eta\eta}/2$. There exist universal positive constants $c_1, c_2, c_3 > 0$ such that*

$$(\phi_N)^2 \le \frac{n\zeta_f}{t_{m_n}^2}\left(1 - \frac{2}{t_{m_n}^2} + \frac{c_1}{t_{m_n}^3}\right) =: M^2$$

*with probability at least $1 - c_2\exp\left(-c_3\frac{n}{\log^5(d/n)}\right)$ over the draws of $h_1, h_2, z^{(1)}, z^{(2)}$ and $\xi$.*

The proof of the proposition is deferred to Appendix C.1. As described in Section 4.3, in a second uniform convergence step, we bound the constraint set $\Gamma$ from Equation (9):

**Proposition 7.** *Let the assumptions of Theorem 2 hold and let $\Gamma$ be as in Equation* (9) *with $M$ from Proposition 6. Define a set $\Gamma_0$ as a set of all $(\nu, b, \alpha, \eta_{\mathcal{S}})$ that satisfy:*

$$|\nu - \nu_f|^2 \lesssim \frac{1}{\log(d/m_n)} \quad \text{and} \quad \eta_{\mathcal{S}}^2 \lesssim \frac{1}{\log^{5/4}(d/m_n)}$$

$$\text{and} \quad \left|b^2\,\|\gamma(\alpha)\|_2^2 - \frac{2\zeta_f}{\zeta_{\eta\eta}\log(d/m_n)}\right| \lesssim \frac{1}{\log^{5/4}(d/m_n)}$$

*with $m_n = n\zeta_{\eta\eta}/2$. There exist universal constants $c_1, c_2, c_3, c_4 > 0$ such that $\Gamma \subset \Gamma_0$ with probability at least $1 - c_1\exp\left(-c_2\frac{n}{\log^5(d/n)}\right) - c_3\exp\left(-c_4\frac{n}{\log n \log^{3/2}(d/n)}\right)$ over the draws of $h_1, h_2, z^{(1)}, z^{(2)}$ and $\xi$.*

The proof of the proposition is deferred to Appendix C.2. As a consequence, when applying Proposition 2 we can upper and lower bound $\phi_+$ and $\phi_-$:

$$\phi_+ \le \left[1 + \frac{\min\limits_{(b,\alpha)\in\Gamma_0} b^2\,\|\gamma(\alpha)\|_2^2 + \min\limits_{\eta_{\mathcal{S}}\in\Gamma_0}\eta_{\mathcal{S}}^2}{\max\limits_{\nu\in\Gamma_0}\nu^2}\right]^{-1/2} \le 1 - \frac{\zeta_f}{\zeta_{\eta\eta}\nu_f^2}\frac{1}{\log(d/m_n)}\left(1 - \frac{c}{\log(d/m_n)^{1/4}}\right)$$

$$\phi_- \ge \left[1 + \frac{\max\limits_{(b,\alpha)\in\Gamma_0} b^2\,\|\gamma(\alpha)\|_2^2 + \max\limits_{\eta_{\mathcal{S}}\in\Gamma_0}\eta_{\mathcal{S}}^2}{\min\limits_{\nu\in\Gamma_0}\nu^2}\right]^{-1/2} \ge 1 - \frac{\zeta_f}{\zeta_{\eta\eta}\nu_f^2}\frac{1}{\log(d/m_n)}\left(1 + \frac{c}{\log(d/m_n)^{1/4}}\right)$$

Where we slightly abuse the notation by writing $(b, \alpha) \in \Gamma_0$ and similar for $\nu \in \Gamma_0$ and $\eta_{\mathcal{S}} \in \Gamma_0$. Finally, the proof follows when applying Proposition 1 and using the exact same series expansion for risk as in Equation (15).

### C.1 Proof of Localization Proposition 6

Recall the upper bound for $\phi_N$ from Proposition 2. Since $w^*$ is $s$-sparse vector, we have that $\|w^*\|_1 \le \sqrt{s}$, and we can further upper bound $\phi_N$ as follows:

$$\phi_N \le \min_{\nu,b,\alpha} |\nu|\sqrt{s} + b\,\|\gamma(\alpha)\|_1 \quad \text{s.t} \quad \frac{1}{n}b^2\,\|h\|_\infty^2 \ge f_n\left(\nu, b\,\|\gamma(\alpha)\|_2\right) \tag{32}$$

Given that $(\tilde{\nu}, \tilde{b}, \tilde{\alpha})$ is a feasible point for a given upper bound, we have $\phi_N \le |\tilde{\nu}|\sqrt{s} + \tilde{b}\,\|\gamma(\tilde{\alpha})\|_1$. Thus, in the following discussion, our goal is to find a single feasible point of the constraint set from Equation (32).

In order to show that a point satisfies the constraint above, it is necessary to evaluate the function $f_n(\nu, b\,\|\gamma(\alpha)\|_2)$ at this point. We do this by using the concentration of Lipschitz continuous function from Lemma 3. Namely, recall that we defined $f = \mathbb{E}[f_n]$ and thus according to Lemma 3 for any $\nu, \eta$ holds that:

$$\mathbb{P}\left(|f_n(\nu, \eta) - f(\nu, \eta)| \ge \epsilon\right) \le 2\exp\left(-c\frac{n\epsilon^2}{\nu^2 + \eta^2}\right) \tag{33}$$

with some universal constant $c > 0$. Therefore, with high probability we can approximate the evaluation of the function $f_n$ at a point by the evaluation of the function $f$ at the same point.

From definition of $\gamma(\alpha)$ we know that $\|\gamma(\alpha)\|_1 = \alpha$ and hence we can upper bound $\phi_N$ by an optimization problem over $\nu > 0$ and $b_\alpha := b\alpha$ as follows:

$$\phi_N \leq \min_{\nu, b_\alpha, \alpha} \nu\sqrt{s} + b_\alpha \ \ \text{s.t} \ \ \frac{1}{n}\frac{b_\alpha^2}{\alpha^2}\|h\|_\infty^2 \geq f_n\left(\nu, b_\alpha\frac{\|\gamma(\alpha)\|_2}{\alpha}\right) \tag{34}$$

Using Equation (33) with $\epsilon = \zeta_f t_{m_n}^{-3}$ and for a feasible point $(\nu, b_\alpha\frac{\|\gamma(\alpha)\|_2}{\alpha})$ we have that:

$$\frac{b_\alpha^2\|h\|_\infty^2}{n\alpha^2} \geq f\left(\nu, b_\alpha\frac{\|\gamma(\alpha)\|_2}{\alpha}\right) + \frac{\zeta_f}{t_{m_n}^3} \tag{35}$$

with probability at least $1 - 2\exp\left(-c\frac{n}{t_{m_n}^6\left(\nu^2 + b_\alpha^2\|\gamma(\alpha)\|_2^2/\alpha^2\right)}\right)$.

Recall that we defined $\nu_f := \arg\min f(\nu, 0)$. Now, let us choose $\tilde{\nu} = \nu_f$ and show that there exists a pair $(b, \alpha)$ such that $(\nu_f, b, \alpha)$ is feasible for constraint (35). We propose to search for a point with parameter $(b, \alpha)$ such that $b\|\gamma(\alpha)\|_2 = b_\alpha\frac{\|\gamma(\alpha)\|_2}{\alpha}$ is close to zero. We show in Lemma 8 that $f$ is infinitely differentiable function and thus, using Taylor series approximation of the function $f(\nu_f, \cdot) : \eta \mapsto f(\nu_f, \eta)$ around the point $(\nu_f, 0)$ we can rewrite the constraint (35) as:

$$\frac{b_\alpha^2\|h\|_\infty^2}{n\alpha^2} \geq \zeta_f + \frac{1}{2}\zeta_{\eta\eta}b_\alpha^2\frac{\|\gamma(\alpha)\|_2^2}{\alpha^2} + O\left(b_\alpha^3\frac{\|\gamma(\alpha)\|_2^3}{\alpha^3}\right) + \frac{\zeta_f}{t_{m_n}^3} \tag{36}$$

with $\zeta_\eta := \left.\frac{\partial f(\nu_f, \eta)}{\partial \eta}\right|_{\eta=0} = 0$ and where we recall that by definition $\zeta_f = f(\nu_f, 0)$, $\zeta_{\eta\eta} = \left.\frac{\partial^2 f(\nu_f, \eta)}{\partial \eta^2}\right|_{\eta=0}$ and $m_n = \frac{1}{2}\zeta_{\eta\eta}n$.

As we mentioned in Section A.1, $\gamma(\alpha)$ is a piecewise linear function with break points at $\alpha_m$ for $m = 2, \ldots, d$. Therefore, instead of optimizing over $\alpha$, we optimize over $m$. Rearranging the terms from Equation (36) we get:

$$b_\alpha^2 \geq \frac{n\alpha_m^2}{\|h\|_\infty^2}\frac{\zeta_f\left(1 + \frac{1}{t_{m_n}^3}\right)}{1 - \frac{1}{2}n\zeta_{\eta\eta}\frac{\|\gamma(\alpha_m)\|_2^2}{\|h\|_\infty^2} - O\left(b_\alpha n\frac{\|\gamma(\alpha_m)\|_2^3}{\alpha_m\|h\|_\infty^2}\right)} \tag{37}$$

Note that we have only one constraint but two free variables $(b, \alpha)$ and so we can set $\tilde{\alpha} = \alpha_{m_n}$ with $m_n = \frac{1}{2}\zeta_{\eta\eta}n$. To gain an intuition for why this choice is approximately optimal, one can follow a similar argument as in Remark 1 in Wang et al. (2022) and show that $m_n$ approximately maximizes expression:

$$\frac{\|h\|_\infty^2}{\alpha_m^2}\left(1 - \frac{1}{2}n\zeta_{\eta\eta}\frac{\|\gamma(\alpha_m)\|_2^2}{\|h\|_\infty^2} - O\left(b_\alpha n\frac{\|\gamma(\alpha_m)\|_2^3}{\alpha_m\|h\|_\infty^2}\right)\right)$$

Thus, $m_n$ approximately minimizes expression on the right hand side of Equation (37) and maximally relaxes this constraint on $b_\alpha^2$. We now claim that

$$\tilde{b}_\alpha^2 = \frac{n\alpha_{m_n}^2}{\|h\|_\infty^2}\frac{\zeta_f\left(1 + \frac{1}{t_{m_n}^3}\right)}{1 - \frac{1}{2}n\zeta_{\eta\eta}\frac{\|\gamma(\alpha_{m_n})\|_2^2}{\|h\|_\infty^2} - O\left(\frac{1}{t_{m_n}^3}\right)}$$

satisfies inequality (37) with probability at least $1 - 6\exp\left(-\frac{2m_n}{\log^5(d/m_n)}\right)$. Using Proposition 3 we have with high probability that:

$$1 - \frac{1}{2}n\zeta_{\eta\eta}\frac{\|\gamma(\alpha_{m_n})\|_2^2}{\|h\|_\infty^2} - O\left(\frac{1}{t_{m_n}^3}\right) > 1 - \frac{1}{2}n\zeta_{\eta\eta}\frac{2}{m_n t_{m_n}^2} - O\left(\frac{1}{t_{m_n}^3}\right) = 1 - \frac{2}{t_{m_n}^2} - O\left(\frac{1}{t_{m_n}^3}\right) > 0$$

for $d, n$ sufficiently large. Applying Proposition 3 once again we can upper bound $\tilde{b}_\alpha$:

$$\tilde{b}_\alpha^2 \leq \frac{n\zeta_f}{t_{m_n}^2}\left(1 + \frac{1}{t_{m_n}^3}\right)\left(1 - \frac{4}{t_{m_n}^2} + \frac{c}{t_{m_n}^4}\right)\frac{1}{1 - \frac{2}{t_{m_n}^2} - O\left(\frac{1}{t_{m_n}^3}\right)} \leq \frac{n\zeta_f}{t_{m_n}^2}\left(1 - \frac{2}{t_{m_n}^2} + \frac{c}{t_{m_n}^3}\right)$$

Now applying Proposition 3 we see that $O\left(\tilde{b}_\alpha n \frac{\|\gamma(\alpha_{m_n})\|_2^3}{\alpha_{m_n}\|h\|_\infty^2}\right) = O\left(\frac{\sqrt{n}}{t_{m_n}}n\frac{1}{m_n\sqrt{m_n}t_{m_n}^2}\right) = O\left(\frac{1}{t_{m_n}^3}\right)$ and $\tilde{b}_\alpha^2$ indeed satisfy Equation (37). From the upper bound of the sparsity, we have $\nu_f\sqrt{s} \lesssim \frac{\sqrt{n}}{t_{m_n}^4}$. Since $(\tilde{\nu}, \tilde{b}, \tilde{\alpha})$ is a feasible point, from Equation (34) and derived bounds on $\nu_f\sqrt{s}$ and $\tilde{b}_\alpha$ follows that

$$M := \sqrt{\frac{n\zeta_f}{t_{m_n}^2}\left(1 - \frac{2}{t_{m_n}^2} + \frac{\tilde{c}}{t_{m_n}^3}\right)}$$

is an upper bound on $\phi_N$ with probability at least $1 - 2\exp\left(-c\frac{n}{\log^3(d/n)\left(\nu_f^2 + b_\alpha^2\|\gamma(\alpha_{m_n})\|_2^2/\alpha_{m_n}^2\right)}\right) - 6\exp\left(-c\frac{n}{\log^5(d/n)}\right)$. The proposition is proved after noting that $\nu_f^2 + \tilde{b}_\alpha^2\|\gamma(\alpha_{m_n})\|_2^2/\alpha_{m_n}^2 = O(1)$.

## C.2 Proof of Uniform Convergence Proposition 7

The proof of the proposition follows from several steps where in each step we approximate $f_n$ using the bounds on $(\nu, \eta_{\mathcal{S}^c}, \eta_{\mathcal{S}})$ from the previous steps to obtain a tighter bound on $(\nu, \eta_{\mathcal{S}^c}, \eta_{\mathcal{S}})$ using the tools developed in Wang et al. (2022). The probability statement in Proposition 7 follows when taking the union bound over all equations which we condition on throughout the proof.

Furthermore, we note that the set $\Gamma$ from Proposition 2 is not empty as clearly the choice $(\tilde{\nu}, \tilde{b}, \tilde{\alpha}, 0)$ from Section C.1 leads with high probability to a feasible point due to the choice of $M$. Moreover, we can even relax set $\Gamma$ from Proposition 2 and bound the variables that are elements of the following set:

$$\left\{(\nu, b, \alpha, \eta_{\mathcal{S}}) \text{ s.t } \frac{1}{n}(2\sqrt{s_{\max}}\eta_{\mathcal{S}} + b\|h\|_\infty)^2 \geq f_n(\nu, \sqrt{b^2\|\gamma(\alpha)\|_2^2 + \eta_{\mathcal{S}}^2}) \text{ and } b\alpha \leq M\right\} \supset \Gamma. \quad (38)$$

where we implicitly assume bounds $\eta_{\mathcal{S}} \geq 0, b \geq 0, \alpha \in [1, \alpha_{\max}]$ in all of the following discussion. The inclusion of $\Gamma$ in the above set holds, since any point satisfying $\max\left\{|\nu|\|w_*^{(\mathcal{S})}\|_1 - \sqrt{s}\eta_{\mathcal{S}}, 0\right\} + b\alpha \leq M$ satisfies $b\alpha \leq M$ as well. In what follows, we bound the variables of interest from Proposition 7 if they are elements of the above given set, which, by inclusion, implies high probability bounds of the same variables in the set $\Gamma$.

**Bound 1:** $\nu^2, \eta_{\mathcal{S}^c}^2, \eta_{\mathcal{S}}^2 = O(1)$

In order to apply Lemma 7 in the next step, which gives tight bounds for $f_n$, we first need to show that, with high probability, $\nu^2, \eta^2, \eta_{\mathcal{S}}^2 = O(1)$. This is the goal of this first step. More specifically, the goal of this first step is to show that there exist universal constants $B_{\nu,1}, B_{\eta_{\mathcal{S}^c},1}, B_{\eta_{\mathcal{S}},1} > 0$ such that for any element $(\nu, b, \alpha, \eta_{\mathcal{S}})$ of $\Gamma_0$ we have $\nu^2 \leq B_{\nu,1}^2$, $\eta_{\mathcal{S}^c} = b\|\gamma(\alpha)\|_2 \leq B_{\eta_{\mathcal{S}^c},1}$ and $\eta_{\mathcal{S}} \leq B_{\eta_{\mathcal{S}},1}$ with high probability over the draws of $h_1, h_2, z^{(1)}, z^{(2)}$ and $\xi$.

For this first step, we use the fact that in the presence of label noise, $f_n$ is lower bounded by a quadratic function as stated in Lemma 6 i.e. we have that

$$f_n(\nu, \sqrt{b^2\|\gamma(\alpha)\|_2^2 + \eta_{\mathcal{S}}^2}) \geq c_\nu\nu^2 + c_\eta(b^2\|\gamma(\alpha)\|_2^2 + \eta_{\mathcal{S}}^2) \geq c_\eta\eta_{\mathcal{S}}^2$$

holds with probability $\geq 1 - \exp(-cn)$. As a result, we can relax the first constraint in Definition (38) of $\Gamma$ to

$$\frac{1}{n}(2\sqrt{s_{\max}}\eta_{\mathcal{S}} + b\|h\|_\infty)^2 \geq c_\nu\nu^2 + c_\eta b^2\|\gamma(\alpha)\|_2^2 + c_\eta\eta_{\mathcal{S}}^2 \quad (39)$$

This implies that $c_\eta \eta_{\mathcal{S}}^2 \leq \frac{1}{n}(2\sqrt{s_{\max}}\eta_{\mathcal{S}} + b\|h\|_\infty)^2 \leq \frac{8}{n}s_{\max}\eta_{\mathcal{S}}^2 + \frac{2}{n}b^2\|h\|_\infty^2$. Thus for some universal constants $c_1, c_2 > 0$ we have

$$\eta_{\mathcal{S}}^2 \leq \frac{2}{c_\eta n}b^2\|h\|_\infty^2\left(1 - \frac{8}{c_\eta n}s_{\max}\right)^{-1} \leq \frac{2}{c_\eta n}b^2\|h\|_\infty^2\left(1 + \frac{c_1}{t_{m_n}^8}\right) \leq \frac{c_2}{n}b^2\|h\|_\infty^2$$

where we used that $s_{\max} = \Theta\left(\frac{n}{t_{m_n}^8}\right)$. Now define universal constant $c > 0$ as the smallest constant satisfying

$$\frac{1}{n}(2\sqrt{s_{\max}}\eta_{\mathcal{S}} + b\|h\|_\infty)^2 \leq \frac{2}{n}b^2\|h\|_\infty^2\left(1 + \frac{4c_2}{n}s_{\max}\right) \leq \frac{c}{n}b^2\|h\|_\infty^2 \tag{40}$$

Combining Equations (39) and (40) we can relax the first constraint of $\Gamma$ to

$$\frac{c}{n}b^2\|h\|_\infty^2 \geq c_\nu\nu^2 + c_\eta b^2\|\gamma(\alpha)\|_2^2 + c_\eta\eta_{\mathcal{S}}^2.$$

This approximation leads to an optimization problem similar to the one discussed in Lemma 1 in Wang et al. (2022). After further relaxations we obtain exactly the same form of the inequality, and hence we can use the arguments from Wang et al. (2022). Define the following set:

$$\Gamma_1 = \left\{(\nu, b, \alpha, \eta_{\mathcal{S}}) \ \text{ s.t } \ \frac{c}{n}b^2\|h\|_\infty^2 \geq c_\nu\nu^2 + c_\eta b^2\|\gamma(\alpha)\|_2^2 + c_\eta\eta_{\mathcal{S}}^2 \ \text{ and } \ b\alpha \leq M\right\}$$

It is evident from the previous discussion that $\Gamma \subset \Gamma_1$ with high probability. Thus, deriving high-probability bounds on $\Gamma_1$ gives valid bounds for $\Gamma$ as well. In the following three steps, we bound variables $\eta_{\mathcal{S}^c}, \nu, \eta_{\mathcal{S}}$ from the set $\Gamma_1$, respectively.

**Step 1.1: Upper bound on $\eta_{\mathcal{S}^c}$.** In this step, as well as in almost every step that follows, we use the fact that, by relaxing constraints from the definition of the set $\Gamma_1$ and bounding the variables on this larger set, we obtain valid bounds for the variables in $\Gamma_1$ and, more specifically, in $\Gamma$. Moreover, recall that by our reparametrization from Section 4.2 we have $\eta_{\mathcal{S}^c}^2 = \|w_\perp^{(\mathcal{S}^c)}\|_2^2 = b^2\|\gamma(\alpha)\|_2^2$. Hence, we relax the first constraint in definition of $\Gamma_1$ to show that:

$$\eta_{\mathcal{S}^c}^2 \leq \max_{(\nu, b, \alpha, \eta_{\mathcal{S}}) \in \Gamma_1} b^2\|\gamma(\alpha)\|_2^2 \leq \max_{b, \alpha}\left[b^2\|\gamma(\alpha)\|_2^2 \ \text{ s.t } \ \frac{c}{n}b^2\|h\|_\infty^2 \geq c_\eta b^2\|\gamma(\alpha)\|_2^2 \ \text{ and } \ b\alpha \leq M\right]$$

$$= \max_{1 \leq \alpha \leq \alpha_{\max}}\left[M^2\frac{\|\gamma(\alpha)\|_2^2}{\alpha^2} \ \text{ s.t } \ \frac{c}{n}\|h\|_\infty^2 \geq c_\eta\|\gamma(\alpha)\|_2^2\right] \tag{41}$$

Now note that as discussed in Section A.1 $\|\gamma(\alpha)\|_2^2$ is convex. Therefore, the set of feasible $\alpha$ that satisfy the last constraint is a nonempty interval. Indeed, to see that the interval is not empty, recall that we defined $M$ in such a way that $(b, \alpha_{m_n}) \in \Gamma$ with high probability for $b\alpha_{m_n} \leq M$. As $\Gamma \subset \Gamma_1 \subset \{\alpha \ \text{ s.t } \ \frac{c}{n}\|h\|_\infty^2 \geq c_\eta\|\gamma(\alpha)\|_2^2\}$, with high probability $\alpha_{m_n}$ satisfies the constraint in Equation (41). Furthermore, since $\frac{\|\gamma(\alpha)\|_2^2}{\alpha^2}$ is monotonically decreasing, to upper bound Equation (41) it is sufficient to find $\underline{\alpha} < \alpha_{m_n}$ such that the constraint from Equation (41) does not hold, i.e. we should have:

$$\frac{\|\gamma(\underline{\alpha})\|_2^2}{\|h\|_\infty^2} > \frac{c}{c_\eta n}. \tag{42}$$

It is sufficient to only consider the discretized version of $\alpha$, i.e., $\alpha_m$, for which we have access to the tight concentration inequalities from Proposition 3. We now claim that $\alpha_{\underline{m}}$ with $\underline{m} = \lambda_{\underline{m}}\frac{n}{\log(d/n)}$ satisfies the inequality (42) for some positive universal constant $\lambda_{\underline{m}} > 0$. Using the characterization $t_m^2 = 2\log(d/m) + O(\log\log(d/m))$ and concentration inequalities from Section A.1 we show that $\underline{m}$ satisfies Equation (42) since

$$\frac{2}{\underline{m}t_{\underline{m}}^2}\left(1 - O\left(\frac{1}{t_{\underline{m}}^2}\right)\right) > \frac{1}{n\lambda_{\underline{m}}}\left(1 - O\left(\frac{\log\log(d/n)}{\log(d/n)}\right)\right) > \frac{c}{c_\eta n},$$

where last inequality holds for $d/n$ sufficiently large and $\lambda_{\underline{m}}$ small enough.

Therefore, from Equation (41) and the concentration inequality from Proposition 3, we get:

$$\eta_{\mathcal{S}^c}^2 \le M^2 \frac{\|\gamma(\alpha_{\underline{m}})\|_2^2}{\alpha_{\underline{m}}^2} \le \frac{n\zeta_f}{t_{m_n}^2} \frac{2}{\underline{m}} \left(1 + O\left(\frac{1}{t_{m_n}^2}\right)\right) =: B_{\eta_{\mathcal{S}^c},1}^2,$$

with $B_{\eta_{\mathcal{S}^c},1} = \Theta(1)$, as desired.

**Step 1.2: Upper bound on $\nu$.** Similarly as in the previous step, we first relax the first constraint from definition of $\Gamma_1$ and use obtained constraints to upper bound $\nu^2$ as follows:

$$\nu^2 \le \max_{(\nu,b,\alpha,\eta_{\mathcal{S}}) \in \Gamma_1} \nu^2$$

$$\le \max_{\nu,b,\alpha,\eta_{\mathcal{S}}} \left[\nu^2 \ \text{ s.t } \ \frac{c}{n}b^2 \|h\|_\infty^2 \ge c_\nu \nu^2 \ \text{ and } \ \frac{c}{n}b^2 \|h\|_\infty^2 \ge c_\eta b^2 \|\gamma(\alpha)\|_2^2 \ \text{ and } \ b\alpha \le M\right]$$

$$= \frac{c}{nc_\nu} \|h\|_\infty^2 \max_{b,\alpha} \left[b^2 \ \text{ s.t } \ \frac{c}{n}\|h\|_\infty^2 \ge c_\eta \|\gamma(\alpha)\|_2^2 \ \text{ and } \ b\alpha \le M\right]$$

$$= \frac{c}{nc_\nu} M^2 \|h\|_\infty^2 \min_{1 \le \alpha \le \alpha_{\max}} \left[\frac{1}{\alpha^2} \ \text{ s.t } \ \frac{c}{n}\|h\|_\infty^2 \ge c_\eta \|\gamma(\alpha)\|_2^2\right] \tag{43}$$

Since $\frac{1}{\alpha^2}$ is a monotonically decreasing function, we can use exactly the same reasoning as in the Step 1.1 to obtain a high probability upper bound $\frac{1}{\alpha^2} \le \frac{1}{\alpha_{\underline{m}}^2}$. Hence, using Equation (43) and the concentration results from Proposition 3 we upper bound $\nu$ as follows:

$$\nu^2 \le \frac{c}{nc_\nu} M^2 \frac{\|h\|_\infty^2}{\alpha_{\underline{m}}^2} \le \frac{c\zeta_f}{c_\nu} \frac{t_{\underline{m}}^2}{t_{m_n}^2} \left(1 + O\left(\frac{1}{t_{m_n}^2}\right)\right) =: B_{\nu,1}^2,$$

and in particular, after using the characterization $t_m^2 = 2\log(d/m) + O(\log\log(d/m))$ from Section A.1, we have again that $B_{\nu,1} = \Theta(1)$.

**Step 1.3: Upper bound on $\eta_{\mathcal{S}}$.** Replacing $\nu$ by $\eta_{\mathcal{S}}$ and applying exactly the same procedure as in the Step 1.2, we obtain that with high probability:

$$\eta_{\mathcal{S}}^2 \le \frac{c}{nc_\eta} M^2 \frac{\|h\|_\infty^2}{\alpha_{\underline{m}}^2} \le \frac{c\zeta_f}{c_\eta} \frac{t_{\underline{m}}^2}{t_{m_n}^2} \left(1 + O\left(\frac{1}{t_{m_n}^2}\right)\right) =: B_{\eta_{\mathcal{S}},1}^2,$$

for $B_{\eta_{\mathcal{S}},1} = \Theta(1)$, which completes the first part of the proof.

**Bound 2:** $\triangle\nu^2, \eta_{\mathcal{S}^c}^2, \eta_{\mathcal{S}}^2 = O\left(\frac{1}{\log(d/n)}\right)$

Recall that $\nu_f := \arg\min f(\nu, 0)$ and define $\triangle\nu = \nu - \nu_f$. Conditioning on the event where the bounds from the first step hold for $\nu, \eta_{\mathcal{S}^c}, \eta_{\mathcal{S}}$, the goal of this second step is to show that for any element $(\nu, b, \alpha, \eta_{\mathcal{S}})$ of $\Gamma$ we have $\triangle\nu^2 = O\left(\frac{1}{\log(d/n)}\right)$, $\eta_{\mathcal{S}^c}^2 = b^2 \|\gamma(\alpha)\|_2^2 = O\left(\frac{1}{\log(d/n)}\right)$ and $\eta_{\mathcal{S}}^2 = O\left(\frac{1}{\log(d/n)}\right)$ with high probability over the draws of $h_1, h_2, z^{(1)}, z^{(2)}$ and $\xi$.

From the previous step, we know that, with high probability, $\nu^2 \le B_\nu^2$, $\eta_{\mathcal{S}^c} \le B_{\eta_{\mathcal{S}^c},1}$ and $\eta_{\mathcal{S}} \le B_{\eta_{\mathcal{S}},1}$. Hence we can use Lemma 7 to obtain a tight lower bound for $f_n$, which is based on uniform convergence of $f_n$ to its expectation in Proposition 10, and relax the constraint from definition of the set $\Gamma$ as follows:

$$\frac{1}{n}(2\sqrt{s_{\max}}\eta_{\mathcal{S}} + b\|h\|_\infty)^2 \ge f_n(\nu, \sqrt{b^2\|\gamma(\alpha)\|_2^2 + \eta_{\mathcal{S}}^2}) \ge f(\nu, \sqrt{b^2\|\gamma(\alpha)\|_2^2 + \eta_{\mathcal{S}}^2}) - \mathcal{O}_c \tag{44}$$

$$\ge \zeta_f + \widetilde{c}_\nu \triangle\nu^2 + \widetilde{c}_\eta b^2 \|\gamma(\alpha)\|_2^2 + \widetilde{c}_\eta \eta_{\mathcal{S}}^2 - \mathcal{O}_c,$$

where we choose $\mathcal{O}_c = O\left(\frac{1}{t_{m_n}^3}\right)$ and hence the bound holds uniformly with probability at least $1 - \exp\left(-c_2 \frac{n}{t_{m_n}^6}\right) - \exp\left(-c_3 \frac{n}{t_{m_n}^3 \log n}\right)$.

Now we show how we can relax and simplify the LHS from Equation (44). Since $c_1 m_n \leq d$, we have, according to Equation (44) that $\frac{1}{n}(2\sqrt{s_{\max}}\eta_{\mathcal{S}} + b\|h\|_{\infty})^2 \geq \frac{1}{2}\zeta_f$. As before, we also have $\frac{1}{n}(2\sqrt{s_{\max}}\eta_{\mathcal{S}} + b\|h\|_{\infty})^2 \leq \frac{8s_{\max}}{n}\eta_{\mathcal{S}}^2 + \frac{2}{n}b^2\|h\|_{\infty}^2$. Combining last two expressions with the bound $\eta_{\mathcal{S}} \leq B_{\eta_{\mathcal{S}},1}$ from Step 1.3 we have:

$$\frac{1}{n}b^2\|h\|_{\infty}^2 \geq \frac{1}{4}\zeta_f - \frac{4s_{\max}}{n}B_{\eta_{\mathcal{S}},1}^2 \geq \frac{1}{8}\zeta_f$$

for $n, d$ large enough since $s_{\max} = \Theta\left(\frac{n}{t_{m_n}^8}\right)$. Thus we have:

$$\frac{1}{n}(2\sqrt{s_{\max}}\eta_{\mathcal{S}} + b\|h\|_{\infty})^2 = \frac{1}{n}b^2\|h\|_{\infty}^2\left(1 + \frac{2\sqrt{s_{\max}}\eta_{\mathcal{S}}}{b\|h\|_{\infty}}\right)^2$$

$$\leq \frac{1}{n}b^2\|h\|_{\infty}^2\left(1 + 2B_{\eta_{\mathcal{S}},1}\sqrt{\frac{8}{\zeta_f}}\sqrt{\frac{s_{\max}}{n}}\right)^2$$

and $\frac{1}{n}(2\sqrt{s_{\max}}\eta_{\mathcal{S}} + b\|h\|_{\infty})^2 \leq \frac{1}{n}b^2\|h\|_{\infty}^2\left(1 + c\sqrt{\frac{s_{\max}}{n}}\right)$ for a large enough constant $c > 0$. Furthermore, define $\mathcal{O}_b = c\sqrt{\frac{s_{\max}}{n}} = \Theta\left(\frac{1}{t_{m_n}^4}\right)$.

Motivated by Equation (44) and discussion after it, we define the following set:

$$\Gamma_2 = \left\{(\nu, b, \alpha, \eta_{\mathcal{S}}) \text{ s.t } \frac{1}{n}b^2\|h\|_{\infty}^2(1 + \mathcal{O}_b) \geq \zeta_f + \widetilde{c}_\nu \triangle \nu^2 + \widetilde{c}_\eta b^2\|\gamma(\alpha)\|_2^2 + \widetilde{c}_\eta \eta_{\mathcal{S}}^2 - \mathcal{O}_c \right.$$

$$\left. \text{and } b\alpha \leq M\right\}$$

Again, from the discussion in this section, we have that with high probability $\Gamma \subset \Gamma_2$. Similarly as in the previous bound, we will bound variables of interest i.e. $\eta_{\mathcal{S}^c}, \nu, \eta_{\mathcal{S}}$ in the set $\Gamma_2$ and use the inclusion of the set $\Gamma$ in $\Gamma_2$ to claim that these bounds are valid even in $\Gamma$.

**Step 2.1: Upper bound on $\eta_{\mathcal{S}^c}$.** Similarly to the Equation (41) in Step 1.1, we relax constraints of $\Gamma_2$ to obtain:

$$\eta_{\mathcal{S}^c}^2 \leq \max_{(\nu, b, \alpha, \eta_{\mathcal{S}}) \in \Gamma_2} b^2\|\gamma(\alpha)\|_2^2$$

$$\leq \max_{b,\alpha}\left[b^2\|\gamma(\alpha)\|_2^2 \text{ s.t } \frac{1}{n}b^2\|h\|_{\infty}^2(1 + \mathcal{O}_b) \geq \zeta_f + \widetilde{c}_\eta b^2\|\gamma(\alpha)\|_2^2 - \mathcal{O}_c \text{ and } b\alpha \leq M\right]$$

$$\leq \max_{b,\alpha}\left[b^2\|\gamma(\alpha)\|_2^2 \text{ s.t } b^2 \geq (\zeta_f - \mathcal{O}_c)\left(\frac{1}{n}\|h\|_{\infty}^2(1 + \mathcal{O}_b) - \widetilde{c}_\eta\|\gamma(\alpha)\|_2^2\right)^{-1} \text{ and } b \leq \frac{M}{\alpha}\right]$$

$$= \max_\alpha\left[\frac{M^2}{\alpha^2}\|\gamma(\alpha)\|_2^2 \text{ s.t } \frac{1}{n}\frac{M^2}{\alpha^2}\|h\|_{\infty}^2(1 + \mathcal{O}_b) \geq \zeta_f + \widetilde{c}_\eta\frac{M^2}{\alpha^2}\|\gamma(\alpha)\|_2^2 - \mathcal{O}_c\right]. \quad (45)$$

Multiplying the constraint on both sides with $\alpha^2$ and using the fact that $\|\gamma(\alpha)\|_2^2$ is convex shows that the set of feasible $\alpha$ is again a (non-empty) interval. Thus, by the monotonicity of $\frac{\|\gamma(\alpha)\|_2^2}{\alpha^2}$ the problem reduces again to finding $\alpha_{\underline{m}} < \alpha_{m_n}$ (where we use again that $\alpha_{m_n}$ satisfies the constraints with high probability) such that $\alpha_{\underline{m}}$ violates the constraint in Equation (45), i.e.,

$$\frac{\zeta_f - \mathcal{O}_c}{1 + \mathcal{O}_b}\frac{n\alpha_{\underline{m}}^2}{M^2\|h\|_{\infty}^2} + \frac{\widetilde{c}_\eta}{1 + \mathcal{O}_b}n\frac{\|\gamma(\alpha_{\underline{m}})\|_2^2}{\|h\|_{\infty}^2} > 1 \quad (46)$$

We now show that we can choose $\underline{m} = \lambda_{\underline{m}}m_n$ with a universal constant $\lambda_{\underline{m}} \in (0,1)$. Indeed, applying Proposition 3 and using the characterization $t_m^2 = 2\log(d/m) - \log\log(d/m) - \log(\pi) + \frac{\log\log(d/m)}{2\log(d/m)} + O\left(\frac{1}{\log(d/m)}\right)$

from Section A.1 we get:

$$\frac{\zeta_f - \mathcal{O}_c}{1 + \mathcal{O}_b} \frac{n\alpha_{\underline{m}}^2}{M^2 \|h\|_\infty^2} = 1 + \frac{2\log \lambda_{\underline{m}} - 2}{t_{m_n}^2} + O\left(\frac{1}{t_{m_n}^3}\right)$$

$$\text{and} \quad \frac{\widetilde{c}_\eta}{1 + \mathcal{O}_b} n \frac{\|\gamma(\alpha_{\underline{m}})\|_2^2}{\|h\|_\infty^2} = \frac{1}{t_{m_n}^2} \frac{4\widetilde{c}_\eta}{\zeta_{\eta\eta}\lambda_{\underline{m}}} + O\left(\frac{1}{t_{m_n}^4}\right)$$

where $O(.)$ has hidden dependencies on $\lambda_{\underline{m}}$. Hence, it is straight forward to see that for any $d \geq cn$ with universal constant $c > 0$ (and thus $t_{m_n}$ lower bounded), we can find a universal constant $\lambda_{\underline{m}}$ such that Equation (46) holds.

Hence, we can upper bound $\eta_{\mathcal{S}^c}^2$ in Equation (45) as follows:

$$\eta_{\mathcal{S}^c}^2 \leq M^2 \frac{\|\gamma(\alpha_{\underline{m}})\|_2^2}{\alpha_{\underline{m}}^2} \leq \frac{n\zeta_f}{t_{m_n}^2} \frac{2}{\underline{m}} \left(1 + O\left(\frac{1}{t_{\underline{m}}^2}\right)\right) \leq \frac{2\zeta_f}{\zeta_{\eta\eta}\lambda_{\underline{m}}\log(d/n)} \left(1 + O\left(\frac{1}{\log(d/n)}\right)\right) =: \frac{B_{\eta_{\mathcal{S}^c},2}^2}{t_{m_n}^2}$$

with $B_{\eta_{\mathcal{S}^c},2}^2 = \Theta(1)$.

**Step 2.2: Upper bound on $\triangle\nu$.** Instead of directly bounding $\nu$, here we upper bound $\triangle\nu^2$ with $\nu = \nu_f + \triangle\nu$ and thus obtain both an upper and a lower bound for $\nu$. Similarly as before, we have:

$$\triangle\nu^2 \leq \max_{(\nu,b,\alpha,\eta_{\mathcal{S}})\in\Gamma_2} \triangle\nu^2 \leq \max_{\nu,b,\alpha} \left[\triangle\nu^2 \text{ s.t } \frac{1}{n}b^2 \|h\|_\infty^2 (1 + \mathcal{O}_b) \geq \zeta_f + \widetilde{c}_\nu\triangle\nu^2 - \mathcal{O}_c \right.$$

$$\left. \text{and} \quad \frac{1}{n}b^2 \|h\|_\infty^2 (1 + \mathcal{O}_b) \geq \zeta_f + \widetilde{c}_\eta b^2 \|\gamma(\alpha)\|_2^2 - \mathcal{O}_c \quad \text{and} \quad b\alpha \leq M\right]$$

$$= \max_{b,\alpha} \left[\frac{1}{\widetilde{c}_\nu} \left(\frac{1}{n}b^2 \|h\|_\infty^2 (1 + \mathcal{O}_b) - \zeta_f + \mathcal{O}_c\right)\right.$$

$$\left. \text{s.t} \quad \frac{1}{n}b^2 \|h\|_\infty^2 (1 + \mathcal{O}_b) \geq \zeta_f + \widetilde{c}_\eta b^2 \|\gamma(\alpha)\|_2^2 - \mathcal{O}_c \quad \text{and} \quad b\alpha \leq M\right]$$

$$= \max_\alpha \left[\frac{1}{\widetilde{c}_\nu} \left(\frac{1}{n}\frac{M^2}{\alpha^2} \|h\|_\infty^2 (1 + \mathcal{O}_b) - \zeta_f + \mathcal{O}_c\right)\right.$$

$$\left. \text{s.t} \quad \frac{1}{n}\frac{M^2}{\alpha^2} \|h\|_\infty^2 (1 + \mathcal{O}_b) \geq \zeta_f + \widetilde{c}_\eta \frac{M^2}{\alpha^2} \|\gamma(\alpha)\|_2^2 - \mathcal{O}_c\right] \tag{47}$$

As in Step 1.2 we use that $\frac{1}{\alpha^2}$ is a monotonically decreasing function and the fact that $\alpha_{\underline{m}}$ from the previous step, with $\underline{m} = \lambda_{\underline{m}}m_n$ and $\alpha_{\underline{m}} \leq \alpha_{m_n}$, does not satisfy the constraint in Equation (47). Thus we can upper bound $\triangle\nu^2$ as follows:

$$\triangle\nu^2 \leq \frac{1}{n\widetilde{c}_\nu} \frac{M^2}{\alpha_{\underline{m}}^2} \|h\|_\infty^2 (1 + \mathcal{O}_b) - \frac{\zeta_f}{\widetilde{c}_\nu} + \frac{\mathcal{O}_c}{\widetilde{c}_\nu} \leq \frac{\zeta_f t_{\underline{m}}^2}{t_{m_n}^2 \widetilde{c}_\nu} \left(1 + \frac{2}{t_{m_n}^2}\right) - \frac{\zeta_f}{\widetilde{c}_\nu} + O\left(\frac{1}{t_{m_n}^3}\right)$$

$$= \frac{\zeta_f(2 - 2\log(\lambda_{\underline{m}}))}{\widetilde{c}_\nu 2\log(d/n)} \left(1 + O\left(\frac{1}{\log(d/n)}\right)\right) =: \frac{B_{\triangle\nu,2}^2}{t_{m_n}^2}$$

for some $B_{\triangle\nu,2}^2 = \Theta(1)$.

**Step 2.3: Upper bound on $\eta_{\mathcal{S}}$.** Following the same steps as in Step 2.2 with $\nu$ replaced by $\eta_{\mathcal{S}}$ we can show that there exists universal constant $B_{\eta_{\mathcal{S}},2} = \Theta(1)$ such that:

$$\eta_{\mathcal{S}}^2 \leq \frac{1}{n\widetilde{c}_\eta} \frac{M^2}{\alpha_{\underline{m}}^2} \|h\|_\infty^2 (1 + \mathcal{O}_b) - \frac{\zeta_f}{\widetilde{c}_\eta} + \frac{\mathcal{O}_c}{\widetilde{c}_\eta} \leq \frac{\zeta_f(2 - 2\log(\lambda_{\underline{m}}))}{\widetilde{c}_\eta 2\log(d/n)} \left(1 + O\left(\frac{1}{\log(d/n)}\right)\right) =: \frac{B_{\eta_{\mathcal{S}},2}^2}{t_{m_n}^2}$$

## Bound 3: Proof of the proposition

We already know that $\nu$ is concentrated around $\nu_f$. However, to obtain a tight expression for the risk and also a valid lower bound, we need to obtain tighter bounds for $\eta_{\mathcal{S}^c}^2$ and $\eta_{\mathcal{S}}^2$ conditioning on the bounds of the previous step, leading to Proposition 7.

Note that $f$ is an infinitely differentiable function as we prove in Lemma 8. Thus, in this part of the proof we can use the Taylor series approximation of the function $f$ where we use the result from the last step to bound the higher-order terms involving $\triangle\nu, \eta_{\mathcal{S}^c}$ and $\eta_{\mathcal{S}}$. Similarly as in equation (44), we obtain from Proposition 10 and the second order Taylor series approximation of $f$ around the point $(\nu_f, 0)$ that with high probability,

$$\frac{1}{n}b^2\|h\|_\infty^2(1+\mathcal{O}_b) \geq \zeta_f + \frac{1}{2}\zeta_{\nu\nu}\triangle\nu^2 + \frac{1}{2}\zeta_{\eta\eta}b^2\|\gamma(\alpha)\|_2^2 + \frac{1}{2}\zeta_{\eta\eta}\eta_{\mathcal{S}}^2 - \mathcal{O}_c - \mathcal{O}_f$$

with $\mathcal{O}_f = O(\triangle\nu^3 + \eta_{\mathcal{S}^c}^3 + \eta_{\mathcal{S}}^3) = O\left(\frac{1}{t_{m_n}^3}\right)$ and $\mathcal{O}_c, \mathcal{O}_b = O\left(\frac{1}{t_{m_n}^3}\right)$.

**Step 3.1: Upper and lower bound on $\eta_{\mathcal{S}^c}$.** We proceed in the same manner as in the previous two steps. We relax the constraint in definition of $\Gamma$ and define the following set:

$$\Gamma_3^{\eta_{\mathcal{S}^c}} = \left\{(\nu, b, \alpha, \eta_{\mathcal{S}}) \text{ s.t } \frac{1}{n}b^2\|h\|_\infty^2(1+\mathcal{O}_b) \geq \zeta_f + \frac{1}{2}\zeta_{\eta\eta}b^2\|\gamma(\alpha)\|_2^2 - \mathcal{O}_c - \mathcal{O}_f \text{ and } b\alpha \leq M\right\}$$

Clearly, we have again with high probability that $\Gamma \subset \Gamma_3^{\eta_{\mathcal{S}^c}}$. The only difference between $\Gamma_3^{\eta_{\mathcal{S}^c}}$ and $\Gamma_2$ lies in the constant $\widetilde{c}_\eta$ which is replaced by the tighter constant $\zeta_{\eta\eta}/2$. However, this makes a big difference, as this allows us to choose $\underline{m} < m_n < \overline{m}$ much tighter. Similar to Equation (46) we again require that $m = \underline{m}, \overline{m}$ satisfies

$$\frac{\zeta_f - \mathcal{O}_c - \mathcal{O}_f}{1 + \mathcal{O}_b}\frac{\alpha_m^2}{\|h\|_\infty^2}\frac{n}{M^2} + \frac{\zeta_{\eta\eta}}{2(1+\mathcal{O}_b)}n\frac{\|\gamma(\alpha_m)\|_2^2}{\|h\|_\infty^2} > 1. \tag{48}$$

However, this expression allows us to choose $\underline{m}$ and $\overline{m}$ as in Lemma 2, with $\kappa := 1/2$, $m_* := m_n$ and parameter $\lambda > 0$. We only show it for $\underline{m}$ as the same argument holds for $\overline{m}$. Applying Proposition 3, the LHS from Equation (48) can be bounded by

$$\frac{\zeta_f - \mathcal{O}_c - \mathcal{O}_f}{1 + \mathcal{O}_b}\frac{\alpha_{\underline{m}}^2}{\|h\|_\infty^2}\frac{n}{M^2} = \frac{t_{m_n}^2}{t_{\underline{m}}^2}\left(1 - \frac{4}{t_{\underline{m}}^2} + \frac{2}{t_{m_n}^2} + O\left(\frac{1}{t_{m_n}^3}\right)\right)$$

$$= 1 - \frac{\lambda}{t_{m_n}^{5/2}} - \frac{2}{t_{m_n}^2} + O\left(\frac{1}{t_{m_n}^3}\right) + O\left(\frac{1}{t_{m_n}^2 m_n}\right)$$

$$\text{and } \frac{\zeta_{\eta\eta}}{2(1+\mathcal{O}_b)}n\frac{\|\gamma(\alpha_{\underline{m}})\|_2^2}{\|h\|_\infty^2} = \frac{2}{t_{m_n}^2} + \frac{\lambda}{t_{m_n}^{5/2}} + \frac{\lambda^2}{4t_{m_n}^3} + O\left(\frac{1}{t_{m_n}^3}\right) + O\left(\frac{1}{t_{m_n}^2 m_n}\right)$$

with $O(.)$ having hidden dependencies on universal constant $\lambda$. In particular, as a result, we see that we can choose $\lambda$ such that Equation (48) holds for any $d > cn$ with universal constant $c > 0$. Hence we can upper bound $\eta_{\mathcal{S}^c}^2$ as follows:

$$\eta_{\mathcal{S}^c}^2 \leq M^2\frac{\|\gamma(\alpha_{\underline{m}})\|_2^2}{\alpha_{\underline{m}}^2} \leq \frac{n\zeta_f}{t_{m_n}^2}\frac{2}{\underline{m}}\left(1 + O\left(\frac{1}{t_{m_n}^2}\right)\right) \leq \frac{4\zeta_f}{\zeta_{\eta\eta}}\frac{1}{t_{m_n}^2}\left(1 + \frac{\lambda}{2\sqrt{t_{m_n}}} + O\left(\frac{1}{t_{m_n}}\right)\right)$$

Furthermore, we also obtain a lower bound for $\eta_{\mathcal{S}^c}^2$. Similar as in Lemma 5/6 Wang et al. (2022), we can lower bound (using again the monotonicity of $\frac{\|\gamma(\alpha)\|_2}{\alpha}$ and the fact that any feasible $\alpha \leq \alpha_{\overline{m}}$)

$$\eta_{\mathcal{S}^c}^2 \geq \min_b\left[b^2\|\gamma(\alpha_{\overline{m}})\|_2^2 \text{ s.t } b^2 \geq \frac{\zeta_f - \mathcal{O}_c - \mathcal{O}_f}{\frac{\|h\|_\infty^2}{n}(1+\mathcal{O}_b) - \frac{1}{2}\zeta_{\eta\eta}\|\gamma(\alpha_{\overline{m}})\|_2^2}\right]$$

$$= \frac{\zeta_f - \mathcal{O}_c - \mathcal{O}_f}{\frac{\|h\|_\infty^2}{n}(1+\mathcal{O}_b) - \frac{1}{2}\zeta_{\eta\eta}\|\gamma(\alpha_{\overline{m}})\|_2^2}\|\gamma(\alpha_{\overline{m}})\|_2^2 \geq \frac{4\zeta_f}{\zeta_{\eta\eta}}\frac{1}{t_{m_n}^2}\left(1 - \frac{\lambda}{2\sqrt{t_{m_n}}} + O\left(\frac{1}{t_{m_n}}\right)\right)$$

**Step 3.2: Upper bound on $\eta_{\mathcal{S}}$.** In order to upper bound $\eta_{\mathcal{S}}$ we further constrain $\Gamma_3^{\eta_{\mathcal{S}}c}$ and define a set:

$$\Gamma_3^{\eta_{\mathcal{S}}} = \left\{ (\nu, b, \alpha, \eta_{\mathcal{S}}) \ \text{s.t} \ \frac{1}{n}b^2 \|h\|_\infty^2 (1 + \mathcal{O}_b) \geq \zeta_f + \frac{1}{2}\zeta_{\eta\eta}b^2 \|\gamma(\alpha)\|_2^2 + \frac{1}{2}\zeta_{\varsigma\varsigma}\eta_{\mathcal{S}}^2 - \mathcal{O}_c - \mathcal{O}_f \right.$$

$$\left. \text{and} \ \ b\alpha \leq M \right\}$$

Note that $\Gamma_3^{\eta_{\mathcal{S}}} \subset \Gamma_3^{\eta_{\mathcal{S}}c}$ and thus we can use bounds $\underline{m}, \overline{m}$ from the previous part. Upper bounding $\eta_{\mathcal{S}}$ by other variables from the first constraint of $\Gamma_3^{\eta_{\mathcal{S}}}$ and using that $\frac{1}{\alpha^2}$ and $-\frac{\|\gamma(\alpha)\|_2^2}{\alpha^2}$ are monotonically decreasing and increasing in $\alpha$, respectively, we obtain the following high probability bound:

$$\eta_{\mathcal{S}}^2 \leq \frac{2}{\zeta_{\eta\eta}} \left( \frac{M^2}{n} \left( \frac{\|h\|_\infty^2}{\alpha_{\underline{m}}^2}(1 + \mathcal{O}_b) - \frac{1}{2}\zeta_{\eta\eta}n\frac{\|\gamma(\alpha_{\overline{m}})\|_2^2}{\alpha_{\overline{m}}^2} \right) - \zeta_f + \mathcal{O}_c + \mathcal{O}_f \right)$$

$$= \frac{2\zeta_f}{\zeta_{\eta\eta}} \left[ \frac{1}{t_{m_n}^2} \left( 1 - \frac{2}{t_{m_n}^2} + \frac{\tilde{c}}{t_{m_n}^3} \right) \left( t_{\underline{m}}^2 \left( 1 + \frac{4}{t_{\underline{m}}^2} + \frac{c_2}{t_{m_n}^3} \right) - \frac{2m_n}{\overline{m}} \left( 1 + \frac{c_3}{t_{\overline{m}}^2} \right) \right) - 1 \right] + O\left( \frac{1}{t_{m_n}^3} \right)$$

where the second line follows again from concentration results from Proposition 3. Multiplying all the terms gives $\eta_{\mathcal{S}}^2 \lesssim \frac{1}{t_{m_n}^{5/2}}$, as we wanted to show.

Note that we could prove in the exact same way that $\triangle\nu^2 = O\left( \frac{1}{t_{m_n}^{5/2}} \right)$, but this does not change tightness of our result in Theorem 2 and hence we skip this step and conclude the proof of Proposition 7.

# D  Technical Lemmas

## D.1  Application of CGMT: Proof of Proposition 1

The proof essentially follows exactly the same steps as in Koehler et al. (2021) and (Donhauser et al., 2022) except for a few simple modifications, which we describe next.

In order to apply Lemma 1 we first rewrite $\Phi_N$ using the Lagrange multipliers $v \in \mathbb{R}^n$ as follows:

$$\Phi_N = \min_w \max_{v \geq 0} \|w\|_1 + \langle v, 1 - D_y Xw \rangle$$

$$= \min_{(w_\|, w_\perp)} \max_{v \geq 0} \|w_\| + w_\perp\|_1 + \langle v, 1 - D_y X_\| w_\| \rangle - \langle v, D_y X_\perp w_\perp \rangle$$

where $D_y = \text{diag}(y_1, y_2, \ldots, y_n)$. Since $D_y$ and $X_\perp$ are independent, we note that $D_y X_\perp \in \mathbb{R}^{n \times d}$ has i.i.d. entries distributed according to the standard normal distribution, and hence $D_y X_\perp \stackrel{d}{=} X_\perp$ with $\stackrel{d}{=}$ denoting equivalence of random variables in distribution. When comparing the expression obtained with the definition of $\Phi$ from Lemma 1, it is obvious that we should take $X_1 := X_\perp, w_1 := w_\perp, w_2 := w_\|$ and the function $\psi(w, v) := \|w_\| + w_\perp\|_1 + \langle v, 1 - D_y Xw_\| \rangle$, which is a continuous convex-concave function on the whole domain since every norm is a convex function. Motivated by expression for $\phi$ from Lemma 1, we further define

$$\tilde{\phi}_N := \min_{(w_\|, w_\perp)} \max_{v \geq 0} \|w_\| + w_\perp\|_1 + \langle v, 1 - D_y X_\| w_\| \rangle - \|w_\perp\|_2 \langle v, g \rangle - \|v\|_2 \langle w_\perp, h \rangle$$

$$= \min_{(w_\|, w_\perp)} \max_{\lambda \geq 0} \|w_\| + w_\perp\|_1 - \lambda \left( \langle w_\perp, h \rangle - \left\| \left( 1 - D_y X_\| w_\| - g\|w_\perp\|_2 \right)_+ \right\|_2 \right)$$

$$= \min_{(w_\|, w_\perp)} \|w_\| + w_\perp\|_1 \ \text{s.t} \ \langle w_\perp, h \rangle \geq \left\| \left( 1 - D_y X_\| w_\| - g \|w_\perp\|_2 \right)_+ \right\|_2$$

where in the second equality we set $\lambda := \|v\|_2$. Define $w_\perp^{(\mathcal{S})} = \Pi_\mathcal{S} w_\perp$, $w_\perp^{(\mathcal{S}^c)} = \Pi_{\mathcal{S}^c} w_\perp$ where $\Pi_\mathcal{S}$ and $\Pi_{\mathcal{S}^c}$ are projections on $\mathrm{supp}(w^*)$ and the other $d - s$ entries, respectively. So we can rewrite $\tilde{\phi}_N$ as:

$$\tilde{\phi}_N = \min_{(w_\|, w_\perp^{(\mathcal{S})}, w_\perp^{(\mathcal{S}^c)})} \|w_\| + w_\perp^{(\mathcal{S})}\|_1 + \|w_\perp^{(\mathcal{S}^c)}\|_1$$

$$\text{s.t} \quad \langle w_\perp^{(\mathcal{S})}, h_1 \rangle + \langle w_\perp^{(\mathcal{S}^c)}, h_2 \rangle \geq \|(1 - D_y X_\| w_\| - g\sqrt{\|w_\perp^{(\mathcal{S})}\|_2^2 + \|w_\perp^{(\mathcal{S}^c)}\|_2^2})_+\|_2$$

with $h_1 \sim \mathcal{N}(0, I_s)$ and $h_2 \sim \mathcal{N}(0, I_{d-s})$, independent of each other. Under the constraint that $\langle w_\perp^{(\mathcal{S})}, h_1 \rangle + \langle w_\perp^{(\mathcal{S}^c)}, h_2 \rangle \geq 0$ we can square the last inequality and scale with $\frac{1}{n}$ to obtain the following RHS:

$$\frac{1}{n}\|(1 - D_y X_\| w_\| - g\sqrt{\|w_\perp^{(\mathcal{S})}\|_2^2 + \|w_\perp^{(\mathcal{S}^c)}\|_2^2})_+\|_2^2$$

$$= \frac{1}{n}\sum_{i=1}^n (1 - \xi_i \mathrm{sgn}(\langle (x_\|)_i, w_*^{(\mathcal{S})} \rangle)\langle (x_\|)_i, w_\| \rangle - g_i \|w_\perp\|_2)_+^2,$$

which is exactly the function $f_n(\langle w_\|, w^* \rangle, \|w_\perp\|_2)$, as defined in Equation (5). Therefore, comparing with the expression for $\phi_N$ from Proposition 1 we note that $\tilde{\phi}_N \equiv \phi_N$.

In order to complete the proof of the proposition, we need to discuss the compactness of the feasible sets in the optimization problem so that we can apply Lemma 1 to $\Phi_N$ and $\phi_N$. For this purpose, we define the following truncated optimization problems $\Phi_N^r(t)$ and $\phi_N^r(t)$ for some $r, t \geq 0$:

$$\Phi_N^r(t) := \min_{\|w\|_1 \leq t} \max_{\substack{\|v\| \leq r \\ v \geq 0}} \|w\|_1 + \langle v, 1 - D_y X w \rangle$$

$$\phi_N^r(t) := \min_{\|w_\| + w_\perp^{(\mathcal{S})}\|_1 + \|w_\perp^{(\mathcal{S}^c)}\|_1 \leq t} \max_{0 \leq \lambda \leq nr} \|w_\| + w_\perp^{(\mathcal{S})}\|_1 + \|w_\perp^{(\mathcal{S}^c)}\|_1$$

$$- \lambda \left( \frac{1}{n}(\langle w_\perp^{(\mathcal{S})}, h_1 \rangle + \langle w_\perp^{(\mathcal{S}^c)}, h_2 \rangle) - \sqrt{f_n(w)} \right).$$

By definition it follows that $\phi_N^{r_1}(t) \geq \phi_N^{r_2}(t)$ for any $r_1 \geq r_2$, and thus we have that

$$\mathbb{P}(\phi_N \geq t|\xi) \geq \lim_{r \to \infty} \mathbb{P}(\phi_N^r(t) \geq t|\xi). \tag{49}$$

Furthermore, by making use of the simple (linear) dependency on $\lambda$ in the optimization objective in the definition of $\Phi_N$, a standard limit argument as in the proof of Lemma 7 in Koehler et al. (2021) shows that:

$$\lim_{r \to \infty} \mathbb{P}(\Phi_N^r(t) > t|\xi) = \mathbb{P}(\Phi_N > t|\xi).$$

Finally, the proof follows when noting that we can apply Lemma 1 directly to $\Phi_N^r(t)$ and $\phi_N^r(t)$ for any $r, t \geq 0$, which gives us $\mathbb{P}(\Phi_N^r > t|\xi) \leq 2\mathbb{P}(\phi_N^r \geq t|\xi)$. Combining the last inequality with Equations (49) and D.1 completes the proof for $\Phi_N$.

The proof for $\Phi_+$ and $\Phi_-$ uses the same steps as discussed above. We only detail the proof for $\Phi_-$ here, as the proof for $\Phi_+$ follows from the exact same reasoning.

Now, let $MB_1 = \{w \in \mathbb{R}^d : \|w\|_1 \leq M\}$ be an $\ell_1$-ball of radius $M$ and note that we optimize over $(w_\|, w_\perp^{(\mathcal{S})}, w_\perp^{(\mathcal{S}^c)}) \in S_w$ where $S_w = \{w \text{ s.t } \|w\|_2 \geq \delta\} \cap MB_1$ is a compact set. Furthermore, define the function $\psi$ by $\psi(w, v) := \frac{\langle w_\|, w_*^{(\mathcal{S})} \rangle}{\|w\|_2} + \langle v, 1 - D_y X_\| w_\| \rangle$, which is a continuous function on $S_w$ since $\|w\|_2 \geq \delta$. Similarly as above, we can overcome the issue of the compactness of the set $S_v$ by using a truncation argument as proposed in Lemma 4 in Koehler et al. (2021). In particular, we define

$$\Phi_-^r := \min_{w \in S_w} \max_{\substack{\|v\| \leq r \\ v \geq 0}} \frac{\langle w, w^* \rangle}{\|w\|_2} + \langle v, 1 - D_y X w \rangle,$$

$$\phi_-^r := \min_{w \in S_w} \max_{0 \leq \lambda \leq nr} \frac{\langle w_\|, w^* \rangle}{\|w\|_2} - \lambda \left( \frac{1}{n}(\langle w_\perp^{(\mathcal{S})}, h_1 \rangle + \langle w_\perp^{(\mathcal{S}^c)}, h_2 \rangle) - \sqrt{f_n(w)} \right).$$

for which we have

$$\mathbb{P}(\Phi_- < t|\xi) \leq \lim_{r\to\infty} \mathbb{P}(\Phi_-^r < t|\xi) \quad \text{and} \quad \lim_{r\to\infty} \mathbb{P}(\phi_-^r \leq t|\xi) = \mathbb{P}(\phi_- \leq t|\xi).$$

We note that the first statement follows from the definition of $\Phi_-$ and the monotonicity of $\Phi_-^r$ in $r$, while the second statement follows from a limit argument as in Lemma 4 in Koehler et al. (2021). Finally, we conclude the proof by applying the first part of Lemma 1 to $\Phi_-^r$ and $\phi_-^r$ and defining $z^{(1)} = \langle X_\|, w^* \rangle$ with $X_\|$ the row-wise projection of $X$ in the subspace spanned by $w^*$

### D.2 Lower bounds for $f_n$ in noiseless setting

Recall that $\nu = \langle w_\|, w_*^{(\mathcal{S})} \rangle, \eta_{\mathcal{S}} = \|w_\perp^{(\mathcal{S})}\|_2, \eta_{\mathcal{S}^c} = \|w_\perp^{(\mathcal{S}^c)}\|_2$ and $\eta = \|w_\perp\|_2 = \sqrt{\eta_{\mathcal{S}}^2 + \eta_{\mathcal{S}^c}^2}$. In the noiseless setting we defined the following two functions:

$$f_n(\nu,\eta) = \frac{1}{n}\sum_{i=1}^n (1 - \nu|z_i^{(1)}| - z_i^{(2)}\eta)_+^2$$

$$f(\nu,\eta) = \mathbb{E}f_n(\nu,\eta) = \mathbb{E}_{Z^{(1)},Z^{(2)}\sim\mathcal{N}(0,1)}(1 - \nu|Z^{(1)}| - Z^{(2)}\eta)_+^2.$$

In this section we show multiple lower bounds of $f_n$. First, we show a bound with non-tight constants and then show a tight result based on uniform convergence of $f_n$ to $f$. At the end we give a corollary of the uniform convergence proposition which is used in the proof of the Proposition 5.

**Lower bounding $f_n$ with non-tight constants**

We show the following proposition:

**Proposition 8.** *Assume that $\nu$ satisfies $c_1 \leq \nu \leq \nu_{\max}$ for some universal constant $c_1 > 0$. There exist universal constants $\kappa_1, \kappa_2, c_2$ such that for any $\nu, \eta$ that satisfy the given assumption, the inequality*

$$f_n(\nu,\eta) \geq \kappa_1 \frac{1}{\nu} + \kappa_2 \frac{\eta^2}{\nu}$$

*holds with probability $\geq 1 - 2\exp\left(-c_2\frac{n}{(\nu_{\max})^2}\right)$ over the draws of $z^{(1)}, z^{(2)}$.*

*Proof.* Similarly to the above, we have the following:

$$f_n(\nu,\eta) = \frac{1}{n}\sum_{i=1}^n (1 - \nu|z_i^{(1)}| - z_i^{(2)}\eta)_+^2 \geq \frac{1}{n}\sum_{i=1}^n (1 - \nu|z_i^{(1)}| + c_1\eta)_+^2 \mathbb{1}\{z_i^{(2)} \leq -c_1\}$$

$$\gtrsim \frac{1}{n}\sum_{i=1}^n (1 - \nu|z_i^{(1)}| + c_1\eta)^2 \mathbb{1}\{1 - \nu|z_i^{(1)}| \geq \frac{1}{2}, z_i^{(2)} \leq -c_1\}$$

$$\gtrsim (1 + \eta^2)\frac{1}{n}\sum_{i=1}^n \mathbb{1}\{1 - \nu|z_i^{(1)}| \geq \frac{1}{2}, z_i^{(2)} \leq -c_1\}$$

Moreover, from independence of $Z^{(1)}$ and $Z^{(2)}$, the fact that $\mathbb{P}\left(Z^{(2)} \leq -c_1\right) = \Phi^{\complement}(c_1) \geq c_2$ and concentration of Bernoulli random variables we obtain that $f_n(\nu,\eta) \gtrsim (1 + \eta^2)\frac{1}{n}\sum_{i=1}^n \mathbb{1}\{1 - \nu|z_i^{(1)}| \geq \frac{1}{2}\}$ with probability $\geq 1 - \exp(-c_3 n)$. Now in order to lower bound the last term we note that:

$$\mathbb{P}\left(|Z^{(1)}| \leq \frac{1}{2\nu}\right) = \text{erf}\left(\frac{1}{2\sqrt{2}\nu}\right) \gtrsim \frac{1}{\nu}$$

where we used Taylor approximation $\text{erf}\left(\frac{1}{2\sqrt{2}\nu}\right) \gtrsim \frac{1}{\nu}$ for any $\nu \geq c_1$ with $c_1 > 0$ sufficiently large. From Lemma 5 with $\epsilon \asymp \sqrt{n}/\nu_{\max}$ we obtain that uniformly over $\nu, \eta$ $f_n(\nu,\eta) \gtrsim \frac{1}{\nu} + \frac{\eta^2}{\nu}$ with probability at least $1 - 2\exp(-c_2 n/(\nu_{\max})^2)$.

$\square$

**Uniform convergence of $f_n$ to $f$**

Similarly as in Section D we define a random variable $X = (Z^{(1)}, Z^{(2)})$ and a set of functions $\mathcal{G}_0 := \{(Z^{(1)}, Z^{(2)}) \mapsto (1 - \nu|Z^{(1)}| - Z^{(2)}\eta)_+^2 \, | \nu_{\max} \geq \nu \geq \nu_{\min}, \eta \leq \eta_{\max}\}$ with $\nu_{\min} = \Theta(\nu_{\max}), \nu_{\min} = \Omega(n^{1/6})$ and $\eta_{\max} \leq c_2$ for some universal constant $c_2 > 0$. Using notation of Section A.2 we have that $Pg_{\nu,\eta} = \mathbb{E}g_{\nu,\eta}(Z^{(1)}, Z^{(2)}) = f(\nu, \eta)$ and $P_n g_{\nu,\eta} = f_n(\nu, \eta)$, we show the following result:

**Proposition 9.** *There exist positive universal constants $c_1, c_2, c_3 > 0$ such that for any $\epsilon \gtrsim \frac{\log n}{\sqrt{n}}$ holds*

$$\mathbb{P}\left( \|P_n - P\|_{\mathcal{G}_0} \leq c_1 \frac{\log n}{\sqrt{n}} + \epsilon \right) \geq 1 - c_2 \exp\left(-c_3 n \epsilon^2\right).$$

*Proof.* The proof is based on Theorem 3. We choose $\alpha = 1$ and show that the condition from Theorem 3 requiring finite Orlicz norms is satisfied for this choice of $\alpha$. We divide the proof into three steps, where in a first step we bound the variable $\psi_{\mathcal{G}_0}$, then we bound $\mathcal{R}_n(\mathcal{G}_0)$, and finally we bound $\sigma_{\mathcal{G}_0}^2$ and apply Theorem 3.

**Step 1: Bounding $\psi_{\mathcal{G}_0}$** By the definition of Orlicz norms, $\psi_{\mathcal{G}_0}$ is given by:

$$\psi_{\mathcal{G}_0} = \inf\{\lambda > 0 : \ \mathbb{E}[\exp(\frac{1}{\lambda} \max_{1 \leq i \leq n} \sup_{g_{\nu,\eta} \in \mathcal{G}_0} \frac{1}{n}|g_{\nu,\eta}(z_i^{(1)}, z_i^{(2)}) - \mathbb{E}[g_{\nu,\eta}]| - 1]) \leq 1\} \tag{50}$$

Note that $(1 - \nu|z^{(1)}|)_+ \leq 1$ and thus we have $g_{\nu,\eta}(z^{(1)}, z^{(2)}) = (1 - \nu|z^{(1)}| - z^{(2)}\eta)_+^2 \lesssim 1 + (z^{(2)})^2 \eta^2$ for any $z^{(1)}, z^{(2)}, \eta, \nu$, implying that

$$\max_i \sup_{\nu,\eta} |g_{\nu,\eta}(z_i^{(1)}, z_i^{(2)})| = \max_i \sup_{\nu,\eta} |(1 - \nu|z_i^{(1)}| - z_i^{(2)}\eta)_+^2| \leq c_1 z_{\max}^{(2)}$$

with vector $z_{\max}^{(2)} = \max_{1 \leq i \leq n} |z_i^{(2)}|$. Furthermore, it also holds $\mathbb{E}[g_{\nu,\eta}] \lesssim 1 + \eta^2 \mathbb{E}(Z^{(2)})^2 \leq 1 + \eta_{\max}^2 \leq c_3$ for some universal constant $c_3 > 0$.

Using these results and applying the triangle inequality, the term inside of expectation in Equation (50) can be bounded as:

$$\mathbb{E}\left[ \exp\left( \frac{1}{\lambda} \max_i \sup_{\nu,\eta} \frac{1}{n}\Big|(1 - \nu|z_i^{(1)}| - z_i^{(2)}\eta)_+^2 - \mathbb{E}[(1 - \nu|Z^{(1)}| - Z^{(2)}\eta)_+^2]\Big| \right) \right]$$

$$\leq \mathbb{E}\left[ \exp\left( \frac{1}{n\lambda} \max_i \sup_{\nu,\eta}(1 - \nu|z_i^{(1)}| - z_i^{(2)}\eta)_+^2 \right) \right]$$

$$\cdot \exp\left( \frac{1}{n\lambda} \sup_{\nu,\eta} \mathbb{E}[(1 - \nu|Z^{(1)}| - Z^{(2)}\eta)_+^2] \right) \leq \mathbb{E}\left[ \exp\left( \frac{c_1}{n\lambda} z_{\max}^2 \right) \right] \exp\left( \frac{c_3}{n\lambda} \right) \tag{51}$$

for some positive universal constants $c_1, c_3$. Now we split the expectation from the above inequality into two terms:

$$\mathbb{E}\left[ \mathbb{1}\Big[z_{\max} < \sqrt{2\log(n)}\Big] \exp\left( \frac{c_1}{n\lambda} z_{\max}^2 \right) \right] \leq \exp\left( \frac{2c_1 \log n}{n\lambda} \right)$$

and

$$\mathbb{E}\left[ \mathbb{1}\Big[z_{\max} \geq \sqrt{2\log n}\Big] \exp\left( \frac{c_1}{n\lambda} z_{\max}^2 \right) \right] = 2n\mathbb{E}\left[ \mathbb{1}\Big[z_{\max} = |z_1|, |z_1| \geq \sqrt{2\log n}\Big] \exp\left( \frac{c_1}{n\lambda} z_1^2 \right) \right]$$

$$\lesssim n \int_{z_1 = \sqrt{2\log n}}^{\infty} \int_{-z_1}^{z_1} \cdots \int_{-z_1}^{z_1} \exp\left( \frac{c_1}{n\lambda} z_1^2 \right) \left[ \prod_{i=2}^{2n} \frac{\exp(-\frac{1}{2}z_i^2)}{\sqrt{2\pi}} dz_i \right] dz_1$$

$$\lesssim n \int_{\sqrt{2\log n}}^{\infty} \exp\left( -z_1^2 \left( \frac{1}{2} - \frac{c_1}{n\lambda} \right) \right) dz_1 \lesssim \frac{\exp\left( \frac{2c_1 n}{n\lambda} \right)}{\sqrt{\log n}(1 - \frac{2c_1}{n\lambda})} \tag{52}$$

where we assumed that $\lambda > \frac{2c_1}{n}$. Now choosing $\lambda = c_\lambda \frac{\log n}{n}$ with a positive constant $c_\lambda$ sufficiently large, we find that the condition in Equality (50) is satisfied for this $\lambda$, which implies that $\psi_{\mathcal{G}_0} \leq c_\lambda \frac{\log n}{n}$.

**Step 2: Bounding** $\mathcal{R}_n(\mathcal{G}_0)$   In order to apply Theorem 3 we need to upper bound $\mathbb{E}\left\|P_n - P\right\|_{\mathcal{G}_0}$. Since $\mathbb{E}\left\|P_n - P\right\|_{\mathcal{G}_0} \leq 2\mathcal{R}_n(\mathcal{G}_0)$, we can instead upper bound the Rademacher complexity $\mathcal{R}_n(\mathcal{G}_0)$, which we do next. Recall the definition of the Rademacher complexity:

$$\mathcal{R}_n(\mathcal{G}_0) = \mathbb{E}\left[\sup_{g_{\nu,\eta}\in\mathcal{G}_0}\left|\frac{1}{n}\sum_{i=1}^{n}\epsilon_i g_{\nu,\eta}(z_i^{(1)}, z_i^{(2)})\right|\right] \tag{53}$$

Define random variable $\tilde{z} := |z^{(1)}|1\{|z^{(1)}| \leq \frac{1+\eta_{\max}\sqrt{3\log n}}{\nu_{\min}}\}$ and note that for all $\nu, \eta$ and $1 \leq i \leq n$ holds

$$(1-\nu|z_i^{(1)}| - z_i^{(2)}\eta)_+^2 1\{z_{\max}^{(2)} \leq \sqrt{3\log n}\} = (1-\nu\tilde{z}_i - z_i^{(2)}\eta)_+^2 1\{z_{\max}^{(2)} \leq \sqrt{3\log n}\}.$$

We now apply the triangle inequality to Equation (53) to obtain:

$$\mathbb{E}\sup_{\nu,\eta}\left|\frac{1}{n}\sum_{i=1}^{n}\epsilon_i(1-\nu|z_i^{(1)}| - z_i^{(2)}\eta)_+^2\right| \leq \mathbb{E}\sup_{\nu,\eta}\left|\frac{1}{n}\sum_{i=1}^{n}\epsilon_i(1-\nu\tilde{z}_i - z_i^{(2)}\eta)_+^2 1\{z_{\max}^{(2)} \leq \sqrt{3\log n}\}\right|$$

$$+ \mathbb{E}\sup_{\nu,\eta}\left|\frac{1}{n}\sum_{i=1}^{n}\epsilon_i(1-\nu|z_i^{(1)}| - z_i^{(2)}\eta)_+^2 1\{z_{\max}^{(2)} > \sqrt{3\log n}\}\right| \tag{54}$$

Then, using that $(\cdot)_+$ is 1-Lipschitz, we can bound expectation of the first term from Equation (54) as follows:

$$\mathbb{E}\sup_{\nu,\eta}\left|\frac{1}{n}\sum_{i=1}^{n}\epsilon_i(1-\nu\tilde{z}_i - z_i^{(2)}\eta)_+^2 1\{z_{\max}^{(2)} \leq \sqrt{3\log n}\}\right|$$

$$\lesssim \mathbb{E}\sup_{\nu,\eta}\left|\frac{1}{n}\sum_{i=1}^{n}\epsilon_i(1-\nu\tilde{z}_i - z_i^{(2)}\eta)^2 1\{z_{\max}^{(2)} \leq \sqrt{3\log n}\}\right|$$

$$= \mathbb{E}\sup_{\nu,\eta}\left|\frac{1}{n}\sum_{i=1}^{n}\epsilon_i\left[(1-\nu\tilde{z}_i)^2 - 2(1-\nu\tilde{z}_i)z_i^{(2)}\eta + (z_i^{(2)})^2\eta^2\right]1\{z_{\max}^{(2)} \leq \sqrt{3\log n}\}\right|$$

We use again the triangle inequality and consider each of the three terms above:

- Note that $|\nu\tilde{z}_i| \leq \frac{\nu_{\max}}{\nu_{\min}}\eta_{\max}\sqrt{3\log n} \lesssim \sqrt{\log n}$ and using concentration of sub-exponential random variables from Lemma 4 we obtain:

$$\mathbb{E}\sup_{\nu}\left|\frac{1}{n}\sum_{i=1}^{n}\epsilon_i(1-\nu\tilde{z}_i)^2 1\{z_{\max}^{(2)} \leq \sqrt{3\log n}\}\right| \leq \mathbb{E}\sup_{\nu}\left|\frac{1}{n}\sum_{i=1}^{n}\epsilon_i\nu^2\tilde{z}_i^2 1\{z_{\max}^{(2)} \leq \sqrt{3\log n}\}\right|$$

$$+ \mathbb{E}\sup_{\nu}\left|\frac{1}{n}\sum_{i=1}^{n}\epsilon_i(-2\nu\tilde{z}_i)1\{z_{\max}^{(2)} \leq \sqrt{3\log n}\}\right| + \mathbb{E}\left|\frac{1}{n}\sum_{i=1}^{n}\epsilon_i 1\{z_{\max}^{(2)} \leq \sqrt{3\log n}\}\right| \lesssim \frac{\log n}{\sqrt{n}}$$

- Similarly as in the previous case, we use triangle inequality to split expectation into two terms and then use that $|z_i^{(2)}\eta| \leq z_{\max}^{(2)}\eta_{\max} \lesssim \sqrt{\log n}$ and $|\nu\tilde{z}_i z_i^{(2)}\eta| \leq 3\frac{\nu_{\max}}{\nu_{\min}}\eta_{\max}^2\log n \lesssim \log n$, and apply concentration from Lemma 4 to get:

$$\mathbb{E}\sup_{\nu,\eta}\left|\frac{1}{n}\sum_{i=1}^{n}2\epsilon_i(1-\nu\tilde{z}_i)z_i^{(2)}\eta 1\{z_{\max}^{(2)} \leq \sqrt{3\log n}\}\right| \lesssim \frac{\log n}{\sqrt{n}}$$

- Last, use that $\eta^2(z_i^{(2)})^2 \leq \eta_{\max}^2(z_{\max}^{(2)})^2 \lesssim \log n$, and again concentration of sub-exponential random variables from Lemma 4 to obtain:

$$\mathbb{E}\sup_{\eta}\left|\frac{1}{n}\sum_{i=1}^{n}\epsilon_i(z_i^{(2)})^2\eta^2 1\{z_{\max}^{(2)} \leq \sqrt{3\log n}\}\right| \lesssim \frac{1}{\sqrt{n}}$$

Thus, we bounded the first term from Equation (54). Now, we bound the second term. Since $|\epsilon_i(1-\nu|z_i^{(1)}| - z_i^{(2)}\eta)_+^2| \leq (1+z_i^{(2)}\eta)^2$ we obtain:

$$\mathbb{E}\sup_{\nu,\eta}\left|\frac{1}{n}\sum_{i=1}^n \epsilon_i(1-\nu|z_i^{(1)}| - z_i^{(2)}\eta)_+^2 1\{z_{\max}^{(2)} > \sqrt{3\log n}\}\right|$$

$$\lesssim \mathbb{E}\sup_{\eta}\frac{1}{n}\sum_{i=1}^n (1+z_i^{(2)}\eta)^2 1\{z_{\max}^{(2)} > \sqrt{3\log n}\}$$

$$\lesssim \frac{1}{n}\mathbb{E}\sum_{i=1}^n (1+(z_i^{(2)})^2)1\{z_{\max}^{(2)} > \sqrt{3\log n}\} \lesssim \mathbb{E}\left[(z_{\max}^{(2)})^2 1\{z_{\max}^{(2)} > \sqrt{3\log n}\}\right]$$

$$\lesssim n\int_{z_1=\sqrt{3\log n}}^{\infty} z_1^2 \exp(-z_1^2/2)dz_1 \lesssim \frac{\sqrt{\log n}}{\sqrt{n}}$$

where in the last step we used the same approach as for obtaining Equation (52). After adding all terms, we obtain $\mathcal{R}_n(\mathcal{G}_0) \lesssim \frac{\log n}{\sqrt{n}}$.

**Step 3: Proof of the statement** To apply Theorem 3, we also need to bound the variance $\sigma_{\mathcal{G}_0}^2$. But, it is straightforward that there exists some positive universal constant $c_{\sigma_{\mathcal{G}_0}} > 0$ such that the variance is bounded as follows:

$$\sigma_{\mathcal{G}_0}^2 \leq \sup_{g_{\nu,\eta}\in\mathcal{G}_0}\mathbb{E}\left[g_{\nu,\eta}^2\right] \leq c_{\sigma_{\mathcal{G}_0}}\left(1+\eta_{\max}^4\right)$$

Substituting all derived bounds into the probability statement from Theorem 3 we obtain for $\epsilon \gtrsim \frac{\log n}{\sqrt{n}}$:

$$\mathbb{P}\left(\|P_n - P\|_{\mathcal{G}_\sigma} \geq 2(1+t)\mathcal{R}_{\mathcal{G}_\sigma} + \epsilon\right) \leq \exp\left(-c_2 n\epsilon^2\right) + 3\exp\left(-c_3\frac{n\epsilon}{\log n}\right) \leq c_4\exp(-c_2 n\epsilon^2)$$

with $c_2^{-1} = 2(1+\delta)c_{\sigma_{\mathcal{G}_0}}\left(1+\eta_{\max}^4\right)$ and $c_3^{-1} = Cc_\lambda$, which concludes the proof. $\square$

**Corollary 1.** *There exist positive universal constants $c_1, c_2$ such that for any $\nu, \eta$ satisfying constraint in $\mathcal{G}_0$ and $\epsilon \gtrsim \frac{\log n}{\sqrt{n}}$, inequality*

$$f_n(\nu,\eta) \geq \frac{\sqrt{2}}{3\sqrt{\pi}}\frac{1}{\nu} + \sqrt{\frac{2}{\pi}}\frac{\eta^2}{\nu} - \epsilon$$

*holds with probability at least $1 - c_1\exp(-c_2 n\epsilon^2)$ over the draws of $z^{(1)}, z^{(2)}$.*

*Proof.* Recall that $f(\nu,\eta) = \mathbb{E}[f_n(\nu,\eta)]$. From Proposition 9 we have $f_n(\nu,\eta) \geq f(\nu,\eta) - \epsilon$ uniformly over all admissible $(\nu,\eta)$ with probability $\geq 1 - c_1\exp(-c_2 n\epsilon^2)$. According to Lemma 8, $f$ is an infinitely differentiable function and thus we can express it by Taylor series. First, we determine the coefficients of the series of $f(\nu,\cdot): \eta \mapsto f(\nu,\eta)$.

The constant coefficient is given by:

$$f(\nu,0) = \mathbb{E}(1-\nu|Z^{(1)}|)_+^2 = \frac{2}{\sqrt{2\pi}}\int_0^{1/\nu}(1-\nu z)^2\exp\left(-\frac{z^2}{2}\right)dz$$

$$= (\nu^2+1)\mathrm{erf}\left(\frac{1}{\sqrt{2}\nu}\right) + \sqrt{\frac{2}{\pi}}\nu\left(\exp\left(-\frac{1}{2\nu^2}\right) - 2\right) = \frac{\sqrt{2}}{3\sqrt{\pi}}\frac{1}{\nu} + O\left(\frac{1}{\nu^3}\right)$$

where we used the Taylor expansion around 0 for functions erf and exp. The first derivative coefficient is given by

$$\frac{\partial}{\partial\eta}f(\nu,\eta)|_{\eta=0} = -2\mathbb{E}[Z^{(2)}(1-\nu|Z^{(1)}| - \eta Z^{(2)})_+]|_{\eta=0} = 0$$

since $Z^{(1)}$ and $Z^{(2)}$ are independent random variables and $\mathbb{E}[Z^{(2)}] = 0$. Now consider the second derivative coefficient:

$$\frac{\partial^2}{\partial \eta^2} f(\nu, \eta)|_{\eta=0} = 2\mathbb{E}\left[1\{1 - \nu|Z^{(1)}| - \eta Z^{(2)}\}(Z^{(2)})^2\right]|_{\eta=0} = 2\mathbb{P}\left(|Z^{(1)}| \leq \frac{1}{\nu}\right)$$

$$= 2\mathrm{erfc}\left(\frac{1}{\sqrt{2}\nu}\right) = 2\sqrt{\frac{2}{\pi}}\frac{1}{\nu} + O\left(\frac{1}{\nu^3}\right)$$

where in the last step we used the Taylor series approximation of the error function around zero. Now, in order to analyze higher order derivatives, we show using Leibniz integral rule that:

$$\frac{\partial^3}{\partial \eta^3} f(\nu, \eta) = \frac{2}{\pi}\frac{\partial}{\partial \eta}\int_{Z^{(2)}=-\infty}^{1/\eta}\int_{Z^{(1)}=0}^{(1-\eta Z^{(2)})/\nu}(Z^{(2)})^2 \exp\left(-\frac{1}{2}(Z^{(2)})^2\right)\exp\left(-\frac{1}{2}(Z^{(1)})^2\right)dZ^{(1)}dZ^{(2)}$$

$$= -\frac{2}{\pi\nu}\int_{Z^{(2)}=-\infty}^{1/\eta}(Z^{(2)})^3 \exp\left(-\frac{1}{2}(Z^{(2)})^2\right)\exp\left(-\frac{1}{2}\left(\frac{1-\eta Z^{(2)}}{\nu}\right)^2\right)dZ^{(2)} \tag{55}$$

Now, note that for higher order derivatives, the term that comes from differentiating the upper bound $1/\eta$ is equal 0 for $\eta = 0$ since it is of the form $\mathrm{poly}(1/\eta)\exp(-1/(2\eta^2))$ which is zero for any polynomial. Thus, the main term which we need to consider comes from the term $\exp\left(-\frac{1}{2}\left(\frac{1-\eta Z^{(2)}}{\nu}\right)^2\right)$. Note that after taking the differential with respect to this term, we obtain an additional multiplicative factor $1/\nu^2$. However, we also obtain the multiplicative term $(1 - \nu Z^{(2)})$, which can be further differentiated with respect to $\eta$. Taking all this into account one can show that for $k = 2, 3, \ldots$

$$\frac{\partial^{2k}}{\partial \eta^{2k}} f(\nu, \eta)\bigg|_{\eta=0} =$$

$$O\left(\frac{1}{\nu^{2k-1}}\int_{Z^{(2)}=-\infty}^{1/\eta}(Z^{(2)})^{2k}(1 - \eta Z^{(2)})\exp\left(-\frac{1}{2}(Z^{(2)})^2\right)\exp\left(-\frac{1}{2}\left(\frac{1-\eta Z^{(2)}}{\nu}\right)^2\right)dZ^{(2)}\bigg|_{\eta=0}\right)$$

with all other terms either vanishing at $\eta = 0$ or having in front of the integral multiplicative constant $\frac{1}{\nu^p}$ with $p > 2k-1$. Thus, for $\eta = 0$, using that the Gaussian moments are bounded, we obtain $\frac{\partial^{2k}}{\partial \eta^{2k}} f(\nu, \eta)|_{\eta=0} = O\left(\frac{1}{\nu^{2k-1}}\right)$. Similarly to Equation (55), one can show that every odd differential at $\eta = 0$ is equal to the scaled odd moments of the standard Gaussian random variable, implying that $\frac{\partial^{2k+1}}{\partial \eta^{2k+1}} f(\nu, \eta)|_{\eta=0} = 0$.

Taking all derived coefficients into consideration, we can express $f$ using the following Taylor series:

$$f(\nu, \eta) = \frac{\sqrt{2}}{3\sqrt{\pi}}\frac{1}{\nu} + \sqrt{\frac{2}{\pi}}\frac{\eta^2}{\nu} + O\left(\frac{1}{\nu^3}, \frac{\eta^4}{\nu^3}\right) \tag{56}$$

At the end, since $\eta = O(1)$ and $\nu = \Omega(n^{1/6})$ we have $O\left(\frac{1}{\nu^3}, \frac{\eta^4}{\nu^3}\right) = o(\epsilon)$, which finishes the proof. □

### D.3 Lower bounds for $f_n$ in noisy setting

Recall that we have defined $\nu = \langle w_\parallel, w^* \rangle, \eta_{\mathcal{S}} = \|w_\perp^{(\mathcal{S})}\|_2, \eta_{\mathcal{S}^c} = \|w_\perp^{(\mathcal{S}^c)}\|_2$ and $\eta = \|w_\perp\|_2 = \sqrt{\eta_{\mathcal{S}}^2 + \eta_{\mathcal{S}^c}^2}$, and also the following two functions:

$$f_n(\nu, \eta) = \frac{1}{n}\sum_{i=1}^n (1 - \xi_i\nu|z_i^{(1)}| - z_i^{(2)}\eta)_+^2$$

$$f(\nu, \eta) = \mathbb{E}f_n(\nu, \eta) = \mathbb{E}_{Z^{(1)}, Z^{(2)}\sim\mathcal{N}(0,1)}\mathbb{E}_{\xi_{\mathrm{RV}}\sim\mathbb{P}(\cdot|Z^{(1)})}(1 - \xi_{\mathrm{RV}}\nu|Z^{(1)}| - Z^{(2)}\eta)_+^2. \tag{57}$$

In this section we show three lower bounds for $f_n$ of increasing tightness. First, we show a lower bound by a quadratic form in $\nu$ and $\eta$, after that we bound $f_n$ by a sum of a quadratic form and a constant, and the last bound is based on the uniform convergence of $f_n$ to $f$ which we prove at the end of this subsection.

**Lower bounding $f_n$ by a quadratic form**

We show the following lemma.

**Lemma 6.** *There exist universal positive constants $c_\nu, c_\eta$ only depending on $\mathbb{P}_\sigma$ and $c$ such that for any $\nu, \eta$ we have that:*

$$f_n(\nu, \eta) \geq c_\nu \nu^2 + c_\eta \eta^2$$

*with probability at least $1 - \exp\left(-cn\right)$ over the draws of $z^{(1)}, z^{(2)}, \xi$.*

*Proof.* We can assume that $\nu \geq 0$ since the other cases follow exactly from the same argument. First, we show an auxiliary statement which we use later in the proof. Namely, we claim that there exists some positive constant $c_1$ such that for all $z \in [z_1, z_2]$, $\mathbb{P}_\sigma\left(\xi = -1; z\right) > c_1$ for some $z_1, z_2 \in \mathbb{R}$ and $z_1 \neq z_2$. Let us prove this statement by contradiction and assume that there exists no $z \in [z_1, z_2]$ that satisfies the previous equation. Then, for almost any $z \sim \mathcal{N}(0, 1)$, we have $\mathbb{P}_\sigma(\xi; z) = +1$ and hence the minimum of the function $f(\nu, \eta) = \mathbb{E}f_n(\nu, \eta)$ is obtained for $\nu = \infty$. However, this is in contradiction with Assumption 1 in Section 3.2. Hence there exists some $z$ for which $\mathbb{P}\left(\xi = -1; z\right) > c_1$. By the assumption on $\mathbb{P}_\sigma$ in Section 3.2 we assume piecewise continuity of $z \to \mathbb{P}_\sigma(\xi = -1; z)$ and hence there exists some interval $[z - \delta, z + \delta] =: [z_1, z_2]$ in which the given probability is bounded away from zero.

We can assume without loss of generality that this interval does not contain zero, since in that case we can always define a new interval of the form $[\epsilon, z_2]$ or $[z_1, -\epsilon]$ for $\epsilon > 0$ small enough, which does not contain zero. Let us define $\tilde{z} = \min\{|z_1|, |z_2|\}$.

We can now bound $f_n(\nu, \eta)$ as follows:

$$f_n(\nu, \eta) = \frac{1}{n}\sum_{i=1}^{n}(1 - \xi_i\nu|z_i^{(1)}| - z_i^{(2)}\eta)_+^2$$

$$\geq \frac{1}{n}\sum_{i=1}^{n}1\{\xi_i = -1, z_i^{(1)} \in [z_1, z_2], z_i^{(2)} < -c_2\}(1 - \xi_i\nu|z_i^{(1)}| - z_i^{(2)}\eta)_+^2$$

$$\geq (1 + \tilde{z}\nu + c_2\eta)^2 \frac{1}{n}\sum_{i=1}^{n}1[\xi_i = -1, z_i^{(0)} \in [z_1, z_2], z_i^{(1)} < -c_2]$$

From Section 4.2 we have that $Z^{(2)}$ is independent of $\xi_{\mathrm{RV}}$ and $Z^{(1)}$. Hence:

$$\mathbb{P}(\xi_{\mathrm{RV}} = -1, Z^{(1)} \in [z_1, z_2], Z^{(2)} < -c_2)$$
$$= \mathbb{P}(\xi_{\mathrm{RV}} = -1|Z^{(1)} \in [z_1, z_2])\mathbb{P}(Z^{(1)} \in [z_1, z_2])\mathbb{P}(Z^{(2)} < -c_2) \geq c_1\left(\Phi^{\complement}(z_1) - \Phi^{\complement}(z_2)\right)\Phi^{\complement}(c_2) \geq c$$

for some positive universal constant $c$. Now using concentration of i.i.d. Bernoulli random variables we obtain:

$$f_n(\nu, \eta) \geq (1 + \tilde{z}\nu + c_2\eta)^2 \frac{c}{2} \gtrsim \nu^2 + \eta^2$$

with probability at least $1 - \exp\left(-cn\right)$. □

**Lower bounding $f_n$ by a quadratic form with constant**

Recall that $\triangle\nu = \nu - \nu_f$. We show the following lemma.

**Lemma 7.** *Let $B_\nu, B_\eta > 0$ be universal positive constants. Then, there exist positive constants $\widetilde{c}_\nu, \widetilde{c}_\eta > 0$ and $c_1, c_2, c_3 > 0$ only depending on $\mathbb{P}_\sigma$, such that for any $\epsilon \geq \frac{c_1}{\sqrt{n}}$ and any $\nu^2 \leq B_\nu^2, \eta \leq B_\eta$ we have that:*

$$f_n(\nu, \eta) \geq \zeta_f + \widetilde{c}_\nu\left(\triangle\nu\right)^2 + \widetilde{c}_\eta\eta^2 - \epsilon$$

*with probability at least $1 - \exp\left(-c_2n\epsilon^2\right) - \exp\left(-c_3\frac{n\epsilon}{\log n}\right)$ over the draws of $z^{(1)}, z^{(2)}, \xi$.*

*Proof.* First note that from the uniform convergence result in Proposition 10 we have that $f(\nu, \eta) \geq f_n(\nu, \eta) - \epsilon$, with $f$ from Equation (57), with high probability. Thus, it is sufficient to study $f$. Clearly, by the convexity of $f$ we have that $f \geq \zeta_f$ with $\zeta_f = f(\nu_f, 0)$ where we use the simple fact that $(\nu_f, 0)$ is the global minimizer of $f$, which follows from the assumption on $\mathbb{P}_\sigma$ in Section 3.2. Furthermore, it is not difficult to check that for for any $\nu, \eta$, $\nabla^2 f(\nu, \eta) \succ 0$ and therefore, $f$ is strictly convex on every compact set. Hence, the proof follows. $\qquad\square$

**Uniform convergence of $f_n$ to $f$**

Recall that $Z^{(1)}, Z^{(2)} \sim \mathcal{N}(0,1)$ are independent Gaussian random variables and $\xi_{\mathrm{RV}}$ a random variable with $\xi_{\mathrm{RV}}|Z^{(1)} \sim \mathbb{P}_\sigma(.; Z^{(1)})$. Using notation introduced in Section A.2 with random variable $X = (Z^{(1)}, Z^{(2)}, \xi_{\mathrm{RV}})$, and $\mathcal{G}_\sigma = \{g_{\nu, \eta} \mid |\nu| \leq B_\nu, \eta \leq B_\eta\}$, we note that

$$Pg_{\nu, \eta} = \mathbb{E}g_{\nu, \eta}(Z^{(1)}, Z^{(2)}, \xi_{\mathrm{RV}}) = f(\nu, \eta) \quad \text{and} \quad P_n g_{\nu, \eta} = f_n(\nu, \eta).$$

We show the following result:

**Proposition 10.** *There exist positive universal constants $c_1, c_2, c_3 > 0$ such that*

$$\mathbb{P}\left(\|P_n - P\|_{\mathcal{G}_\sigma} \leq \frac{c_1}{\sqrt{n}} + \epsilon\right) \geq 1 - \exp\left(-c_2 n \epsilon^2\right) - \exp\left(-c_3 \frac{n\epsilon}{\log n}\right)$$

*Proof.* The proof of the proposition is based on the application of Theorem 3 and follows exactly the same steps as proof of Proposition 9. In order to apply Theorem 3 we need to upper bound three terms - $\psi_{\mathcal{G}_\sigma}, \sigma^2_{\mathcal{G}_\sigma}$ and $\mathcal{R}_n(\mathcal{G}_\sigma)$. Similarly as in proof of Proposition 9 we split proof into three steps:

**Step 1: Bounding $\psi_{\mathcal{G}_\sigma}$**   Recall the definition of $\psi_{\mathcal{G}_\sigma}$ from Theorem 3:

$$\psi_{\mathcal{G}_\sigma} = \inf\{\lambda > 0 : \ \mathbb{E}[\exp(\frac{1}{\lambda} \max_i \sup_{\nu, \eta} \frac{1}{n}|g_{\nu, \eta}(z_i^{(1)}, z_i^{(2)}, \xi_i) - \mathbb{E}[g_{\nu, \eta}]| - 1]) \leq 1\}$$

Since $|\nu|, \eta$ are bounded by constants, we have that

$$\mathbb{E}[g_{\nu, \eta}] = \mathbb{E}[(1 - \xi_{\mathrm{RV}} \nu |Z^{(1)}| - Z^{(2)} \eta)^2_+] \leq c(1 + B_\nu^2 + B_\eta^2) \leq c_2 \tag{58}$$

for some positive universal constants $c_2$ that may depend on $B_\nu, B_\eta$. Furthermore, we have:

$$(1 - \xi_i \nu |z_i^{(1)}| - z_i^{(2)} \eta)^2_+ \leq c(1 + (B_\nu^2 + B_\eta^2) z_{\max}^2) \leq c_1 z_{\max}^2 \tag{59}$$

where $z_{\max} = \max_{1 \leq i \leq 2n}\{|z_i^{(1)}|, |z_i^{(2)}|\}$. Similarly to inequality (51), we apply the triangle inequality and bound the two terms using Equations (58) and (59) to obtain:

$$\mathbb{E}\left[\exp\left(\frac{1}{\lambda} \max_i \sup_{\nu, \eta} \frac{1}{n}\left|(1 - \xi_i \nu |z_i^{(1)}| - z_i^{(2)} \eta)^2_+ - \mathbb{E}[(1 - \xi_{\mathrm{RV}} \nu |Z^{(1)}| - Z^{(2)} \eta)^2_+]\right|\right)\right]$$
$$\leq \mathbb{E}\left[\exp\left(\frac{c_1}{n\lambda} z_{\max}^2\right)\right] \exp\left(\frac{c_2}{n\lambda}\right)$$

Thus we obtain that $\psi_{\mathcal{G}_\sigma} \leq \inf\{\lambda > 0 : \ \mathbb{E}[\exp(\frac{c_1}{n\lambda} z_{\max}^2) \exp(\frac{c_2}{n\lambda}) - 1] \leq 1\}$, which is similar to expression (51) in the proof of Proposition 9. Hence following the same argument we conclude that $\psi_{\mathcal{G}_\sigma} \leq c_\lambda \frac{\log n}{n}$ for some universal constant $c_\lambda > 0$.

**Step 2: Bounding $\mathcal{R}_n(\mathcal{G}_\sigma)$**   The upper bound on the Rademacher complexity is derived as follows. First use the fact that $(\cdot)_+$ is 1-Lipschitz to obtain:

$$\mathcal{R}_n(\mathcal{G}_\sigma) = \mathbb{E}\left[\sup_{g_{\nu, \eta} \in \mathcal{G}_\sigma}\left|\frac{1}{n}\sum_{i=1}^n \epsilon_i g_{\nu, \eta}(z_i^{(1)}, z_i^{(2)}, \xi_i)\right|\right]$$
$$\leq 2\mathbb{E}\left[\sup_{|\nu| \leq B_\nu, \eta \leq B_\eta}\left|\frac{1}{n}\sum_{i=1}^n \epsilon_i (1 - \xi_i \nu |z_i^{(1)}| - z_i^{(2)} \eta)^2\right|\right], \tag{60}$$

then expand quadratic form and apply triangle inequality for every term to obtain that (60) is upper bounded by:

$$2\mathbb{E}\left[\left|\frac{1}{n}\sum_{i=1}^{n}\epsilon_i\right|\right] + 2\mathbb{E}\left[\sup_{|\nu|\le B_\nu, \eta\le B_\eta}\left|\frac{1}{n}\sum_{i=1}^{n}\epsilon_i 2\xi_i \nu|z_i^{(1)}||z_i^{(2)}\eta)\right|\right]$$

$$+2\mathbb{E}\left[\sup_{\eta\le B_\eta}\left|\frac{1}{n}\sum_{i=1}^{n}\epsilon_i(-2z_i^{(2)}\eta)\right|\right] + 2\mathbb{E}\left[\sup_{\eta\le B_\eta}\left|\frac{1}{n}\sum_{i=1}^{n}\epsilon_i(z_i^{(2)})^2\eta^2\right|\right]$$

$$+2\mathbb{E}\left[\sup_{|\nu|\le B_\nu}\left|\frac{1}{n}\sum_{i=1}^{n}\epsilon_i(-2\xi_i\nu|z_i^{(1)}|)\right|\right] + 2\mathbb{E}\left[\sup_{|\nu|\le B_\nu}\left|\frac{1}{n}\sum_{i=1}^{n}\epsilon_i\nu^2(z_i^{(1)})^2\right|\right]$$

Finally, since sums above do not depend on $\nu$ and $\eta$ any more, we can use standard concentration results for sub-exponential random variables to obtain that $\mathcal{R}_n(\mathcal{G}_\sigma) \lesssim \frac{1}{\sqrt{n}}$.

**Step 3: Proof of the statement**   Similarly to Equation (58), we can bound the variance straightforwardly as follows:

$$\sigma_{\mathcal{G}_\sigma}^2 \le \sup_{g_{\nu,\eta}\in\mathcal{G}_\sigma}\mathbb{E}\left[g_{\nu,\eta}^2\right] \le c_{\sigma_{\mathcal{G}_\sigma}}\left(1 + B_\nu^4 + B_\eta^4\right)$$

for some positive universal constant $c_{\sigma_{\mathcal{G}_\sigma}} > 0$.

Combining all derived bounds and using that $\mathbb{E}\|P_n - P\|_{\mathcal{G}_\sigma} \le 2\mathcal{R}_n(\mathcal{G}_\sigma)$ we obtain from Theorem 3:

$$\mathbb{P}\left(\|P_n - P\|_{\mathcal{G}_\sigma} \ge 2(1+t)\mathcal{R}_{\mathcal{G}_\sigma} + \epsilon\right) \le \exp\left(-c_2 n\epsilon^2\right) + 3\exp\left(-c_3\frac{n\epsilon}{\log n}\right)$$

with $c_2^{-1} = 2(1+\delta)c_{\sigma_{\mathcal{G}_\sigma}}\left(1 + B_\nu^4 + B_\eta^4\right)$ and $c_3^{-1} = Cc_\lambda$, which concludes the proof.

□

## D.4   Additional lemmas

**Lemma 8.** *The function* $(\nu, \eta) \mapsto \mathbb{E}_{Z^{(1)}, Z^{(2)}\sim\mathcal{N}(0,1)}(1 - \nu|Z^{(1)}| - Z^{(2)}\eta)_+^2$ *is an infinitely differentiable function. Furthermore, under Assumption 1 from Section 2, the function* $(\nu, \eta) \mapsto \mathbb{E}_{Z^{(1)}, Z^{(2)}\sim\mathcal{N}(0,1)}$ $\mathbb{E}_{\xi_{\mathrm{RV}}\sim\mathbb{P}(\cdot|Z^{(1)})}(1 - \xi_{\mathrm{RV}}\nu|Z^{(1)}| - Z^{(2)}\eta)_+^2$ *is also an infinitely differentiable function.*

*Proof.* Note that the conditional expectation of the first function is given by:

$$\mathbb{E}_{Z^{(2)}|Z^{(1)}=z^{(1)}}[(1 - \nu|z^{(1)}| - \eta Z^{(2)})_+^2]$$

$$= \int_{-\infty}^{\frac{1}{\eta}(1-\nu|z^{(1)}|)}\frac{1}{\sqrt{2\pi}}\exp\left(-\frac{1}{2}(z^{(2)})^2\right)(1 - \nu|z^{(1)}| - \eta z^{(2)})^2 dz^{(2)}$$

$$= \eta(1 - \nu|z^{(1)}|)\exp\left(-\frac{1}{2\eta^2}(1 - \nu|z^{(1)}|)^2\right) + ((1 - \nu|z^{(1)}|)^2 + \eta^2)\Phi\left(\frac{1}{\eta}(1 - \nu|z^{(1)}|)\right),$$

which is an infinitely differentiable function in $\nu$ and $\eta$. Since the function given in the lemma is an expectation of an infinitely differentiable function, it is also infinitely differentiable, which finishes the first part of the proof.

Now, note that using Assumption 1 we can rewrite the second function as:

$$\mathbb{E}_{Z^{(1)}}\left[\mathbb{P}(\xi_{\mathrm{RV}} = 1|Z^{(1)})\mathbb{E}_{Z^{(2)}|Z^{(1)}}[(1 - \nu|Z^{(1)}| - \eta Z^{(2)})_+^2]\right.$$

$$\left. + \mathbb{P}(\xi_{\mathrm{RV}} = -1|Z^{(1)})\mathbb{E}_{Z^{(2)}|Z^{(1)}}[(1 + \nu|Z^{(1)}| - \eta Z^{(2)})_+^2]\right].$$

But, similarly to above, we can show that $\mathbb{E}_{Z^{(2)}|Z^{(1)}}[(1 + \nu|Z^{(1)}| - \eta Z^{(2)})_+^2]$ is infinitely differentiable, implying that the whole function is also infinitely differentiable. □

