# OpenReview forum: "Tight bounds for maximum $\ell_1$-margin classifiers"
_TMLR — Withdrawn by Authors_

### Review · Reviewer_XY3q · 2023-07-08

**Summary Of Contributions:**

This submission studies the statistical properties of max $l_1$-margin classifiers for binary classification. On one hand, authors investigate whether max $l_1$-margin classifiers can achieve a statistical error that only depends on $|| w^* ||_0$, i.e., the hard sparsity, as other more advanced algorithms as in one-bit compressive-sensing literature. The answer is negative, as the authors show the lower bound of $||w^*||_1^{2/3} / n^{1/3}$ even with noiseless labels. On the other hand, when there are random label noises, the statistical rate can decay independently of $||w^*||_1$, even though at a very slow sub-logarithmic rate (if I understand it correctly). The proof of the two theorems is the entirety of the paper.

**Audience:**

Yes

**Broader Impact Concerns:**

This is a purely theoretical paper, and there are no ethical concerns.

**Claims And Evidence:**

Yes

**Requested Changes:**

Below are some suggestions + questions for me to better understand this paper. But I believe if authors can address them, it would make the paper stronger.

- It is quite counter-intuitive and confusing in the way that two results are presented. Even for the noiseless setting, we obtain a negative result, but then for the noisy setting, how we can expect something positive in which sense? How does the noise help? I can see that the logarithmic decay in Theorem 3.2 is extremely slower than $n^{-1/3}$ in Theorem 3.1, so there seems to be some subtlety in interpreting two theorems, but I wish this explanation be much more crisp. Besides, why we cannot obtain the same sub-logarithmic rate for the noiseless case?

- At some point in the proof sketch, it would have been nicer if there is a part that explains why $n^{-1/3}$ is the right statistical rate, why we expect some poly dependence on $\|w^*\|_1$, and so on. Similarly, $\log^{-1/2} (d/n)$ does look quite an arbitrary poly-logarithmic rate, but still, the theorem says this is tight, so it would have been much appreciated if the paper explains the meaning of this rate.


**Strengths And Weaknesses:**

I couldn't quite follow the message from the second result (Theorem 3.2) in this paper, so my evaluation is mostly based on the first result (Theorem 3.1).

Strength

- It is an interesting study to investigate whether the simplest algorithm for a particular problem can achieve the best performance. The authors showed both tight upper and lower bounds (for this specific method), and this is a solid contribution to the literature.


Weakness

- The proof is very long and technical, and it is hard to digest as little intuition is provided.


Overall, I believe this is a good paper, but it would be a very hard paper to read. Please see below the requested changes.

---

> ### Author Response · Authors · 2023-08-10
> **Rebuttal**
>
> We would like to thank the reviewer for their comments on the presentation of the paper.
>
> **Length of the Proof and lack of Intuition:**
>
>
> > The proof is very long and technical, and it is hard to digest as little intuition is provided.
>
>
> Indeed, we acknowledge that the proof is long and technical, but we believe that this is not per se a weakness as it was necessary to get tight bounds on the estimator, which have not been found by previous works before. We would also prefer a simpler proof, and we would of course appreciate any idea for how to shorten it.
>
> Regarding efforts to convey intuition: We tried our hardest in our extensive proof sketch to provide as much intuition as possible as to how we could improve previous analyses. If the reviewer finds specific paragraphs/sentences there unclear, we would be happy to revise them.
>
> > At some point in the proof sketch, it would have been nicer if there is a part that explains why n^{-⅓} is the right statistical rate, why we expect some poly dependence on $\|w^\star\|_1$, and so on.
>
> >Similarly, $log^{-1/2}(d/n)$  does look quite an arbitrary poly-logarithmic rate, but still, the theorem says this is tight, so it would have been much appreciated if the paper explains the meaning of this rate.
>
> Concerning the intuition: the exact rates are the result from a lengthy technical analysis and follow from careful concentration arguments. We highlight here that we provide both matching high probability upper and lower bounds, which is rather uncommon in the literature and requires a very careful (technical) analysis for which intuition is much harder to come by. In particular, in contrast to worst-case minimax lower bounds which are standard in the literature, our high probability lower bounds are both algorithm dependent and apply to every instance of $w^*$ with high probability.
>
> **Comparing Theorems 3.1 and 3.2:**
>
>
> > “It is quite counter-intuitive and confusing in the way that two results are presented. Even for the noiseless setting, we obtain a negative result, but then for the noisy setting, how we can expect something positive in which sense? How does the noise help? I can see that the logarithmic decay in Theorem 3.2 is extremely slower than $n^{−1/3}$ in Theorem 3.1, so there seems to be some subtlety in interpreting two theorems, but I wish this explanation be much more crisp. Besides, why we cannot obtain the same sub-logarithmic rate for the noiseless case?”
>
> Thanks for raising this concern, which is helpful for us - we’ll try to address it now and revise the manuscript accordingly to be more clear. We would like to refer the reviewer to the discussion addressing all reviewers and in particular the paragraph **Presentation of the main results**
>
> May we ask if this answers the reviewer's question?

---

### Review · Reviewer_yu67 · 2023-07-25

**Summary Of Contributions:**

The authors of this study investigate the adaptivity to sparsity of the (interpolating) maximum $\ell_1$-margin classifier in high-dimensional discriminative learning tasks. The main result, presented as in Theorem 1, demonstrates that the maximum $\ell_1$-norm classifier fails to adapt to the sparsity of the ground truth. Theorem 1 also highlights the high-probability lower bound, which interestingly matches the tight upper bound. This observation indicates that the upper bound in Theorem 1 cannot be improved under the sparsity assumption.

Additionally, the authors explore the effects of noise by considering a comprehensive and general noise model. Within this framework, they establish the vanishing convergence rates for the maximum $\ell_1$-margin classifier. Importantly, their findings unveil the presence of benign overfitting for this classifier, a crucial discovery reported here for the first time.


**Audience:**

Yes

**Claims And Evidence:**

Yes

**Requested Changes:**

1.	The conditions in Theorem 2 requiring $\kappa_2 n \leq d$ indicate that the dimension $d$ needs to be of the same or larger order than the sample size $n$. This setting corresponds to the super high-dimensional case, where the number of features is relatively large compared to the number of samples. However, a natural question arises about the behavior of the convergence rate when $d$ is a constant or of low order in $n$, represented as $d = n^{\delta}$ with $\delta \in [0,1)$.
Investigating the convergence rate under such settings with a limited number of features relative to the sample size could be a valuable direction for revision. It would provide a deeper understanding of the model's performance in scenarios where data dimensionality is not extremely high, potentially revealing insights that might differ from the super high-dimensional case.

2.	In the presence of noise, it would be beneficial to consider modifications to the maximum $\ell_1$ margin classifier, like incorporating an $\ell_1$-norm constraint, to tackle the overfitting phenomenon. By alleviating overfitting, there is a possibility of achieving faster convergence rates. I believe that exploring these modifications and understanding their effects on the convergence rates could significantly enhance the quality and effectiveness of your research.



**Strengths And Weaknesses:**

Strengths：
1.	Theorem 1 introduces a high-probability lower bound for the maximum $\ell_1$ margin classifier, a novel and rarely seen result in the existing literature.
2.	The matching upper and lower bounds in Theorem 1 offer crucial insights into the convergence rates of the maximum $\ell_1$ margin classifier under the sparsity assumption. This observation demonstrates that the achieved convergence rate cannot be improved, providing a clear understanding of the limitations and capabilities of the classifier in such scenarios. Moreover, the faster convergence rate compared to the result presented by Chinot et al. (2021) by a logarithm factor showcases the superiority stemming from sparsity.
3.	This paper for the first time investigates the maximum $\ell_1$ margin classifier in the presence of a constant fraction of corrupted samples.

Weakness:
Theorems 1 and 2 both present negative results regarding convergence rates. Specifically, Theorem 1 establishes a convergence rate of order $O(n^{-1/3})$, which is slower compared to the rate of $O(n^{-1/2})$ demonstrated in previous works by Zhang et al. (2014) and Awasthi et al. (2016). Meanwhile, the rates in Theorem 2 are only of logarithmic order, falling short of being min-max optimal. While the slower rate in Theorem 2 could be attributed to interpolation with noisy data, it raises an intriguing question: Can the rates in Theorem 2 be improved to approach the min-max optimal lower bound of $O(n^{-1/2})$?

---

> ### Author Response · Authors · 2023-08-10
> **Rebuttal**
>
> We would like to thank the reviewer for their in-depth review and for appreciating our work.
>
> **Regarding $d \asymp n^{\delta}$ for $\delta \in [0,1)$ for Theorem 2:**
>
>
> >How does the convergence rate behave when $d$ is a constant or of low order in $n$, represented as $d=n^\delta$ with $\delta \in [0,1)$? Investigating the convergence rate under such conditions with a limited number of features relative to the sample size could be a promising direction for revision.
>
>
> Thanks for the question. We would like to refer to our discussion about the choice of the regime in the general comments. As a bottomline, we consider $d>n$ since in this paper we analyze the overparameterized regime where interpolation is possible even for noisy data.
>
>
> **Incorporating Modifications to Tackle Overfitting:**
>
>
> > In the presence of noise, it would be beneficial to consider modifications to the maximum $\ell_1$ margin classifier, like incorporating an $\ell_1$-norm constraint, to tackle the overfitting phenomenon. By alleviating overfitting, there is a possibility of achieving faster convergence rates. I believe that exploring these modifications and understanding their effects on the convergence rates could significantly enhance the quality and effectiveness of your research.
>
> Indeed, “modifications” such as regularization are well-studied and known to successfully combat overfitting and lead to minimax optimal estimators. This is already extensively studied for the Lasso in regression but also for classification (see e.g. Zhang et al. 2014, etc). We would like to emphasize that the primary objective of this paper (and the line of work on “benign overfitting”) is to study overparameterized **interpolators** that fit the data perfectly - their behavior is much less studied and understood in the literature and has received considerate attention only recently. In our case of linear classification, we study the minimum-$\ell_1$-norm interpolator, which is obtained when running coordinate descent (or Adaboost) to convergence, as detailed in the Introduction. One critical observation we've made is that this classifier, while capable of fitting even noisy labels, can asymptotically (high-dimensional asymptotics, when $d>n$ and $n->\infty$) learn the underlying ground truth. This capability of a classifier to perform well despite fitting to noisy data is referred to as “benign overfitting” in the community (see Bartlett et al. 2020).
>
> We would further like to emphasize that we do not advocate for the use of maximum-$\ell_1$-margin classifiers in practical settings, especially in the presence of noise. For practical purposes, it is always best to use regularization with cross-validated hyperparameter search.  In contrast, the goal of this paper (and others in our area) is to shed light on when interpolators can lead to how good of a performance theoretically. In particular, a comprehensive understanding of the maximum-$\ell_1$-margin classifier—both in noiseless and noisy environments—is absent in the existing literature.
>
>
> **Regarding Improvement in Rates in Theorem 2:**
>
> > can the rates in Theorem 2 be improved to approach the min-max optimal lower bound of $O(n^{-1/2})$
>
> We are uncertain that we understand the question correctly: Theorem 2 establishes matching upper and lower bounds for the max-l1-margin classifier, showing that the rates are sharp and cannot be further improved. They’re indeed not minimax optimal, and that’s precisely the gap we find interesting to point out. We’d appreciate it if the reviewer could provide further clarification of their question.

---

### Review · Reviewer_13bU · 2023-07-29

**Summary Of Contributions:**

This paper theoretically investigates the non-asymptotic behavior of prediction error in maximum $\ell_1$-margin classifiers under a hard sparse discriminative regime. Assume $n$ feature vectors in $\mathbb{R}^d$ are sampled from $\mathcal{N}(0,I_d)$ in an i.i.d. manner. Labels $y_i$ are created as $\mathrm{sign}\left(\langle x_i\vert w^* \rangle\right)\zeta_i$ for some ground truth $w^*\in\mathcal{S}^{d-1}$. The i.i.d. random variables $\zeta_i\in \\{-1,+1\\}$ represent the label noise.

Paper has analyzed the generalization error of learning $w^*$ through a maximum $\ell_1$-margin classifier (Page 3 - 3rd Equation) in both noiseless ($\zeta_i=+1$ for all $i$) and noisy regimes. Thm 1 and 2 show that when sparsity of $w^*$, denoted by $s$, is bounded as $s\leq \tilde{O}(n^{2/3})$, and the dimension of the model grows superlinearly/sublinearly as a function of $n$, the generalization error is tightly bounded. More surprisingly, it has been shown that the above mentioned bounds are the same (up to logarithmic factors) to the case where ground truth is not sparse. This indicates that maximum $\ell_1$-margin classifiers fail to "adapt to sparsity" of the ground truth.

This work considers a very specific non-asymptotic regime for parameters $n$, $d$ and $s$. I am not completely familiar with this area of research and cannot assess the generality of such assumptions, and/or how widely such scenarios are being considered in the community. The current work mostly builds upon a previous paper Chinot et al. (2021), which is still on arXiv, by either improving their bounds or analyzing their classifiers in hard sparse regimes. However, the proof techniques seem to significantly deviate from Chinot et al's work. Authors in this paper use Convex Gaussian Minmax Theorem -- (C)GMT as the main tool in their proofs, while previous works seem to mostly rely on geometric approaches. I have not completely read the proofs, however, I have not found any notable technical mistakes so far.

Paper is fairly well-written, however, authors have used several keywords which might not be well-known by researchers from even slightly different areas. I believe the writing of the paper can be improved a lot. However, the most important problem with this work is lack of positioning with several related works which consider different asymptotic/non-asymptotic regimes for the parameters. Authors have not properly justified their assumptions, and not many papers with similar settings have been reviewed. Given that the assuming regime is somehow justified (or is being largely considered by similar works), the paper seems to be a descent purely theoretical work.

**Audience:**

Yes

**Broader Impact Concerns:**

-

**Claims And Evidence:**

Yes

**Requested Changes:**

**Please also read the "Weaknesses" section, for more questions and clarifications which might be required**

The main weakness of the paper is lack of proper poistioning w.r.t. existing works on sparse regression/classification. The paper mostly positions itself as an extension of Chinot et al. (2021), which is still on arXiv. However, there is already a large body of work in this area which achieve better rates of convergence, however, under different assumptions (specially w.r.t. dimension $d$).

In particular, all the bounds presented in the main theorems (i.e., Thm 1 and 2) assume the dimension $d$ to grow linearly, or sublinearly as a function of training dataset size $n$, which is in stark contrast to many existing works in which $d$ is usually kept fixed and the behavior of the generalization error is analyzed as a function of $n$. A natural question would be: What happens to the bounds of Thm. 1 and 2 if $d$ is kept fixed, or grows with a different pace? In other words, what is the motivation behind assuming this regime? How many other works have analyzed the problem under such circumstances? and how the current paper is positioned w.r.t. them? Explanations on page 11 does not seem to be enough, and dddressing these issues help the reader to better evaluate the current work and its contribution.

I think authors should address this issue by reviewing a larger number of related works, and/or giving more explanation (for example, in the introduction section)

**Minor requests**

- It might be good to number all equations (at least during the review process), to make referencing more easier.
- Please explain why the 3rd Equation on P.3 corresponds to the "maximum $\ell_1$-margin classification". In other words, how the optimization maximizes the $\ell_1$ distance of the support vector from the decision boundary?
- Please give more explanations about the "Suboptimality of the maximum $\ell_1$-margin" section on P.4 .

**Strengths And Weaknesses:**

**Strengths**

- Paper is fairly well-written, and except for (IMO) overuse of some specific keywords and terms, paper is easy to read.
- I have not found any notable technical mistakes (have not completely read the proofs)
- The results of Thm 1 and 2 seem to be interesting and useful for the community. I am not completely familiar with this area, however, the results are new as far as I know. The main problem might be the assumptions behind those results (please see the Weaknesses section). In any case, authors have managed to tightly bound (from above and below) the prediction error or maximum $\ell_1$-margin classifiers in both noiseless and noisy regimes.
- Authors claim that Thm 2 is the first to show that prediction error vanishes as $n\rightarrow\infty$ while a constant fraction of labels are noisy. However, they need $d$ to also go to infinity with even a faster rate.
- The techniques used in the proofs seem to be new in this context, which might be interesting for the MLT community.

---
**Weaknesses**

- Paper uses a relatively large body of keywords which might not be familiar for researchers from slightly different areas. For example, "benign overfitting", "adaptive to sparsity" and etc. may need some extra explanations. This would hugely impact the paper's readability.

- The paper heavily relies on the Gaussianity of the feature vectors, i.e., $x_i\sim\mathcal{N}(0,I_d)$ for all $i$. This assumption is more relaxed in some other works, under the general notion of "Gaussian Universality Principle" [1]. Also, as claimed by the authors, Chinot et al. (2021) does not need this assumption, which decreases the level of contribution for this work.

- The most important contribution of this paper is to show that maximum $\ell_1$-margin classifiers do not adapt to sparsity. In other words, the rates of convergence for the prediction error for a hard sparse ground truth $w^*$ stays the same as that of a general (non-sparse) one. However, this exciting result has been achieved under a number of very important (and maybe restrictive?) assumptions: 1) In both noiseless and noisy regimes, $d$ is **required** to grow as a linear/sublinear function of $n$. This assumption obviously degrades the generalization capability, and hence the results of this paper correspond to a very specific non-asymptotic regime, which needs to be further discussed throughout the paper. 2) the assumption on the sparsity level $s$ is too mild (both theorems only need $s\leq \tilde{O}(n^{2/3})$). This might be the reason that the bounds are tight. Question: Does the the error lowerbound decrease significantly if $s$ is more severly restricted?



[1] L. Pesce, F. Krzakala, B. Loureiro, and L. Stephan. Are Gaussian data all you need? Extents and limits of universality in high-dimensional generalized linear estimation. arXiv:2302.08923, 2023.

---

> ### Author Response · Authors · 2023-08-10
> **Rebuttal**
>
> Thanks for your time you took to read the paper and for writing detailed comments and questions. We will answer them in detail in the following paragraphs and would encourage the reviewer to follow up with more clarifying questions should there still be any doubts left after reading our responses.
>
> **High- vs low-dimensional regime and positioning wrt related work:**
>
>
> The reviewer was concerned about growing $d$ in different sections of the review:
>
>
> > Authors claim that Thm 2 is the first to show that prediction error vanishes as n\to \infty while a constant fraction of labels are noisy. However, they need $d$  to also go to infinity with even a faster rate.
>
>
> > However, there is already a large body of work in this area which achieve better rates of convergence, however, under different assumptions (specially w.r.t. dimension d).
>
> >I think authors should address this issue by reviewing a larger number of related works, and/or giving more explanation (for example, in the introduction section)
>
> >How many other works have analyzed the problem under such circumstances? and how the current paper is positioned w.r.t. them?
>
> > In other words, what is the motivation behind assuming this regime (d grows superlinearly in n) ?
>
> We believe that there has been a misunderstanding about the literature our results belong to. We  would like to refer the reviewer to the discussion addressing all reviewers and moreover repeat the main arguments below:
>
> High-dimensional regimes (see e.g. [1] for a good overview), where the dimension $d$ is in the order of $n$ or even larger, have been extensively studied in the statistics literature e.g. for linear regression (see lasso, minimum-norm interpolators, compressive sensing) and linear classification (see sparse logistic regression, maximum-margin classifiers, 1-bit compressive sensing). These problems are considered foundational in high-dimensional statistics and their insights and proof techniques differ vastly from low-dimensional statistics where $d$ is kept fixed. As a consequence, these two communities are often treated as separate fields and are usually not considered as related work.
>
> High-dimensional statistics research has recently been extended by a line of work on (noise-) interpolating overparameterized models motivated by empirical phenomena of large models (e.g., NNs). In particular, minimum-norm interpolators for regression and maximum-margin classifiers have been studied to shed light on these phenomena. Our paper is most related to this large body of work with similar motivation. We have referenced many representative works in this area (see references in “Problem 2” in the introduction). As mentioned above in the response to all reviewers, all of these works focus on overparameterized regimes where $d>n$.
>
> Finally, we remark here that very different tools are necessary for deriving generalization bounds in high-dimensional scenarios compared to low-dimensional ones. We refer the reader to the introduction in [1] for a good overview of high-dimensional statistics and a summary why studying this regime yields important intuitions beyond low-dimensional statistical results.
>
> >What happens to the bounds of Theorem 1 and 2 if $d$ is kept fixed, or grows at a different pace?
>
> We would like to refer the reviewer to  the answer in **Regime for $d,n$ studied in Theorem 3.2. (noisy interpolation)** and **Regime for $d,n$ studied in Theorem 3.1. (noiseless interpolation)**
>
> **Assumption on the sparsity level:**
>
> > the assumption on the sparsity level  is too mild (both theorems only need). This might be the reason that the bounds are tight. Question: Does the the error lower bound decrease significantly if is more severly restricted?
>
> Our lower bound holds for all $s < O(n^{2/3})$, including “more restricted” sparsities such as $s = O(1)$. Note that this is not a minimax lower bound but an instance (hence s) dependent high probability lower bound for the algorithm. We would appreciate it if the reviewer could clarify the question further shall this answer not be addressing their intended question.
>
> **Gaussianity of the data:**
>
> While some works do extend error rates beyond Gaussian data such as the paper the reviewer referred to, they focus on risk convergence in the asymptotic limit. We emphasize that non-asymptotic rates require tighter error control. So far, to the best of our knowledge, there are no  non-asymptotic error rates for minimum-norm interpolators or maximum-margin classifiers with non-Gaussian data, barring the unique case of the minimum-$\ell_2$-norm interpolator.
>
> **Chinot et. al (2021) “still on arxiv”:**
>
> Thanks for pointing this out. By now, the paper is in fact published in a journal and we will update the citation to: Chinot, Geoffrey, et al. "AdaBoost and robust one-bit compressed sensing." Mathematical Statistics and Learning 5.1 (2022): 117-158.

---

> > ### Author Response · Authors · 2023-08-10
> > **Rebuttal Part 2**
> >
> > **Use of keywords:**
> >
> > Thanks for bringing this to our attention. For this purpose we elaborated on the terms “adaptivity to sparsity” and “benign overfitting” right after in the introduction. Their use helps to maintain brevity when referencing the phenomena. Should the reviewers find the explanation in the introduction unclear, we would appreciate it if they could point out what they are still missing.
> >
> > **Further questions:**
> >
> > > Please give more explanations about the "Suboptimality of the maximum $\ell_1$-margin" section on P.4 .
> >
> > Its association with "maximum $\ell_1$-margin classification is due to duality. We'll include a comment for clarity.
> >
> > >Please explain why the 3rd Equation on P.3 corresponds to the "maximum $\ell_1$-margin classification". In other words, how the optimization maximizes the $\ell_1$ distance of the support vector from the decision boundary?
> >
> > Could the reviewer provide more specific feedback or questions?
> >
> >
> > [1] Wainwright, Martin J. High-dimensional statistics: A non-asymptotic viewpoint. Vol. 48. Cambridge university press, 2019.

---

### Author Response · Authors · 2023-08-10
**Rebuttal**

We would like to thank the reviewers and the area chair for dedicating their time to evaluate this paper.


**General comment on the broader research field:**

We believe that there have been some misunderstandings concerning the research field our results belong to and especially with respect  to the high-dimensional regimes studied in this paper.  We therefore would like to give a brief overview of the high-dimensional statistics literature related to this paper.

Broadly speaking, our paper contributes to the (non-asymptotic) high-dimensional statistics literature where both $d$ and $n$ are large, and dependencies on $d$ are explicitly stated in the bounds. Such results are very different in nature to the result from low-dimensional analysis, where $d$ is usually treated as a constant and dependencies on $d$ are often hidden in universal constants. In fact, results from the low-dimensional literature often require that $n >>d $ to be tight and are usually expected to be loose when both $d$ and $n$ are large.

To comment on the relevance of results studying regimes where both  $d$ and $n$ are large: the high-dimensional regime is very well studied for linear models and in particular in the context of sparse estimation/classification problems (e.g. in compressive sensing, 1-bit compressive sensing, LASSO regression). One of the main questions behind these works, and also Problem 1 (Theorem 3.1), is how well a sparse signal can be recovered from data sets where the dimension is much larger than the amount of samples.

Moreover, Problem 2 (Theorem 3.2) belongs to a subfield of high-dimensional statistics that recently emerged and that studies overparameterized models (where number of parameters > samples ) that  **interpolate** (perfect data fit) the training data. This line of research is motivated by the curiously good empirical performance in practice of interpolating neural networks even in the presence of noise (Zhang et al., 2021; Belkin et al., 2019) that could not be explained by classical reasoning on overfitting. Aiming to gain intuition for this phenomenon, recent works study interpolating linear models in the noisy and overparameterized setting;  a linear model is overparameterized when the ambient dimension $d$ is larger than the amount of training samples $n$, i.e. $d > \kappa n$. This subfield  is often also  referred to as  **benign overfitting** (coined by Bartlett et al., 2020) or **harmless interpolation** (coined by Muthukumar et al., 2021).



**Presentation of the main results:**

Based on the response of Reviewer XY3q, we believe that we might not have made it clear how Theorem 3.1 and Theorem 3.2 are related. Hence, we would like to take this opportunity to clarify this question and reiterate the major contribution of the paper.
In this paper, we study linear interpolators in cl assification settings and more precisely the important special case of the maximum $\ell_1$-margin classifier. This classifier is obtained when running  coordinate descent (or Adaboost) until convergence  (see references in the introduction). We present a novel proof technique for studying linear classifiers based on the CGMT (see Section 4) which allows us to answer two open problems at once.

- Problem 1 (Theorem 3.1)  concerns with the adaptivity to sparsity in the noiseless setting. More formally, it asks the question whether the error rates improve when assuming that the ground truth is hard-sparse ( $\ell_0$-norm is at most $s$) vs. the ground truth is soft-sparse ($\ell_0$-norm is at most s). This behavior has been well observed for the minimum-$\ell_1$-norm estimator or basis pursuit estimator, which can be seen as the “corresponding” interpolator in the regression setting. Maybe surprisingly, we show that the maximum $\ell_1$-margin classifier does not adapt to sparsity.

- Problem 2  (Theorem 3.2) concerns showing that the maximum-$\ell_1$-margin classifier is consistent (assuming that $d>>n$, hence in a high-dimensional asymptotic sense)  albeit interpolating a non-vanishing fraction of corrupted labels. Hence, we show bening overfitting for this interpolating classifier; a result which is clearly missing in the current literature.

We want to emphasize at this point that the goal of this paper is not to compare the two results in Theorem 3.1 and 3.2. The reason why we address both problems in one paper is because both results rely on a similar proof technique and consider the same estimator in the overparameterized regime.

---

> ### Author Response · Authors · 2023-08-10
> **Rebuttal Part 2**
>
>
>
> **Regime for $d,n$ studied in Theorem 3.2. (noisy interpolation)**
>
> Interpolating linear classification models in the presence of noise as in Theorem 3.2. only exist if the data is  linearly separable, which  is only guaranteed (almost surely) if $d>n$. This stands in stark contrast to regularized (non-interpolating) linear classifiers which exist for any choice of $d,n$. Such estimators have been widely studied in the literature (e.g. Zhang et al., 2014) and even achieve min-max optimal rates. However, as stated above, in this paper we focus on (overparameterized) models that **interpolate noisy samples**, a field which gained significant popularity within the past 5 years.
>
>
>
>
> **Regime for $d,n$ studied in Theorem 3.1. (noiseless interpolation)**
>
> In stark contrast to the noisy case, in the noiseless case, the maximum-$\ell_1$-margin classifier indeed exists (and is almost surely unique) for all choices of $d,n$. In this paper however, we again focus on the high-dimensional case where $d$ grows with $n$ and the ground truth is sparse. This regime is particularly interesting since it concerns the very well studied problem of how well a sparse signal can be recovered on a high dimensional data set, which is e.g., one of the fundamental problems in (1-bit) compressive sensing.
>
> It is common in the literature to only present high probability upper bounds for the performance of the estimator (see e.g. Chinot et al. (2021), Zhang et al. (2014), Awasthi et al. (2016)), where usually only an assumption on the maximal dimension $d$ is needed. In stark contrast, in this paper we also present high probability lower bounds for the specific algorithm which is uncommon in the literature and requires a very fine grained analysis. In particular, we therefore have to  restrict the analysis to the high-dimensional regime where $d >= n^{2/3}$. We would like to mention here as well that a high probability lower bound for a specific algorithm should not be confused with a minimax lower bound which is an information theoretic bound holding for all algorithms.
>
> Nevertheless, we believe that studying the regime where $d <= n^{2/3}$ is an interesting problem on its own and that there should be a phase transition in the rates occurring around this threshold.  Finally, to comment on the case where the dimension $d$ is fixed and $n$ can be arbitrarily large. In this situation, one could simply apply standard maximum $\ell_2$-margin bounds in the classical low-dimensional regime and make use of the equivalence of norms $\|w\|_2 \leq \|w\|_1 \leq \sqrt{d}\|w\|_2$.

---

### Note · Authors · 2023-09-27

**Comment:**

We thank the reviewers and the AC for spending their time. Since this process took much longer than expected, we decided to withdraw the paper.

**Withdrawal Confirmation:**

I have read and agree with the venue's withdrawal policy on behalf of myself and my co-authors.